# DISTRIBUTIONALLY ROBUST LINEAR REGRESSION WITH BLOCK LEWIS WEIGHTS

**Naren Sarayu Manoj**
Department of Computer Science
Toyota Technological Institute Chicago
Chicago, IL 60637, USA
nsm@ttic.edu

**Kumar Kshitij Patel** *
Institute for Foundations of Data Science
Yale University
New Haven, CT 06510, USA
kkpatel@ttic.edu

## ABSTRACT

We present an algorithm for the empirical group distributionally robust (GDR) least squares problem. Given $m$ groups, a parameter vector in $\mathbb{R}^d$, and stacked design matrices and responses $\mathbf{A}$ and $\boldsymbol{b}$, our algorithm obtains a $(1+\varepsilon)$-multiplicative optimal solution using $\widetilde{O}(\min\{\mathsf{rank}(\mathbf{A}), m\}^{1/3}\varepsilon^{-2/3})$ linear-system-solves of matrices of the form $\mathbf{A}^\top \mathbf{B}\mathbf{A}$ for block-diagonal $\mathbf{B}$. Our technical methods follow from a recent geometric construction, block Lewis weights, that relates the empirical GDR problem to a carefully chosen least squares problem and an application of accelerated proximal methods. Our algorithm improves over known interior point methods for moderate accuracy regimes and matches the state-of-the-art guarantees for the special case of $\ell_\infty$ regression. We also give algorithms that smoothly interpolate between minimizing the average least squares loss and the distributionally robust loss.

## 1 INTRODUCTION

Machine learning algorithms and their training datasets have grown substantially in both size and complexity over the past decade. This increased model complexity has made it challenging to interpret and predict their behavior in unobserved scenarios. Hence, many applications that involve societal decisions still rely on simple, interpretable models like linear regression, often after feature engineering. Examples of such applications include predicting national housing prices, estimating wages across industries, forecasting loan amounts across banks, predicting life insurance premiums across groups, and projecting energy consumption across communities (Cohen-Addad et al., 2024).

A shared safety and sometimes legal concern across the above applications is the potential for wildly different model qualities for different distributions, i.e., outputting a notably worse model for some source data distributions (Data, 2014; Barocas & Selbst, 2016; Hardt et al., 2016; Veale et al., 2018; Selbst et al., 2019; Berk et al., 2021; Corbett-Davies et al., 2023; Chouldechova, 2016; Kleinberg et al., 2018; Agarwal et al., 2019; Cohen-Addad et al., 2024; Song et al., 2024). Specifically, consider fitting a linear model $\boldsymbol{x} \in \mathbb{R}^d$ to make real predictions on some task over $m$ groups where group $i$'s dataset consists of $n_i$ entries and is denoted by $S_i = \{(\boldsymbol{a}_i^j, b_i^j)\}_{j \in [n_i]}$. The *utilitarian* or the total-cost-minimizing objective minimizes the average squared prediction error across groups, i.e.,

$$\min_{\boldsymbol{x} \in \mathbb{R}^d} \frac{1}{m} \sum_{i \in [m]} \frac{1}{n_i} \|\mathbf{A}_{S_i}\boldsymbol{x} - \boldsymbol{b}_{S_i}\|_2^2 \quad, \tag{1}$$

where $\mathbf{A}_{S_i} := [\boldsymbol{a}_i^1 \dots \boldsymbol{a}_i^{n_i}]^\top \in \mathbb{R}^{n_i \times d}$ is the feature matrix and $\boldsymbol{b}_{S_i} := [b_i^1 \dots b_i^{n_i}]^\top \in \mathbb{R}^{n_i}$ is the label vector for group $i \in [m]$.

Due to the inherent heterogeneity of the datasets, the model derived by optimizing the objective equation 1 may be particularly detrimental to some groups, as the prediction error may be disproportionately higher for these groups. To overcome these limitations, the following *egalitarian* or group Distributionally Robust Optimization (DRO) objective has been considered in several recent works (Ben-Tal et al., 2013; Duchi et al., 2016; Sagawa et al., 2019; Levy et al., 2020; Soma et al.,

---

*This work was partly done while the author was a fellow at the Simons Institute for Theory of Computing.

2022; Abernethy et al., 2022; Song et al., 2024),

$$\min_{\boldsymbol{x}\in\mathbb{R}^d} \max_{i\in[m]} \frac{1}{n_i} \left\| \mathbf{A}_{S_i}\boldsymbol{x} - \boldsymbol{b}_{S_i} \right\|_2^2 \quad . \tag{2}$$

Objective 2 is the "fairest" objective among all objectives that balance utility and distributional robustness (Kleinberg et al., 2018; Chouldechova & Roth, 2018; Asadpour et al., 2022; Chen et al., 2022; Rahmattalabi et al., 2019; Golrezaei et al., 2024). Since objective 2 is a convex problem, it is natural to apply standard black-box optimization techniques to solve it. However, we identify several challenges in applying existing methods:

**Efficient first-order algorithms have geometry-dependent rates.** To our knowledge, using an efficient first-order method (such as sub-gradient descent) will incur a geometry-dependent runtime. In particular, if the matrices $\mathbf{A}_{S_i}$ or if the stacked matrix $\mathbf{A} := [\mathbf{A}_{S_1}^\top \ldots \mathbf{A}_{S_M}^\top]^\top$ are poorly conditioned, then this will be reflected accordingly in the convergence rates. This is a drawback of the existing results by Abernethy et al. (2022) and Song et al. (2024).

**Objective equation 2 is not smooth.** Since the objective is the pointwise maximum of several continuous functions, the derivative is not well-defined at the points at which the maximizing function changes. Thus, applying subgradient descent to this objective without a tailored analysis will yield a rather unimpressive $1/\varepsilon^2$ dependence in the iteration complexity.

**Min-max optimization/regret minimization approaches have a $1/\varepsilon^2$ dependence on iteration complexity.** Since problem 2 is a min-max optimization objective, it is also natural to try to use game theory-inspired approaches that use some oracle (such as gradients) for each group as a black box. For instance, we can cast objective 2 as a repeated game between a min player (equipped with a no-regret algorithm) and a max player (equipped with the best response oracle). The main shortcoming of this approach is that even though the function for each group is smooth, the iteration complexity (to get $\varepsilon$ average regret) for smooth online convex optimization still has an unimpressive $1/\varepsilon^2$ dependence (as opposed to $1/\varepsilon$ for smooth convex optimization) (Soma et al., 2022; Zhang et al., 2024a). Thus, this approach is no better than optimizing equation 2 using sub-gradient descent.

**Interior point methods have a poor iteration complexity for large $m$.** Another natural approach (that can partially address the previous two issues), following the discussion by Boyd & Vandenberghe (2004, Section 6.4), is to rewrite problem 2 in its epigraph form and use an interior point method (IPM) to solve the resulting problem (which, in this case, is a quadratically constrained linear program). Unfortunately, this will give an algorithm whose analysis is only known to yield an iteration complexity of $O(\sqrt{m})$, where each iteration solves a linear system in matrices of the form $\mathbf{A}^\top \mathbf{B} \mathbf{A}$ for a block-diagonal $\mathbf{B}$ (see Remark 1.1). A naïve implementation of this algorithm will thus have a superlinear runtime in the number of groups, which is undesirable when the number of groups is large. Alternately, consider an example in which we copy each group $k$ times in the objective. The new objective value does not change from the original objective value, but the iteration complexity from the IPM now blows up to $\sqrt{mk}$. This also signals to us that we should search for an algorithm whose iteration complexity is mostly independent from $m$.

Hence, designing an algorithm without these shortcomings requires novel ideas.

## 1.1 OUR RESULTS

In this paper, we present a new algorithm (Algorithm 1) to approximately optimize objective 2, which addresses the aforementioned difficulties. We state the iteration complexity of our algorithm in the following theorem.

**Theorem 1** (Robust regression). *Let $\mathbf{A}_{S_i} \in \mathbb{R}^{n_i \times d}$ and $\boldsymbol{b}_{S_i} \in \mathbb{R}^{n_i}$ for all $i \in [m]$. Denote their concatenations by $\mathbf{A} := [\mathbf{A}_{S_1}^\top \ldots \mathbf{A}_{S_M}^\top]^\top \in \mathbb{R}^{n \times d}$ and $\boldsymbol{b} := [\boldsymbol{b}_{S_1}^\top \ldots \boldsymbol{b}_{S_M}^\top]^\top \in \mathbb{R}^n$ where $n := \sum_{i \in [m]} n_i$. Let $\varepsilon > 0$. Then Algorithm 1 returns $\widehat{\boldsymbol{x}}$ such that,*

$$\max_{i\in[m]} \frac{1}{\sqrt{n_i}} \left\| \mathbf{A}_{S_i}\widehat{\boldsymbol{x}} - \boldsymbol{b}_{S_i} \right\|_2 \leq (1+\varepsilon) \cdot \min_{\boldsymbol{x}\in\mathbb{R}^d} \max_{i\in[m]} \frac{1}{\sqrt{n_i}} \left\| \mathbf{A}_{S_i}\boldsymbol{x} - \boldsymbol{b}_{S_i} \right\|_2 \quad , \tag{3}$$

*and it runs in*

$$O\left( \frac{\min\{\mathsf{rank}(\mathbf{A}), m\}^{1/3} \left( \log\left(\frac{n\log m}{\varepsilon}\right)^{14/3} + \log(m) \right)}{\varepsilon^{2/3}} \right)$$

*linear-system-solves in matrices of the form $\mathbf{A}^\top \mathbf{B} \mathbf{A}$, where $\mathbf{B}$ is a block-diagonal matrix for which block $i$ has size $n_i \times n_i$.*

We prove Theorem 1 in Appendix C. We compare the guarantee of Theorem 1 against the other baselines in Table 1.

| Algorithm | Iteration Complexity | Each Iteration |
|---|:---:|:---:|
| Subgradient descent | $\dfrac{\|\boldsymbol{x}^\star\|_2 \max_{1 \le i \le m} \frac{1}{\sqrt{n_i}} \|\mathbf{A}_{S_i}\|_{\mathrm{op}}}{\varepsilon^2}$ | Evaluate $\nabla f(\boldsymbol{x})$ |
| Nesterov acceleration on smoothened objective | $\dfrac{\|\boldsymbol{x}^\star\|_2 \left( \max_{1 \le i \le m} \frac{1}{\sqrt{n_i}} \|\mathbf{A}_{S_i}\|_{\mathrm{op}} \right)^{1/2}}{\varepsilon}$ | Evaluate $\nabla \widetilde{f}_{\beta,\delta}(\boldsymbol{x})$ |
| Abernethy et al. (2022) | $\dfrac{\|\boldsymbol{x}^\star\|_2 \max_{1 \le i \le m} \frac{1}{\sqrt{n_i}} \|\mathbf{A}_{S_i}\|_{\mathrm{op}}}{\varepsilon}$ | Evaluate $\nabla \widetilde{f}_{\beta,\delta}(\boldsymbol{x})$ |
| Interior point with log barrier (Boyd & Vandenberghe, 2004) | $m^{1/2} \log \left( \frac{1}{\varepsilon} \right)$ | Linear-system-solve in $\mathbf{A}^\top \mathbf{B} \mathbf{A}$ |
| **This paper** (naïve geometry) | $\dfrac{m^{1/3}}{\varepsilon^{2/3}}$ | Linear-system-solve in $\mathbf{A}^\top \mathbf{B} \mathbf{A}$ |
| $\ell_\infty$ regression with Lewis weights (Jambulapati et al., 2022) | $\dfrac{\mathsf{rank}(\mathbf{A})^{1/3}}{\varepsilon^{2/3}}$ | Linear-system-solve in $\mathbf{A}^\top \mathbf{D} \mathbf{A}$ |
| $\ell_\infty$ regression with IPM (Lee & Sidford, 2019) | $\mathsf{rank}(\mathbf{A})^{1/2} \log \left( \frac{1}{\varepsilon} \right)$ | Linear-system-solve in $\mathbf{A}^\top \mathbf{D} \mathbf{A}$ |
| **This paper (Theorem 1)** | $\dfrac{\min\{\mathsf{rank}(\mathbf{A}), m\}^{1/3}}{\varepsilon^{2/3}}$ | Linear-system-solve in $\mathbf{A}^\top \mathbf{B} \mathbf{A}$ |

Table 1: The complexities of algorithms for optimizing equation 2 or for the special case of $\ell_\infty$ regression, assuming $\mathsf{OPT} = 1$ (the first three guarantees are additive approximations) and ignoring $\mathrm{polylog}(n, m)$ terms. We write $\mathbf{D}$ to be a diagonal matrix and $\mathbf{B}$ to be a block-diagonal matrix where each block has size $(n_i + o(1)) \times (n_i + o(1))$. We remark that in the special case where $n_i = 1$, our algorithm exactly recovers guarantees of Jambulapati et al. (2022). We stress that we include the references to $\ell_\infty$ regression only to show that our algorithm is no worse than that of Jambulapati et al. (2022) in this special case of $n_i = 1$ for all $i$, and none of their algorithms apply to our general setting.

Unlike the aforementioned first-order methods, our algorithm has no geometry-dependent terms. Additionally, our algorithm improves over the standard log-barrier IPM when the desired accuracy $\varepsilon \ge m^{-1/4}$ — this improvement is more pronounced when $m \gg \mathsf{rank}(\mathbf{A})$, i.e. when the number of data sources is much larger than the dimension of the parameter vector $\boldsymbol{x}$. Additionally, for $\varepsilon \ge \mathsf{rank}(\mathbf{A})^{-1/4}$, our guarantee matches the best known guarantee for $\ell_\infty$ regression (Lee & Sidford, 2019; Jambulapati et al., 2022).

**Remark 1.1** (Why use linear-system-solve complexity?). *We benchmark our algorithms using the number of linear-system-solves for a few reasons. First, this is typically how second-order algorithms are compared, such as interior point methods for linear programming (Lee & Sidford, 2019). Second, the particular structure of the linear-system-solves presents the possibility of a faster amortized runtime for the systems over the algorithm's run. This observation, combined with an understanding of how the linear systems changed between iterations, was used recently used to achieve fast runtimes for linear programming (Lee & Sidford, 2019) and $\ell_\infty$ regression (Adil et al., 2024).*

**Interpolating between robust and nonrobust optimization.** We also study the following family of objectives that interpolate between objectives 1 and 2 for different values of $p \ge 2$,

$$\min_{\boldsymbol{x} \in \mathbb{R}^d} \frac{1}{m} \sum_{i \in [m]} \left( \frac{1}{n_i} \|\mathbf{A}_{S_i} \boldsymbol{x} - \boldsymbol{b}_{S_i}\|_2^2 \right)^{p/2} . \tag{4}$$

In particular, note that choosing $p = 2$ in the above objective gives us the average least-squares problem in objective 1, while $p \to \infty$ recovers objective 2. Varying $p$ from 2 to $\infty$ and minimizing gives solutions that interpolate between utilitarian and egalitarian approaches, allowing for a smooth trade-off between utility and robustness. To this end, we give Algorithm 5 to approximately optimize objective 4 and prove the following guarantee about its iteration complexity.

**Theorem 2** (Trading off utility and robustness). *Let $\mathbf{A}_{S_i} \in \mathbb{R}^{n_i \times d}$ and $\boldsymbol{b}_{S_i} \in \mathbb{R}^{n_i}$ for all $i \in [m]$. Denote their concatenations by $\mathbf{A} := [\mathbf{A}_{S_1}^\top \ldots \mathbf{A}_{S_M}^\top]^\top \in \mathbb{R}^{n \times d}$ and $\boldsymbol{b} := [\boldsymbol{b}_{S_1}^\top \ldots \boldsymbol{b}_{S_M}^\top]^\top \in \mathbb{R}^n$ where $n := \sum_{i \in [m]} n_i$. Let $p \geq 2$ and $\varepsilon > 0$. Then Algorithm 5 returns $\widehat{\boldsymbol{x}}$ such that,*

$$\left( \sum_{i=1}^m \left( \frac{1}{\sqrt{n_i}} \| \mathbf{A}_{S_i} \widehat{\boldsymbol{x}} - \boldsymbol{b}_{S_i} \|_2 \right)^p \right)^{1/p} \leq (1+\varepsilon) \cdot \min_{\boldsymbol{x} \in \mathbb{R}^d} \left( \sum_{i=1}^m \left( \frac{1}{\sqrt{n_i}} \| \mathbf{A}_{S_i} \boldsymbol{x} - \boldsymbol{b}_{S_i} \|_2 \right)^p \right)^{1/p} \quad (5)$$

*and runs in*

$$O \left( p^{O(1)} \min \{ \mathsf{rank}\,(\mathbf{A})\,, m \}^{\frac{p-2}{3p-2}} \log \left( \frac{pd}{\varepsilon} \right)^3 \right)$$

*linear-system-solves in matrices of the form $\mathbf{A}^\top \mathbf{B} \mathbf{A}$, where $\mathbf{B}$ is a block-diagonal matrix for which block $i$ has size $n_i \times n_i$.*

We prove Theorem 2 in Appendix D.

In the special case where $n_i = 1$ for all $i$ (and therefore the problem is $\ell_p$ regression for $p \geq 2$), the complexity promised by Theorem 2 is comparable to that promised by Jambulapati et al. (2022) for $\ell_p$ regression. The main difference is that our iteration complexity is unconditionally polynomial in $p$. In contrast, the comparable result from Jambulapati et al. (2022) seems to require mild assumptions on the problem parameters (see the "Discussion on numerical stability" by Jambulapati et al. (2022, Section 4)).

**Remark 1.2** (Large values of $p$). *Note that for values of $p$ larger than $\log(m)$, solving equation 2 is almost equivalent to solving equation 4. To intuitively see this, first recall that for any vector $\boldsymbol{x} \in \mathbb{R}^d$ and $p = \log_2(m)$ we have, $\|\boldsymbol{x}\|_\infty \leq \|\boldsymbol{x}\|_p \leq 2 \cdot \|\boldsymbol{x}\|_\infty$. This implies that for all $i \in [m]$ we have the following for objective equation 4 (for $p = \log_2(m)$) for any $\boldsymbol{x} \in \mathbb{R}^d$,*

$$\max_{i \in [m]} \| \mathbf{A}_{S_i} \boldsymbol{x} - \boldsymbol{b}_{S_i} \|_2 \leq \left( \sum_{i \in [m]} \| \mathbf{A}_{S_i} \boldsymbol{x} - \boldsymbol{b}_{S_i} \|_2^p \right)^{1/p} \leq 2 \cdot \max_{i \in [m]} \| \mathbf{A}_{S_i} \boldsymbol{x} - \boldsymbol{b}_{S_i} \|_2 \ .$$

*In particular, this means that minimizing the interpolating objective equation 4 also minimizes the robust objective equation 2 (up to numerical constants) and vice versa. Thus, for $p = \Omega\left( \log_2(m) \right)$, for our intended applications, it makes sense to minimize the robust objective instead. This is why, in Theorem 2, we do not care too much about the exponent on $p$ in the iteration complexity. Our main goal is to show that we can get a $O(\mathsf{poly}(p, \log\left(\frac{1}{\varepsilon}\right)) \min \{ \mathsf{rank}\,(\mathbf{A})\,, m \}^{1/3})$ iteration complexity.*

## 1.2 PRIOR RESULTS, CONNECTIONS, AND OPEN PROBLEMS

Here, we discuss prior work that conceptually and technically relates to ours. We then suggest natural directions for future work.

**Multi-distribution learning.** Many learning problems involve multiple data sources, for instance, when multiple agents generate their data independently. One can formulate these multi-distribution problems as standard learning/optimization problems by considering a mixture of their distributions, as in objective 1. However, this approach often biases solutions toward dominant data sources, leading to poor performance on outliers—an issue stemming from statistical heterogeneity. This limitation motivates the study of multi-objective optimization problems (Miettinen, 1999; Ehrgott, 2005), where each agent $m$ has a distribution $\mathcal{D}_m$ that defines its objective as $\mathbb{E}_{z \sim \mathcal{D}_m}[f(\boldsymbol{x}_m; z)]$, and where models $\boldsymbol{x}_m$ can vary across agents—a framework known as personalization.

One of the earliest algorithms for such problems was introduced by Blum et al. (2017), where each agent's objective must be minimized to a pre-specified threshold $\epsilon$ with high probability, framed within a PAC learning framework (Valiant, 1984; Vapnik, 2013). Subsequent research has refined

these algorithms, achieving optimal sample complexity guarantees for learning from multiple distributions (Chen et al., 2018; Nguyen & Zakynthinou, 2018; Hanneke & Kpotufe, 2019; Haghtalab et al., 2022; Zhang et al., 2024b). Our objectives 2 and 4 offer different approaches to multi-distribution learning, where data distributions correspond to empirical agent distributions. In particular, Mohri et al. (2019) analyzed objective 2 to establish generalization bounds for unknown mixtures of agents' distributions.

Beyond sample efficiency, researchers have also examined other challenges, such as communication costs in large-scale distributed optimization (McMahan et al., 2016). A particularly relevant study is that of Bullins et al. (2021), which employs an efficient distributed quadratic sub-solver (Woodworth et al., 2020; Patel et al., 2024) to implement an inexact Newton method for optimizing quasi-self-concordant functions (see Definition 2.1).

**Group fairness.** Recently, interest in algorithmic fairness has intensified (Barocas & Selbst, 2016; Abebe et al., 2020; Kasy & Abebe, 2021) with researchers exploring fairness across various domains, including supervised learning (Calders et al., 2009; Dwork et al., 2012; Hardt et al., 2016; Kusner et al., 2017; Goel et al., 2018; Ustun et al., 2019), resource allocation (Bertsimas et al., 2011; 2012; Hooker & Williams, 2012; Donahue & Kleinberg, 2020; Manshadi et al., 2021), scheduling (Mulvany & Randhawa, 2021), online matching (Chierichetti et al., 2019; Ma et al., 2023), assortment planning (Singh & Joachims, 2018; Biega et al., 2018; Singh & Joachims, 2019; Chen et al., 2022), and facility location (Gupta et al., 2022). The extensive literature on algorithmic fairness falls into three main categories: (1) individual fairness, which ensures that similar individuals receive comparable predictions (Dwork et al., 2012; Loi et al., 2019; Chen et al., 2022), (2) group fairness, which aims for equal treatment of different demographic groups, often in terms of resource allocation or performance parity (Singh & Joachims, 2018; Balseiro et al., 2021), and (3) subgroup fairness, which blends aspects of both individual and group fairness (Kearns et al., 2018; 2019).

This paper focuses on a well-studied group fairness notion in machine learning literature: the group DRO problem (Ben-Tal et al., 2013; Duchi et al., 2016; Sagawa et al., 2019). The idea of interpolating between robustness and utility is also common (Golrezaei et al., 2024) and closely related to multi-objective optimization, where scalarization (Miettinen, 1999; Ehrgott, 2005) helps recover desired solutions along the Pareto frontier.

**Linear programming and $\ell_p$ regression.** In the last several years, there has been a surge of work in obtaining second-order, condition-free algorithms for linear programming and $\ell_p$ regression (Bubeck et al., 2018; Lee & Sidford, 2019; Adil et al., 2019; Jambulapati et al., 2022). Observe that $\ell_p$ regression is a special case of the problem we study in objective equation 4, which is recovered when all $n_i = 1$, and $\ell_\infty$ regression is captured by linear programming. Note that neither of these problem families is expressive enough to capture the objectives we study. In general, to achieve iteration complexities in the smaller of the two dimensions for these problems, it appears that a geometric understanding of the solution space is required — these ideas were central to the improvements obtained by Lee & Sidford (2019); Jambulapati et al. (2022) as well as our work.

**Open problems.** Our work raises several open questions. One limitation of Theorem 1 is that its iteration complexity is not high-accuracy, meaning its dependence on $\varepsilon$ is not $\mathrm{polylog}(1/\varepsilon)$. Designing a high-accuracy solver under the same conditions as Theorem 2 with iteration complexity $\widetilde{O}\left(\mathrm{poly}(\min\{\mathrm{rank}(\mathbf{A}), m\}, \log\left(\frac{1}{\varepsilon}\right))\right)$ remains an open problem.

A more ambitious general goal is to design algorithms for convex quadratic programs with the aforementioned iteration complexity. This would generalize analogous results for linear programming (Lee & Sidford, 2019). We view the current work as a first step towards this goal, as the objective equation 2 is a structured convex quadratic program for which we get an iteration complexity independent of $m$. It would also be interesting to consider other complexity measures beyond $\mathrm{rank}(\mathbf{A})$, for instance, assumptions about the ground-truth labeling vector $\boldsymbol{x}_i^\star$ for each group's data $S_i$.

Finally, our results suggest that optimizing for "$\ell_p$-interpolants" between non-robust and robust objectives may be computationally easier than optimizing for the robust objective alone. A more precise statistical characterization of how robustness and utility trade-off as $p$ varies in collaborative, fair, or multi-distributional learning settings would be valuable. Additionally, exploring interpolations or solution concepts along the Pareto frontier of the $m$-dimensional multi-objective optimization problem or other DRO notions (eg Wasserstein DRO (Blanchet et al., 2019; Cisneros-Velarde et al., 2020)) could yield further insights.

## 1.3 PAPER OUTLINE

In the remainder of this paper, we will outline the key details of our approach and provide a proof outline for our theoretical results. In Section 2, we give proof sketches of our main results. In Appendix A, we give an analysis of mirror descent under inexact subproblem solves – we will need this in the proof of Theorem 2. In Appendix B, we modify an acceleration scheme due to Carmon et al. (2022), which we will use to iterate calls to the proximal subproblem solver equation 8 for the proof of Theorem 2. In Appendix C, we prove Theorem 1. In Appendix D, we prove Theorem 2. In Appendix E, we prove some background results that appear in the main body, particularly about block Lewis weights. Finally, in Appendix F we include an empirical comparison of our proposed algorithms against the aforementioned baselines.

## 2 TECHNICAL OVERVIEW

In this section, we sketch our proofs for Theorem 1 and Theorem 2.

**Notation.** Here and in the rest of the paper, we ignore the dataset size normalization factors $1/\sqrt{n_i}$ as we can fold this into $\mathbf{A}_{S_i}$ and $\boldsymbol{b}_{S_i}$. Additionally, let $f(\boldsymbol{x}) := \sum_{i=1}^m \|\mathbf{A}_{S_i}\boldsymbol{x} - \boldsymbol{b}_{S_i}\|_2^p$ if $2 \leq p < \infty$ and let $f(\boldsymbol{x}) := \max_{1 \leq i \leq m} \|\mathbf{A}_{S_i}\boldsymbol{x} - \boldsymbol{b}_{S_i}\|_2$ if $p = \infty$. Note that in the $2 \leq p < \infty$ case, we let $f(\boldsymbol{x})$ be the $p$th power of the objective written in Theorem 2; this is to make future calculations easier and makes a difference of only polynomial factors in $p$ in the iteration complexity. Without loss of generality (by rescaling), let $\mathsf{OPT} = 1$, where $\mathsf{OPT} := f(\boldsymbol{x}^\star)$. So, it is enough to get an $\varepsilon$-additive optimal solution $\widehat{\boldsymbol{x}}$. Also without loss of generality, let $\mathbf{A}$ be such that $\mathrm{rank}(\mathbf{A}) = d$. For a positive semidefinite $\mathbf{M} \in \mathbb{R}^{d \times d}$, denote $\|\boldsymbol{x}\|_{\mathbf{M}} := \sqrt{\boldsymbol{x}^\top \mathbf{M} \boldsymbol{x}}$. As shorthand, for $\boldsymbol{y} \in \mathbb{R}^n$, we will often refer to the norm $\|\boldsymbol{y}\|_{\mathcal{G}_p} := \left(\sum_{i=1}^m \|\boldsymbol{y}_{S_i}\|_2^p\right)^{1/p}$ for $p \geq 1$, where with a slight abuse of notation $\boldsymbol{y}_{S_i}$ denotes the coordinates of $\boldsymbol{y}$ indexed by $S_i$. Finally, in an abuse of notation, for symmetric matrices $\mathbf{M}$, let $\mathbf{M}^{-1}$ denote the pseudoinverse of $\mathbf{M}$.

Recall that many iterative methods for convex optimization can be seen as decomposing a complex problem into a series of simpler subproblems (Nocedal & Wright, 2006). Our algorithms for distributionally robust linear regression follow this pattern, where the simple subproblem resembles

$$\mathcal{O}(\boldsymbol{q}) := \min_{\|\boldsymbol{x} - \boldsymbol{q}\|_{\mathbf{M}} \leq r_{\boldsymbol{q}}} f(\boldsymbol{x}) \ , \tag{6}$$

for some positive semidefinite $\mathbf{M}$ and for some ball radius $r_{\boldsymbol{q}}$ which may depend on the query $\boldsymbol{q}$. Sub-routines like equation 6 are central to many trust-region methods (Conn et al., 2000; Nocedal & Wright, 2006), and, importantly when $f$ is the sum of a linear function and a self-concordant barrier, interior point methods derived from the self-concordant barrier framework * (Nesterov & Nemirovskii, 1994).

With such a subproblem structure in hand, three questions arise. (1) How do we solve the subproblems efficiently? (2) How do we combine our subproblem solutions to arrive at our final answer? (3) How do we choose the "local geometry" $\mathbf{M}$ to optimize the iteration complexity we get from the previous two parts? We address these concerns in order in the following discussion.

## 2.1 SOLVING PROXIMAL SUBPROBLEMS

For this discussion, let $\mathbf{M}$ be any positive semidefinite matrix, as the arguments apply for any geometry $\mathbf{M}$. It will be helpful to assume that $\|\cdot\|_{\mathbf{M}}$ is a good approximation to our objective function in the sense that for some *distortion* $\triangle$ that is as close to 1 as possible, we have

$$\text{for all } \boldsymbol{x} \in \mathbb{R}^d : \qquad \|\boldsymbol{x} - \boldsymbol{b}\|_{\mathbf{M}} \leq \left(\sum_{i=1}^m \|\mathbf{A}_{S_i}\boldsymbol{x} - \boldsymbol{b}_{S_i}\|_2^p\right)^{\frac{1}{p}} \leq \triangle \|\boldsymbol{x} - \boldsymbol{b}\|_{\mathbf{M}} \ .$$

Here, we discuss how to solve problems of the form equation 6 for a fixed query $\boldsymbol{q}$. Our strategy follows two general steps. First, we establish some form of local stability for $\nabla^2 f(\boldsymbol{x})$ within the ball we are solving in, i.e., we want $\nabla^2 f(\boldsymbol{x})$ to not change too much inside the ball $\{\boldsymbol{x} \in \mathbb{R}^d : \|\boldsymbol{x} - \boldsymbol{q}\|_{\mathbf{M}} \leq r_{\boldsymbol{q}}\}$. Second, we use this to demonstrate that an appropriate second-order algorithm exhibits a favorable convergence rate to an approximate solution for our subproblem. We handle the $p = \infty$ and $2 \leq p < \infty$ cases separately below.

---

*In this case, the matrix $\mathbf{M}$ is given by the Hessian of the barrier function evaluated at the subproblem's solution.

### 2.1.1 THE ROBUST CASE ($p = \infty$).

Unfortunately, since $f$ is not even differentiable (it is the pointwise maximum of Euclidean norms, each of which is also not differentiable), we cannot directly argue about the stability of $\nabla^2 f(\boldsymbol{x})$. We therefore first need to find some surrogate objective $\widetilde{f}$ so that:

1. The approximation error $\left\| \widetilde{f} - f \right\|_\infty$ is small;

2. The surrogate objective $\widetilde{f}$ is smooth in $\|\cdot\|_{\mathbf{M}}$ in such a way that we can solve the proximal subproblems fast.

To smoothen $f(\boldsymbol{x})$, we use the family of objectives parameterized by $\beta, \delta$

$$\widetilde{f}_{\beta,\delta}(\boldsymbol{x}) := \beta \log \left( \sum_{i=1}^m \exp \left( \frac{\sqrt{\delta^2 + \|\mathbf{A}_{S_i}\boldsymbol{x} - \boldsymbol{b}_{S_i}\|_2^2} - \delta}{\beta} \right) \right) . \tag{7}$$

This can be seen as composing the softmax function with temperature $\beta$ with "inner functions" $\sqrt{\delta^2 + \|\mathbf{A}_{S_i}\boldsymbol{x} - \boldsymbol{b}_{S_i}\|_2^2} - \delta$. It is straightforward to show that for all $\boldsymbol{x} \in \mathbb{R}^d$, $\left| \widetilde{f}_{\beta,\delta}(\boldsymbol{x}) - f(\boldsymbol{x}) \right| \leq \beta \log m + \delta$. So, setting $\beta = \varepsilon/4 \log m$ and $\delta = \varepsilon/4$, it is sufficient to optimize $\widetilde{f}_{\beta,\delta}$ up to $\varepsilon/2$ additive error to get an $\varepsilon$-additive suboptimal solution to our original objective. Furthermore, we prove that $\widetilde{f}_{\beta,\delta}$ is $O(1/\beta + 1/\delta)$-smooth in the norm $\|\mathbf{A}\boldsymbol{x}\|_{\mathcal{G}_\infty} := \max_{1 \leq i \leq m} \|\mathbf{A}\boldsymbol{x}\|_2$. Thus, if $\|\cdot\|_{\mathbf{M}}$ is a good approximation to $\|\mathbf{A}\boldsymbol{x}\|_{\mathcal{G}_\infty}$, we will get that $\widetilde{f}_{\beta,\delta}$ is also smooth in the norm $\|\boldsymbol{x}\|_{\mathbf{M}}$.

Next, Carmon et al. (2020) show that if $\widetilde{f}_{\beta,\delta}$ satisfies a higher-order smoothness condition called *quasi-self-concordance* with respect to the norm $\|\cdot\|_{\mathbf{M}}$, then we can get the required Hessian stability for a *fixed* $r_{\boldsymbol{q}} = \Theta(1/\varepsilon)$ (in particular, $r_{\boldsymbol{q}}$ does not depend on $\boldsymbol{q}$ here). To clarify, we define quasi-self-concordance as follows.

**Definition 2.1** (Quasi-self-concordance, adapted from (Karimireddy et al., 2018, Appendix A)). *Let $f \colon \mathbb{R}^k \to \mathbb{R}$. We say that $f$ is $\nu$-quasi-self-concordant in the norm $\|\cdot\|$ if for all vectors $\boldsymbol{y} \in \mathbb{R}^k$, directions $\boldsymbol{d} \in \mathbb{R}^k$, and $t \in \mathbb{R}$, we have*

$$\left| \left( \frac{d}{dt} \right)^3 f(\boldsymbol{y} + t\boldsymbol{d}) \right| \leq \nu \|\boldsymbol{d}\| \left( \frac{d}{dt} \right)^2 f(\boldsymbol{y} + t\boldsymbol{d}) .$$

Then, Carmon et al. (2020) shows how to leverage this Hessian stability to implement equation 6 with low linear-system-solve iteration complexity. However, previously, it was only shown that the composition of the softmax function with linear functions is quasi-self-concordant. So, it was unknown whether composing softmax with other functions could also be quasi-self-concordant.

To resolve this, we prove a much more general composition result, which to the best of our knowledge was not known prior to this work and may be of independent interest. It essentially states that if we compose the softmax function with any combination of "inner" functions that are quasi-self-concordant, the resulting function is also quasi-self-concordant. For a more formal statement, see Lemma C.3.

**Lemma C.3** (Composing softmax with quasi-self-concordant functions). *Let $\|\cdot\|$ be an arbitrary norm and $h_1, \ldots, h_m$ be such that $h_i \colon \mathbb{R}^d \to \mathbb{R}$. Let $h$ be the vector formed by concatenating the results of $h_1, \ldots, h_m$. Additionally, let $h_1, \ldots, h_m$ be such that for all $1 \leq i \leq m$ and for all $\boldsymbol{y}, \boldsymbol{d} \in \mathbb{R}^m$ and $t \in \mathbb{R}$,*

$$\left( \frac{d}{dt} \right) h_i(\boldsymbol{y} + t\boldsymbol{d}) \leq \|\boldsymbol{d}\| \qquad\qquad (Lipschitzness)$$

$$\left| \left( \frac{d}{dt} \right)^3 h_i(\boldsymbol{y} + t\boldsymbol{d}) \right| \leq \nu \|\boldsymbol{d}\| \left( \frac{d}{dt} \right)^2 h_i(\boldsymbol{y} + t\boldsymbol{d}) \qquad (quasi\text{-}self\text{-}concordance).$$

*Then, for all $\boldsymbol{y}, \boldsymbol{d} \in \mathbb{R}^m$ and all $t \in \mathbb{R}$, we have*

$$\left| \left( \frac{d}{dt} \right)^3 \beta \log \left( \sum_{i=1}^m \exp \left( \frac{h_i(\boldsymbol{y} + t\boldsymbol{d})}{\beta} \right) \right) \right| \leq \left( \frac{16}{\beta} + \nu \right) \|\boldsymbol{d}\| \left( \frac{d}{dt} \right)^2 \mathsf{lse}_\beta(h(\boldsymbol{y} + t\boldsymbol{d})).$$

Hence, to show the requisite Hessian stability, we use the following steps. We show that the "inner" functions for equation 7, $\sqrt{\delta^2 + \|\mathbf{A}_{S_i}\boldsymbol{x} - \boldsymbol{b}_{S_i}\|_2^2} - \delta$, are each $O(1/\delta)$-quasi-self-concordant in the norm $\|\mathbf{A}_{S_i}\boldsymbol{x}\|_2$. So, we can apply our composition result Lemma C.3 to prove that $\widetilde{f}_{\beta,\delta}$ is $O(1/\beta + 1/\delta)$-quasi-self-concordant in the norm $\max_{i\in[m]} \|\mathbf{A}_{S_i}\boldsymbol{x}\|_2$. Again, assuming that $\|\cdot\|_{\mathbf{M}}$ is a good approximation to $\|\cdot\|_{\mathcal{G}_\infty}$, we will get that $\widetilde{f}_{\beta,\delta}$ is quasi-self-concordant in $\|\boldsymbol{x}\|_{\mathbf{M}}$ as well.

With these analytic inequalities in hand, we can finally apply the recipe given in Carmon et al. (2020) and get our subproblem solver for the $p = \infty$ case.

### 2.1.2 THE INTERPOLATING CASE ($2 \leq p < \infty$).

Instead of explicitly constraining $r_{\boldsymbol{q}}$ like in the $p = \infty$ case, we regularize our movement from $\boldsymbol{q}$ in the norm $\|\cdot\|_{\mathbf{M}}$. Specifically, the subproblem we solve for any query $\boldsymbol{q}$ is

$$\operatorname*{argmin}_{\boldsymbol{x}\in\mathbb{R}^d} f(\boldsymbol{x}) + ep^p \|\boldsymbol{x} - \boldsymbol{q}\|_{\mathbf{M}}^p \quad . \tag{8}$$

This is the natural generalization of the proximal problem that Jambulapati et al. (2022) use to get their results for $\ell_p$ regression, and the outline of our solver for these subproblems is similar to what Jambulapati et al. (2022) use for this special case (see their Section 4).

However, we go a step further and show how to obtain approximate stationary points to equation 8 instead of just getting a small objective value. This is because the acceleration scheme we use to iterate subproblem solutions to get our final answer $\widehat{\boldsymbol{x}}$ requires us to obtain an approximate stationary point for equation 8. The main new technical tool we develop for this purpose is a form of strong convexity for functions of the form $\|\boldsymbol{y}\|_2^p$ for $\boldsymbol{y} \in \mathbb{R}^k$ for any $k \geq 1$. See Lemma D.3.

**Lemma D.3** (Strong convexity of $\|\boldsymbol{y}\|_2^p$). *Let $\boldsymbol{v} \in \mathbb{R}^k$ for $k \geq 1$. For any $\triangle \in \mathbb{R}^k$, we have*

$$\|\boldsymbol{v} + \triangle\|_2^p \geq \|\boldsymbol{v}\|_2^p + p \|\boldsymbol{v}\|_2^{p-2} \langle \boldsymbol{v}, \triangle \rangle + \frac{4}{2^p} \|\triangle\|_2^p \quad .$$

With Lemma D.3, we can argue about the strong convexity of $\|\boldsymbol{x} - \boldsymbol{q}\|_{\mathbf{M}}^p$, which means that we can convert an approximately optimal solution to equation 8 in function value to one that is approximately optimal in parameter space as well. We combine this with a local gradient Lipschitzness property of the objective equation 8 to get our approximate stationary point, which is enough for our purposes. The local gradient Lipschitzness property itself follows from a form of Hessian stability that we show for the objective equation 8. See Lemma D.9.

Finally, to obtain an approximately optimal solution to equation 8 in function value, we again apply the Hessian stability property to conclude that equation 8 is relatively smooth and relatively strongly convex in a simpler reference function. We show how to solve optimization problems in this reference function up to an approximate optimality that is sufficient for the rest of our applications – this requires a mild modification of the standard mirror descent analysis, and we do this in Appendix A. Combining all of these building blocks gives us our subproblem solver for the $2 \leq p < \infty$ case.

### 2.2 ITERATING PROXIMAL CALLS

We now discuss the second item. Recall that we think of $\mathcal{O}(\boldsymbol{q})$ as answering a proximal problem for the query $\boldsymbol{q}$. It is not hard to show that under reasonable conditions on $f$ and on the structure of the subproblems, we can iterate calls to $\mathcal{O}(\boldsymbol{q})$ to optimize $f$ (see, e.g., (Carmon et al., 2020, Appendix A)). This conceptually simple approach will already give us guarantees of the form $\|\boldsymbol{x}_0 - \boldsymbol{x}^\star\|_{\mathbf{M}}/\varepsilon$ for the problems we study.

But we can do better. An acceleration framework originally due to Monteiro & Svaiter (2013) and generalized/refined in subsequent works (Bubeck et al., 2019; Carmon et al., 2020; 2022) gives a recipe to iterate calls of $\mathcal{O}(\boldsymbol{q})$ to optimize the original function $f$. From these, the iteration complexity we need for an $\varepsilon$-additive solution with an initialization $\boldsymbol{x}_0$ and optimum $\boldsymbol{x}^\star$ is roughly $(\|\boldsymbol{x}_0 - \boldsymbol{x}^\star\|_{\mathbf{M}}/\varepsilon)^{2/3}$ (see Theorem B.3 for a more formal statement). This cosmetically resembles the rate we get in Theorem 1. To get something that looks like our rate for Theorem 2, we use our new strong convexity lemma (Lemma D.3). With this, we can demonstrate that after a sufficient number of iterations, we have $\|\boldsymbol{x}_t - \boldsymbol{x}^\star\|_{\mathbf{M}} \leq 0.5 \|\boldsymbol{x}_0 - \boldsymbol{x}^\star\|_{\mathbf{M}}$. Therefore, repeating this argument yields a high-accuracy solution, as required.

Interestingly, our algorithm for the $2 \leq p < \infty$ case employs a form of the accelerated scheme developed in Carmon et al. (2022), which does not require solving an implicit equation for the query point, thereby improving upon the results from Jambulapati et al. (2022) for $\ell_p$ regression. It would be practically relevant to obtain this for the $p = \infty$ case (in Appendix B, we discuss a technical challenge in obtaining this).

## 2.3 The Geometry of the Proximal Subproblems and Block Lewis Weights

At this point, we have the tools we need to get rates of the form $O\left(\left(\|\boldsymbol{x}_0 - \boldsymbol{x}^\star\|_{\mathbf{M}}/\varepsilon\right)^{2/3}\right)$ for the robust objective (Theorem 1) and of the form $O\left(\|\boldsymbol{x}_0 - \boldsymbol{x}^\star\|_{\mathbf{M}}^{(p-2)/(3p-2)}\right)$ for the interpolating objective (Theorem 2). From this, we see that the rates depend on the geometry $\mathbf{M}$ that we impose on our problem. Our goal in this section is to choose this geometry $\mathbf{M}$.

Observe that when we solve equation 6, we are solving an optimization problem over the sublevel sets $\{\boldsymbol{x} \ : \ \|\boldsymbol{x}\|_{\mathbf{M}} \leq r_{\boldsymbol{q}}\}$ – these are ellipsoids. Now, consider choosing the $\ell_2$ geometry that best approximates our loss function. Specifically, recall that earlier in the section, we stated that for some distortion $\triangle \geq 1$ that is as close to 1 as possible, we want

$$\text{for all } \boldsymbol{x} \in \mathbb{R}^d : \qquad \|\boldsymbol{x} - \boldsymbol{b}\|_{\mathbf{M}} \leq \left(\sum_{i=1}^m \|\mathbf{A}_{S_i}\boldsymbol{x} - \boldsymbol{b}_{S_i}\|_2^p\right)^{\frac{1}{p}} \leq \triangle \|\boldsymbol{x} - \boldsymbol{b}\|_{\mathbf{M}} \ .$$

To see what kinds of distortion guarantees we can hope for, let us see what happens when we choose the most "obvious" geometry. By relating $\ell_2^m$ to $\ell_p^m$, we get

$$\left(\sum_{i=1}^m \|\mathbf{A}_{S_i}\boldsymbol{x} - \boldsymbol{b}_{S_i}\|_2^2\right)^{\frac{1}{2}} \leq \left(\sum_{i=1}^m \|\mathbf{A}_{S_i}\boldsymbol{x} - \boldsymbol{b}_{S_i}\|_2^p\right)^{\frac{1}{p}} \leq m^{\frac{1}{2} - \frac{1}{p}} \left(\sum_{i=1}^m \|\mathbf{A}_{S_i}\boldsymbol{x} - \boldsymbol{b}_{S_i}\|_2^2\right)^{\frac{1}{2}},$$

and notice that $\left(\sum_{i=1}^m \|\mathbf{A}_{S_i}\boldsymbol{x} - \boldsymbol{b}_{S_i}\|_2^2\right)^2 = \|\mathbf{A}\boldsymbol{x} - \boldsymbol{b}\|_2$. Thus, setting $\mathbf{M} = \mathbf{A}^\top \mathbf{A}$ (which is what we call the naïve geometry in Table 1) gives us our basic rate of $m^{1/3}\varepsilon^{-2/3}$ in the setting of Theorem 1 and $m^{(p-2)/(3p-2)}$ in the setting of Theorem 2.

But, there exists an improvement over above naïve geometry. Note our loss function is a norm on $\mathbb{R}^d$ – in particular, we can check that for $\boldsymbol{y} \in \mathbb{R}^n$, the functions $\|\boldsymbol{y}\|_{\mathcal{G}_p} = \left(\sum_{i=1}^m \|\boldsymbol{y}_{S_i}\|_2^p\right)^{1/p}$ for $1 \leq p \leq \infty$ are norms. Now, recall John's theorem, a fundamental result in high-dimensional convex geometry.

**Theorem 2.2** (John's theorem, John (1948)). *For any symmetric convex body $K \subset \mathbb{R}^d$, let $\mathcal{E}(K)$ denote the ellipsoid of maximum volume contained within $K$. Then, we have*

$$\mathcal{E}(K) \subseteq K \subseteq \sqrt{d} \cdot \mathcal{E}(K) \ .$$

*Moreover, the $\sqrt{d}$ is worst-case optimal (e.g. let $K$ be the unit $\ell_\infty$ ball).*

It is easy to see that sublevel sets of norms, i.e., sets of the form $\{\boldsymbol{x} \in \mathbb{R}^d \ : \ \|\boldsymbol{x}\| \leq 1\}$, are symmetric convex bodies. Hence, using John's theorem, we see that for our normed losses, there exists $\mathbf{M}$ that achieves distortion $\triangle \leq \sqrt{d}$. From this, it is easy to see that there exists $\mathbf{M}$ for which we can guarantee $\|\boldsymbol{x}_0 - \boldsymbol{x}^\star\|_{\mathbf{M}} \lesssim \sqrt{d}$. Plugging this into the guarantees from the previous subsections, we get that if we choose the $\mathbf{M}$ from John's theorem, and then switch based on whether $m \leq d$, we get exactly the rates quoted in Theorem 1 and Theorem 2.

However, as written, this is only an existence result. To make this useful for us and actually find $\mathbf{M}$, we need an algorithm to calculate John's ellipsoid for the level sets of our losses (or some other ellipsoid that gets an even better approximation factor). To this end, a result of Manoj & Ovsiankin (2025) gives us an efficient algorithm to find this $\ell_2$ geometry for the loss families we consider.

**Theorem 2.3** (Combining Lemmas 5.6, 5.8, Equation (1.8) from Manoj & Ovsiankin (2025)). *Let $p \geq 2$. There exists an algorithm that finds a positive diagonal matrix $\mathbf{W} \in \mathbb{R}^{n \times n}$ such that for all $\boldsymbol{x} \in \mathbb{R}^d$ and all $c \in \mathbb{R}$, we have*

$$\frac{\left\|\mathbf{W}^{\frac{1}{2} - \frac{1}{p}}(\mathbf{A}\boldsymbol{x} - c\boldsymbol{b})\right\|_2}{(2(\operatorname{rank}(\mathbf{A}) + 1))^{\frac{1}{2} - \frac{1}{p}}} \leq \left(\sum_{i=1}^m \|\mathbf{A}_{S_i}\boldsymbol{x} - c\boldsymbol{b}_{S_i}\|_2^p\right)^{\frac{1}{p}} \leq \left\|\mathbf{W}^{\frac{1}{2} - \frac{1}{p}}(\mathbf{A}\boldsymbol{x} - c\boldsymbol{b})\right\|_2 \ .$$

*The algorithm runs in $O(\log m)$ linear-system-solves in matrices of the form $\mathbf{A}^\top \mathbf{D} \mathbf{A}$ for positive diagonal matrices $\mathbf{D}$.*

The diagonal entries of matrix $\mathbf{W}$ are called *block Lewis weights*. This is a generalization of Lewis weights, and both objects have been used previously for various matrix approximation problems (Bourgain et al., 1989; Musco et al., 2022; Jambulapati et al., 2023b;a; Manoj & Ovsiankin, 2025). Furthermore, Lewis weights are central to improvements in the iteration complexities for linear programming and vanilla $\ell_p$ regression (Lee & Sidford, 2019; Jambulapati et al., 2022). We go into greater detail about block Lewis weights in Appendix E.

Additionally, notice that the distortion of $O(\mathsf{rank}\,(\mathbf{A})^{1/2-1/p})$ guaranteed by Theorem 2.3 is optimal. To see this, let $\mathbf{A} \in \mathbb{R}^{n \times d}$ be such that for $i \in [d]$, row $\boldsymbol{a}_i = \boldsymbol{e}_i$, where $\boldsymbol{e}_i$ is the $i$th standard basis vector. Then, for all $d+1 \le i \le n$, let $\boldsymbol{a}_i = 0$. In words, $\mathbf{A}$ is the $d$-dimensional identity matrix atop a large matrix of all 0s. It is easy to see that for any $p$, we have $\|\mathbf{A}\boldsymbol{x}\|_p = \|\boldsymbol{x}\|_p$, and the best distortion we can get for relating $\|\boldsymbol{x}\|_p$ to any $d$-dimensional $\ell_2$ norm is $d^{|1/2-1/p|}$.

With Theorem 2.3 and its near optimality in hand, we choose $\mathbf{M} = \mathbf{A}^\top \mathbf{W}^{1-\frac{2}{p}} \mathbf{A}$ if $\mathsf{rank}\,(\mathbf{A}) \le m$ and $\mathbf{M} = \mathbf{A}^\top \mathbf{A}$ if $\mathsf{rank}\,(\mathbf{A}) \ge m$ (recall that in the latter case, we get a $\sqrt{m}$ distortion for free from relating $\ell_2^m$ to $\ell_\infty^m$). Combining this with the results from the previous two subsections gives us Theorem 1 and Theorem 2.

## 2.4 ALGORITHM FOR DISTRIBUTIONALLY ROBUST REGRESSION

In Algorithm 1, we present pseudocode for the algorithm that yields the guarantee in Theorem 1. We compare the empirical performance of this algorithm against other baselines mentioned in Section 1 and examine the effect of Lewis weights in Appendix F.

---

**Algorithm 1** MinMaxRegression: optimizes equation 2 to $(1+\varepsilon)$-multiplicative error

---

**Require:** Regression problems $(\mathbf{A}_{S_1}, \boldsymbol{b}_{S_1}), \ldots, (\mathbf{A}_{S_m}, \boldsymbol{b}_{S_m})$, accuracy $\varepsilon > 0$
1: Using (Manoj & Ovsiankin, 2025, Algorithm 2) with input $[\mathbf{A}|\boldsymbol{b}]$, find nonnegative diagonal $\mathbf{W}$ and weights $w_1, \ldots, w_m$ such that for all $j \in S_i$, $\mathbf{W}[j][j] = w_i$ and for all $\boldsymbol{x} \in \mathbb{R}^d$ and $c \in \mathbb{R}$,

$$\|\mathbf{A}\boldsymbol{x} - c\boldsymbol{b}\|_{\mathcal{G}_\infty} \le \left\|\mathbf{W}^{1/2}\mathbf{A}\boldsymbol{x} - c\mathbf{W}^{1/2}\boldsymbol{b}\right\|_2 \le \sqrt{2(\mathsf{rank}\,(\mathbf{A})+1)}\,\|\mathbf{A}\boldsymbol{x} - c\boldsymbol{b}\|_{\mathcal{G}_\infty}.$$

2: **if** $\sum_{i=1}^m w_i \ge m$ **then**                    $\triangleright$ $\mathsf{rank}\,(\mathbf{A}) + 1 \le \sum_{i=1}^m w_i \le 2(\mathsf{rank}\,(\mathbf{A})+1)$
3: $\quad$ Reset $\mathbf{W} = \mathbf{I}_n$.
4: Let $\boldsymbol{x}_0 = \left(\mathbf{A}^\top \mathbf{W} \mathbf{A}\right)^{-1} \mathbf{A}^\top \mathbf{W} \boldsymbol{b}$.           $\triangleright$ $\boldsymbol{x}_0 \coloneqq \underset{\boldsymbol{x} \in \mathbb{R}^d}{\mathrm{argmin}} \left\|\mathbf{W}^{1/2}\mathbf{A}\boldsymbol{x} - \mathbf{W}^{1/2}\boldsymbol{b}\right\|_2.$
5: Let

$$\widetilde{f}_{\beta,\delta}(\boldsymbol{x}) \coloneqq \beta \log \left( \sum_{i=1}^m \exp \left( \frac{\sqrt{\delta^2 + \|\mathbf{A}_{S_i}\boldsymbol{x} - \boldsymbol{b}_{S_i}\|_2^2} - \delta}{\beta} \right) \right)$$

$\quad$ where $\beta = \frac{\varepsilon}{4 \log m}$ and $\delta = \frac{\varepsilon}{4}$.              $\triangleright$ *A family of smoothenings of the objective.*
6: Let $\widehat{f}(\boldsymbol{x}) \coloneqq \widetilde{f}_{\varepsilon/4\log m, \varepsilon/4}(\boldsymbol{x}) + \frac{\varepsilon}{1000 \min\{\mathsf{rank}(\mathbf{A}), m\}} \left\|\mathbf{W}^{1/2}\mathbf{A}(\boldsymbol{x} - \boldsymbol{x}_0)\right\|_2^2$.
7: Using (Carmon et al., 2020, Algorithm 3), implement a $\left( \frac{C}{\min\{\mathsf{rank}(\mathbf{A}), m\}}, \frac{C}{\varepsilon} \right)$-ball optimization
$\quad$ oracle for $\widehat{f}$, where $C$ is a universal constant. $\triangleright$ *Iteration complexity guaranteed by Lemma C.5*
8: Using (Carmon et al., 2020, Algorithm 2), implement a $\frac{1}{2}$-MS oracle for $\widehat{f}$.
9: Run (Carmon et al., 2020, Algorithm 1) for $\widetilde{O}\left( \frac{\min\{\mathsf{rank}(\mathbf{A}), m\}^{1/3} \log\left(\frac{d}{\varepsilon}\right)}{\varepsilon^{2/3}} \right)$ iterations using the
$\quad$ MS oracle from the previous line and with initial point $\boldsymbol{x}_0$ and final point $\widehat{\boldsymbol{x}}$.
10: **return** $\widehat{\boldsymbol{x}}$

---

ACKNOWLEDGMENTS

We thank Aaron Sidford for useful comments during the early stage of the project. We also thank the anonymous reviewers at ICLR'26 and NeurIPS'25 for their significant contributions to improving the paper. Most of this work was done when NS and KKP were graduate students at Toyota Technological Institute, Chicago (TTIC). KKP and NS were supported through the NSF TRIPOD Institute on Data, Economics, Algorithms and Learning (IDEAL) and other awards from DARPA and NSF.

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

## A  MIRROR DESCENT WITH INEXACT UPDATES

**Notation warning.**  This section is meant to be a self-contained, standalone analysis of mirror descent under inexact updates. The notation is chosen to be consistent with most material we could find on mirror descent and therefore conflicts with the notation used in the rest of the paper.

In this section, we give an analysis of unconstrained mirror descent when each Bregman proximal problem is solved only approximately (Algorithm 2). Although we expect that this is a standard fact about mirror descent, we could not find an appropriate reference. Hence, we produce it here.

---

**Algorithm 2** ApproximateMirrorDescent: Implements mirror descent to optimize convex and differentiable $f$ given $L$-relative smoothness and $\mu$-relative strong convexity in the reference $h$ when we may not be able to solve each proximal problem exactly.

---

**Require:** Initial point $\boldsymbol{x}_0$, iteration count $T$.
1: Define
$$D_h(\boldsymbol{x}, \boldsymbol{y}) := h(\boldsymbol{x}) - h(\boldsymbol{y}) - \langle \nabla h(\boldsymbol{y}), \boldsymbol{x} - \boldsymbol{y} \rangle$$
$$\boldsymbol{x}^\star := \operatorname*{argmin}_{\boldsymbol{x} \in \mathbb{R}^d} f(\boldsymbol{x}) \qquad .$$
2: **for** $i = 1, \ldots, T$ **do**
3:     $\boldsymbol{x}_i^\star = \operatorname*{argmin}_{\widetilde{\boldsymbol{x}} \in \mathbb{R}^d} f(\boldsymbol{x}_{i-1}) + \langle \nabla f(\boldsymbol{x}_{i-1}), \widetilde{\boldsymbol{x}} - \boldsymbol{x}_{i-1} \rangle + L D_h(\widetilde{\boldsymbol{x}}, \boldsymbol{x}_{i-1})$ ▷ *We may only be able*
    *to approximate $\boldsymbol{x}_i^\star$ – see the next line.*
4:     Let $\boldsymbol{x}_i$ be an approximate stationary point for the above objective.
  **return** $\operatorname*{argmin}_{0 \leq i \leq T} f(\boldsymbol{x}_i)$

---

In Algorithm 2, we assume that the function $f$ is $\mu$-relatively strongly convex and $L$-smooth in a *reference function* $h$. This means that for all $\boldsymbol{x}, \boldsymbol{y} \in \mathbb{R}^d$, we have

$$\mu D_h(\boldsymbol{x}, \boldsymbol{y}) \leq f(\boldsymbol{x}) - f(\boldsymbol{y}) - \langle \nabla f(\boldsymbol{y}), \boldsymbol{x} - \boldsymbol{y} \rangle \leq L D_h(\boldsymbol{x}, \boldsymbol{y}).$$

Using (Lu et al., 2018, Proposition 1.1), when $f$ is twice-differentiable, this condition is equivalent to asking for all $\boldsymbol{x} \in \mathbb{R}^d$,

$$\mu \nabla^2 h(\boldsymbol{x}) \preceq \nabla^2 f(\boldsymbol{x}) \preceq L \nabla^2 h(\boldsymbol{x}).$$

We are now ready to state the performance guarantee of Algorithm 2. See Theorem A.1.

**Theorem A.1.** *Let index $j$ be the index output by Algorithm 2. Let $\triangle_i$ be defined such that*

$$\triangle_i := \nabla f(\boldsymbol{x}_{i-1}) + L\left(\nabla h(\boldsymbol{x}_i) - \nabla h(\boldsymbol{x}_{i-1})\right).$$

*Then, we have*

$$f(\boldsymbol{x}_j) - f(\boldsymbol{x}^\star) \leq L\left(1 - \frac{\mu}{L}\right)^T D_h(\boldsymbol{x}^\star, \boldsymbol{x}_0) + \max_{1 \leq i \leq n} \langle \triangle_i, \boldsymbol{x}_i - \boldsymbol{x}^\star \rangle.$$

To prove Theorem A.1, we begin with a few standard facts about the mirror descent iterations.

**Lemma A.2.** *Let $\boldsymbol{y} \in \mathbb{R}^d$ be arbitrary. We have*

$$\langle \nabla f(\boldsymbol{x}_{i-1}), \boldsymbol{x}_i - \boldsymbol{y} \rangle = L\left(D_h(\boldsymbol{y}, \boldsymbol{x}_{i-1}) - D_h(\boldsymbol{y}, \boldsymbol{x}_i) - D_h(\boldsymbol{x}_i, \boldsymbol{x}_{i-1})\right) + \langle \triangle_i, \boldsymbol{x}_i - \boldsymbol{y} \rangle.$$

*Proof of Lemma A.2.*  By the three point identity (see, e.g., (Sra et al., 2016, Equation (A.9))), we have

$$D_h(\boldsymbol{y}, \boldsymbol{x}_{i-1}) - D_h(\boldsymbol{y}, \boldsymbol{x}_i) - D_h(\boldsymbol{x}_i, \boldsymbol{x}_{i-1}) = -\langle \nabla h(\boldsymbol{x}_i) - \nabla h(\boldsymbol{x}_{i-1}), \boldsymbol{x}_i - \boldsymbol{y} \rangle$$
$$= \frac{1}{L}\langle \nabla f(\boldsymbol{x}_{i-1}) - \triangle_i, \boldsymbol{x}_i - \boldsymbol{y} \rangle,$$

completing the proof of Lemma A.2. $\qquad\square$

**Lemma A.3** (Mirror descent lemma under approximate stationary point updates)**.** *Let $\boldsymbol{y} \in \mathbb{R}^d$ be arbitrary. For every iteration $i$, we have*

$$f(\boldsymbol{x}_i) - f(\boldsymbol{y}) \leq (L - \mu)D_h(\boldsymbol{y}, \boldsymbol{x}_{i-1}) - L D_h(\boldsymbol{y}, \boldsymbol{x}_i) + \langle \triangle_i, \boldsymbol{x}_i - \boldsymbol{y} \rangle.$$

*Proof of Lemma A.3.* The definition of $\mu$-relative strong convexity tells us that

$$f(\boldsymbol{x}_{i-1}) - f(\boldsymbol{y}) \le \langle \nabla f(\boldsymbol{x}_{i-1}), \boldsymbol{x}_{i-1} - \boldsymbol{y} \rangle - \mu D_h(\boldsymbol{y}, \boldsymbol{x}_{i-1}).$$

We now write

$$
\begin{aligned}
f(\boldsymbol{x}_i) - f(\boldsymbol{y}) &\le f(\boldsymbol{x}_{i-1}) - f(\boldsymbol{y}) + \langle \nabla f(\boldsymbol{x}_{i-1}), \boldsymbol{x}_i - \boldsymbol{x}_{i-1} \rangle + L D_h(\boldsymbol{x}_i, \boldsymbol{x}_{i-1}) && \text{($L$-RS)} \\
&\le \langle \nabla f(\boldsymbol{x}_{i-1}), \boldsymbol{x}_i - \boldsymbol{y} \rangle - \mu D_h(\boldsymbol{y}, \boldsymbol{x}_{i-1}) + L D_h(\boldsymbol{x}_i, \boldsymbol{x}_{i-1}) && \text{($\mu$-RSC)} \\
&\le (L - \mu) D_h(\boldsymbol{y}, \boldsymbol{x}_{i-1}) - L D_h(\boldsymbol{y}, \boldsymbol{x}_i) + \langle \triangle_i, \boldsymbol{x}_i - \boldsymbol{y} \rangle, && \text{(Lemma A.2)}
\end{aligned}
$$

completing the proof of Lemma A.3. $\qquad\square$

We now have the tools to complete the proof of Theorem A.1.

*Proof of Theorem A.1.* Let $E_i := f(\boldsymbol{x}_i) - f(\boldsymbol{x}^\star) - \langle \triangle_i, \boldsymbol{x}_i - \boldsymbol{x}^\star \rangle$. Substituting $\boldsymbol{y} = \boldsymbol{x}^\star$ and rearranging the conclusion of Lemma A.3 gives

$$E_i \le (L - \mu) D_h(\boldsymbol{x}^\star, \boldsymbol{x}_{i-1}) - L D_h(\boldsymbol{x}^\star, \boldsymbol{x}_i).$$

We multiply both sides by $\left(\frac{L}{L-\mu}\right)^i$ and write

$$\left(\frac{L}{L-\mu}\right)^i E_i \le \frac{L^i}{(L-\mu)^{i-1}} D_h(\boldsymbol{x}^\star, \boldsymbol{x}_{i-1}) - \frac{L^{i+1}}{(L-\mu)^i} D_h(\boldsymbol{x}^\star, \boldsymbol{x}_i).$$

Adding over all $T$ iterations yields

$$\sum_{i=1}^T \left(\frac{L}{L-\mu}\right)^i E_i \le L D_h(\boldsymbol{x}^\star, \boldsymbol{x}_0) - \left(\frac{L}{L-\mu}\right)^T L D_h(\boldsymbol{x}^\star, \boldsymbol{x}_T) \le L D_h(\boldsymbol{x}^\star, \boldsymbol{x}_0).$$

Expanding out the definition of $E_i$ and rearranging gives

$$\sum_{i=1}^T \left(\frac{L}{L-\mu}\right)^i (f(\boldsymbol{x}_i) - f(\boldsymbol{x}^\star)) \le L D_h(\boldsymbol{x}^\star, \boldsymbol{x}_0) + \sum_{i=1}^T \left(\frac{L}{L-\mu}\right)^i \langle \triangle_i, \boldsymbol{x}_i - \boldsymbol{x}^\star \rangle.$$

By the geometric series summation formula, we define and have

$$C_T := \sum_{i=1}^T \left(\frac{L}{L-\mu}\right)^i = \frac{L}{\mu}\left(\left(1 + \frac{\mu}{L-\mu}\right)^T - 1\right).$$

Let $j$ be the index that Algorithm 2 returns. It is easy to check that

$$\sum_{i=1}^T \left(\frac{L}{L-\mu}\right)^i (f(\boldsymbol{x}_i) - f(\boldsymbol{x}^\star)) \ge C_T (f(\boldsymbol{x}_j) - f(\boldsymbol{x}^\star))$$

and

$$\sum_{i=1}^T \left(\frac{L}{L-\mu}\right)^i \langle \triangle_i, \boldsymbol{x}_i - \boldsymbol{x}^\star \rangle \le C_T \max_{1 \le i \le n} \langle \triangle_i, \boldsymbol{x}_i - \boldsymbol{x}^\star \rangle.$$

This gives us

$$f(\boldsymbol{x}_j) - f(\boldsymbol{x}^\star) \le \frac{L}{C_T} D_h(\boldsymbol{x}^\star, \boldsymbol{x}_0) + \max_{1 \le i \le n} \langle \triangle_i, \boldsymbol{x}_i - \boldsymbol{x}^\star \rangle.$$

Finally, notice that

$$\frac{L}{C_T} = \frac{\mu}{\left(1 + \frac{\mu}{L-\mu}\right)^T - 1} \le L\left(1 - \frac{\mu}{L}\right)^T.$$

Combining everything completes the proof of Theorem A.1. $\qquad\square$

Finally, we add another useful lemma that quantifies the descent, if any, in the objective value between iterations.

**Lemma A.4.** *For every iteration $i$, we have*

$$f(\boldsymbol{x}_i) - f(\boldsymbol{x}_{i-1}) \leq -LD_h(\boldsymbol{x}_{i-1}, \boldsymbol{x}_i) + \langle \triangle_i, \boldsymbol{x}_i - \boldsymbol{x}_{i-1} \rangle.$$

*In particular, if $\langle \triangle_i, \boldsymbol{x}_i - \boldsymbol{x}_{i-1} \rangle \leq LD_h(\boldsymbol{x}_{i-1}, \boldsymbol{x}_i)$, then iteration $i$ is a descent step.*

*Proof of Lemma A.4.* We substitute $\boldsymbol{y} = \boldsymbol{x}_{i-1}$ in the conclusion of Lemma A.3. This gives

$$f(\boldsymbol{x}_i) - f(\boldsymbol{x}_{i-1}) \leq -LD_h(\boldsymbol{x}_{i-1}, \boldsymbol{x}_i) + \langle \triangle_i, \boldsymbol{x}_i - \boldsymbol{x}_{i-1} \rangle,$$

completing the proof of Lemma A.4. □

## B  OPTIMAL MS ACCELERATION UNDER CUSTOM EUCLIDEAN GEOMETRY

In this section, we adapt the bisection-free Monteiro-Svaiter acceleration framework developed in Carmon et al. (2022) to handle custom Euclidean geometries. The object of interest here is Algorithm 3, which we will call with different choices of the oracle $\mathcal{O}_{\mathsf{MS}}$ for our algorithms.

---

**Algorithm 3** OptimalMSAcceleration: optimizes function $f$ given MS oracle $\mathcal{O}_{\mathsf{MS}}$.

---

**Require:** Initial $\boldsymbol{x}_0$, function $f$, oracle $\mathcal{O}_{\mathsf{MS}}$, initial $\lambda_0'$, multiplicative adjustment factor $\alpha > 1$, iteration count $T$
1: Set $\boldsymbol{v}_0 = \boldsymbol{x}_0$, $A_0 = 0$, $A_0' = 0$.
2: Set $\widetilde{\boldsymbol{x}}_1, \lambda_1 = \mathcal{O}(\boldsymbol{x}_0; \lambda_0')$ and $\lambda_1' = \lambda_1$.
3: **for** $t = 0, \ldots, T$ **do**
4:    $a_{t+1}' = \frac{1}{2\lambda_{t+1}'}\left(1 + \sqrt{1 + 4\lambda_{t+1}' A_t}\right)$
5:    $A_{t+1}' = A_t + a_{t+1}'$
6:    $\boldsymbol{q}_t = \frac{A_t}{A_{t+1}'}\boldsymbol{x}_t + \frac{a_{t+1}'}{A_{t+1}'}\boldsymbol{v}_t$
7:    **if** $t > 0$ **then** $\widetilde{\boldsymbol{x}}_{t+1}, \lambda_{t+1} = \mathcal{O}_{\mathsf{MS}}(\boldsymbol{q}_t; \lambda_{t+1}')$
8:    $\gamma_{t+1} = \min\left\{1, \frac{\lambda_{t+1}'}{\lambda_{t+1}}\right\}$
9:    $a_{t+1} = \gamma_{t+1}a_{t+1}'$ and $A_{t+1} = A_t + a_{t+1}$        $\triangleright A_{t+1} = A_{t+1}' - (1 - \gamma_{t+1})a_{t+1}'$
10:    $\boldsymbol{x}_{t+1} = \frac{(1-\gamma_{t+1})A_t}{A_{t+1}}\boldsymbol{x}_t + \frac{\gamma_{t+1}A_{t+1}'}{A_{t+1}}\widetilde{\boldsymbol{x}}_{t+1}$
11:    **if** $\gamma_{t+1} = 1$ **then**
12:        $\lambda_{t+2}' = \frac{1}{\alpha}\lambda_{t+1}'$
13:    **else**
14:        $\lambda_{t+1}' = \alpha\lambda_{t+1}'$
15:    $\boldsymbol{v}_{t+1} = \boldsymbol{v}_t - a_{t+1}\mathbf{M}^{-1}\nabla f(\widetilde{\boldsymbol{x}}_{t+1})$

---

In order to state the performance guarantee of Algorithm 3, we require the notions of an *MS oracle* and a *movement bound*. See Definition B.1 and Definition B.2.

**Definition B.1** (MS oracle, generalization of (Carmon et al., 2022, Definition 1)). *Let $\mathbf{M} \in \mathbb{R}^{d \times d}$ be a positive semidefinite matrix. An oracle $\mathcal{O}: \mathbb{R}^d \times \mathbb{R}_{\geq 0} \to \mathbb{R}^d \times \mathbb{R}_{\geq 0}$ is a $\sigma$-MS oracle for function $f: \mathbb{R}^d \to \mathbb{R}$ if for every $\boldsymbol{q} \in \mathbb{R}^d$ and $\lambda' > 0$, the points $(\boldsymbol{x}, \lambda) = \mathcal{O}(\boldsymbol{q}; \lambda')$ satisfy*

$$\left\| \boldsymbol{x} - \boldsymbol{q} + \frac{1}{\lambda}\mathbf{M}^{-1}\nabla f(\boldsymbol{x}) \right\|_{\mathbf{M}} \leq \sigma \|\boldsymbol{x} - \boldsymbol{q}\|_{\mathbf{M}}.$$

**Definition B.2** (Movement bound (Carmon et al., 2022, Definition 2)). *For a norm $\|\cdot\|_{\mathbf{M}}$ induced by positive semidefinite $\mathbf{M} \in \mathbb{R}^{d \times d}$, numbers $s \geq 1, c, \lambda > 0$, and $\boldsymbol{x}, \boldsymbol{y} \in \mathbb{R}^d$, we say that $(\boldsymbol{x}, \boldsymbol{y}, \lambda)$ satisfies a $(s, c)$-movement bound if*

$$\|\boldsymbol{x} - \boldsymbol{y}\|_{\mathbf{M}} \geq \begin{cases} \left(\frac{\lambda}{c^s}\right)^{\frac{1}{s-1}} & \text{if } s < \infty \\ \frac{1}{c} & \text{if } s = \infty \end{cases}.$$

With these in hand, we are ready to state the convergence guarantee we get with Algorithm 3. See Theorem B.3.

**Theorem B.3** (Modification of (Carmon et al., 2022, Theorem 1)). *Let $f\colon \mathbb{R}^d \to \mathbb{R}$ be convex and differentiable. Consider running Algorithm 3 with parameters $\alpha = \exp\left(3 - \frac{2}{s+1}\right)$ and a $\sigma$-MS oracle with $0 \le \sigma < 0.99$ (Definition B.1). Let $s \ge 1$ and $c > 0$ and suppose that for all $t$ such that $\lambda_t > \lambda'_t$ or $t = 1$, the iterates $(\widetilde{\boldsymbol{x}}_t, \boldsymbol{q}_{t-1}, \lambda_t)$ satisfy an $(s, c)$-movement bound (Definition B.2). Let $C$ be a universal constant. For any iteration count $T$ satisfying*

$$
T \ge C \begin{cases} s\left(\frac{c^s \|\boldsymbol{x}_0 - \boldsymbol{x}^\star\|_{\mathbf{M}}^{s+1}}{\varepsilon}\right)^{\frac{2}{3s+1}} & \text{if } s < \infty \\ (c\|\boldsymbol{x}_0 - \boldsymbol{x}^\star\|_{\mathbf{M}})^{2/3} \log\left(\frac{\lambda_1 \|\boldsymbol{x}_0 - \boldsymbol{x}^\star\|_{\mathbf{M}}^2}{\varepsilon}\right) & \text{if } s = \infty \end{cases},
$$

*we have*

$$
f(\boldsymbol{x}_T) - f(\boldsymbol{x}^\star) \le \varepsilon.
$$

The proof of Theorem B.3 follows the same recipe as the proof of (Carmon et al., 2022, Theorem 1). The only modification needed is that stated in Lemma B.4.

**Lemma B.4** (Replaces (Carmon et al., 2022, Proposition 1)). *In the context of Theorem B.3, let $E_t := f(\boldsymbol{x}_t) - f(\boldsymbol{x}^\star)$, $D_t := \frac{1}{2}\|\boldsymbol{v}_t - \boldsymbol{x}^\star\|_{\mathbf{M}}^2$, $N_{t+1} := \frac{1}{2}\|\widetilde{\boldsymbol{x}}_{t+1} - \boldsymbol{q}_t\|_{\mathbf{M}}^2$. Then, for all $t \ge 0$, we have*

$$
A_{t+1} E_{t+1} + D_{t+1} + (1 - \sigma^2) A'_{t+1} \min\left\{\lambda_{t+1}, \lambda'_{t+1}\right\} N_{t+1} \le A_t E_t + D_t.
$$

*Consequently, for all $T \ge 1$, $\sqrt{A_T} \ge \frac{1}{2}\sum_{t \in \mathcal{S}_T^{\le}} \frac{1}{\sqrt{\lambda'_t}}$,*

$$
E_T \le \frac{D_0}{A_T} \quad \text{and} \quad (1 - \sigma^2) \sum_{t \in \mathcal{S}_T^{\ge}} A_t \lambda'_t N_t \le D_0 - A_T E_T.
$$

*Proof of Lemma B.4.* This proof is a straightforward modification of (Carmon et al., 2022, Proposition 1). We have

$$
D_{t+1} = \frac{1}{2}\|\boldsymbol{v}_{t+1} - \boldsymbol{x}^\star\|_{\mathbf{M}}^2 = \frac{1}{2}\left\|\boldsymbol{v}_t - a_{t+1}\mathbf{M}^{-1}\nabla f(\widetilde{\boldsymbol{x}}_{t+1}) - \boldsymbol{x}^\star\right\|_{\mathbf{M}}^2
$$

$$
= D_t + a_{t+1}\left\langle \mathbf{M}^{-1}\nabla f(\widetilde{\boldsymbol{x}}_{t+1}), \boldsymbol{x}^\star - \boldsymbol{v}_t\right\rangle_{\mathbf{M}} + \frac{a_{t+1}^2}{2}\left\|\mathbf{M}^{-1}\nabla f(\widetilde{\boldsymbol{x}}_{t+1})\right\|_{\mathbf{M}}^2.
$$

By definition of $\boldsymbol{q}_t$ and $A'_{t+1} = A_t + a'_{t+1}$, we have

$$
a'_{t+1}\boldsymbol{v}_t = A'_{t+1}\boldsymbol{q}_t - A_t\boldsymbol{x}_t = a'_{t+1}\widetilde{\boldsymbol{x}}_{t+1} + A'_{t+1}\left(\boldsymbol{q}_t - \widetilde{\boldsymbol{x}}_{t+1}\right) - A_t\left(\boldsymbol{x}_t - \widetilde{\boldsymbol{x}}_{t+1}\right).
$$

Subtracting $a'_{t+1}\boldsymbol{x}^\star$ and taking the inner product with $\mathbf{M}^{-1}\nabla f(\widetilde{\boldsymbol{x}}_{t+1})$ gives

$$
\begin{aligned}
&a'_{t+1}\left\langle \mathbf{M}^{-1}\nabla f(\widetilde{\boldsymbol{x}}_{t+1}), \boldsymbol{x}^\star - \boldsymbol{v}_t\right\rangle_{\mathbf{M}} \\
&= \left\langle \mathbf{M}^{-1}\nabla f(\widetilde{\boldsymbol{x}}_{t+1}), a'_{t+1}(\boldsymbol{x}^\star - \widetilde{\boldsymbol{x}}_{t+1}) + A'_{t+1}(\widetilde{\boldsymbol{x}}_{t+1} - \boldsymbol{q}_t) + A_t(\boldsymbol{x}_t - \widetilde{\boldsymbol{x}}_{t+1})\right\rangle_{\mathbf{M}} \\
&\le a'_{t+1}\left(f(\boldsymbol{x}^\star) - f(\widetilde{\boldsymbol{x}}_{t+1})\right) + A'_{t+1}\left\langle \mathbf{M}^{-1}\nabla f(\widetilde{\boldsymbol{x}}_{t+1}), \widetilde{\boldsymbol{x}}_{t+1} - \boldsymbol{q}_t\right\rangle_{\mathbf{M}} + A_t\left(f(\boldsymbol{x}_t) - f(\widetilde{\boldsymbol{x}}_{t+1})\right) \\
&\le A_t E_t - A'_{t+1}\left(f(\widetilde{\boldsymbol{x}}_{t+1}) - f(\boldsymbol{x}^\star)\right) + A'_{t+1}\left\langle \mathbf{M}^{-1}\nabla f(\widetilde{\boldsymbol{x}}_{t+1}), \widetilde{\boldsymbol{x}}_{t+1} - \boldsymbol{q}_t\right\rangle_{\mathbf{M}}.
\end{aligned}
$$

Rearranging gives

$$
\begin{aligned}
A'_{t+1}\left(f(\widetilde{\boldsymbol{x}}_{t+1}) - f(\boldsymbol{x}^\star)\right) \le{} & A_t E_t + a'_{t+1}\left\langle \mathbf{M}^{-1}\nabla f(\widetilde{\boldsymbol{x}}_{t+1}), \boldsymbol{v}_t - \boldsymbol{x}^\star\right\rangle_{\mathbf{M}} \\
& + A'_{t+1}\left\langle \mathbf{M}^{-1}\nabla f(\widetilde{\boldsymbol{x}}_{t+1}), \widetilde{\boldsymbol{x}}_{t+1} - \boldsymbol{q}_t\right\rangle_{\mathbf{M}}.
\end{aligned}
$$

Next, recall that by Definition B.1, we have

$$
\left\|\mathbf{M}^{-1}\nabla f(\widetilde{\boldsymbol{x}}_{t+1}) + \lambda_{t+1}\left(\widetilde{\boldsymbol{x}}_{t+1} - \boldsymbol{q}_t\right)\right\|_{\mathbf{M}}^2 \le \lambda_{t+1}^2\sigma^2\left\|\widetilde{\boldsymbol{x}}_{t+1} - \boldsymbol{q}_t\right\|_{\mathbf{M}}^2.
$$

We use this to write

$$\lambda_{t+1} \left\langle \mathbf{M}^{-1}\nabla f(\widetilde{\boldsymbol{x}}_{t+1}), \widetilde{\boldsymbol{x}}_{t+1} - \boldsymbol{q}_t \right\rangle_{\mathbf{M}}$$

$$= \frac{1}{2} \left\| \mathbf{M}^{-1}\nabla f(\widetilde{\boldsymbol{x}}_{t+1}) + \lambda_{t+1}(\widetilde{\boldsymbol{x}}_{t+1} - \boldsymbol{q}_t) \right\|_{\mathbf{M}}^2 - \frac{1}{2} \left\| \mathbf{M}^{-1}\nabla f(\widetilde{\boldsymbol{x}}_{t+1}) \right\|_{\mathbf{M}}^2 - \frac{\lambda_{t+1}^2}{2} \left\| \widetilde{\boldsymbol{x}}_{t+1} - \boldsymbol{q}_t \right\|_{\mathbf{M}}^2$$

$$\leq -\lambda_{t+1}^2(1-\sigma^2)N_{t+1} - \frac{1}{2} \left\| \mathbf{M}^{-1}\nabla f(\widetilde{\boldsymbol{x}}_{t+1}) \right\|_{\mathbf{M}}^2,$$

from which we conclude

$$\left\langle \mathbf{M}^{-1}\nabla f(\widetilde{\boldsymbol{x}}_{t+1}), \widetilde{\boldsymbol{x}}_{t+1} - \boldsymbol{q}_t \right\rangle_{\mathbf{M}} \leq -\lambda_{t+1}(1-\sigma^2)N_{t+1} - \frac{1}{2\lambda_{t+1}} \left\| \mathbf{M}^{-1}\nabla f(\widetilde{\boldsymbol{x}}_{t+1}) \right\|_{\mathbf{M}}^2.$$

Substituting back gives

$$A'_{t+1} \left( f(\widetilde{\boldsymbol{x}}_{t+1}) - f(\boldsymbol{x}^\star) \right) \leq A_t E_t + a'_{t+1} \left\langle \mathbf{M}^{-1}\nabla f(\widetilde{\boldsymbol{x}}_{t+1}), \boldsymbol{v}_t - \boldsymbol{x}^\star \right\rangle_{\mathbf{M}}$$

$$+ A'_{t+1} \left\langle \mathbf{M}^{-1}\nabla f(\widetilde{\boldsymbol{x}}_{t+1}), \widetilde{\boldsymbol{x}}_{t+1} - \boldsymbol{q}_t \right\rangle_{\mathbf{M}}$$

$$\leq A_t E_t + a'_{t+1} \left\langle \mathbf{M}^{-1}\nabla f(\widetilde{\boldsymbol{x}}_{t+1}), \boldsymbol{v}_t - \boldsymbol{x}^\star \right\rangle_{\mathbf{M}}$$

$$- A'_{t+1}\lambda_{t+1}(1-\sigma^2)N_{t+1} - \frac{A'_{t+1}}{2\lambda_{t+1}} \left\| \mathbf{M}^{-1}\nabla f(\widetilde{\boldsymbol{x}}_{t+1}) \right\|_{\mathbf{M}}^2.$$

Next, recall that $\gamma_{t+1}a'_{t+1} = a_{t+1}$ and $\gamma_{t+1}\lambda_{t+1} = \min\left\{\lambda_{t+1}, \lambda'_{t+1}\right\}$, by construction. Let $\widehat{\lambda}_{t+1} := \min\left\{\lambda_{t+1}, \lambda'_{t+1}\right\}$ We multiply both sides by $\gamma_{t+1}$ and conclude

$$\gamma_{t+1}A'_{t+1} \left( f(\widetilde{\boldsymbol{x}}_{t+1}) - f(\boldsymbol{x}^\star) \right) \leq \gamma_{t+1}A_t E_t + a_{t+1} \left\langle \mathbf{M}^{-1}\nabla f(\widetilde{\boldsymbol{x}}_{t+1}), \boldsymbol{v}_t - \boldsymbol{x}^\star \right\rangle_{\mathbf{M}}$$

$$- A'_{t+1}\widehat{\lambda}_{t+1}(1-\sigma^2)N_{t+1} - \frac{\gamma_{t+1}A'_{t+1}}{2\lambda_{t+1}} \left\| \mathbf{M}^{-1}\nabla f(\widetilde{\boldsymbol{x}}_{t+1}) \right\|_{\mathbf{M}}^2.$$

Now, by convexity of $f$ and from the definition of $\boldsymbol{x}_{t+1}$, we have

$$f(\boldsymbol{x}_{t+1}) - f(\boldsymbol{x}^\star) \leq \frac{(1-\gamma_{t+1})A_t}{A_{t+1}} \left( f(\boldsymbol{x}_t) - f(\boldsymbol{x}^\star) \right) + \frac{\gamma_{t+1}A'_{t+1}}{A_{t+1}} \left( f(\widetilde{\boldsymbol{x}}_{t+1}) - f(\boldsymbol{x}^\star) \right).$$

Recall the definition of $E_t$, multiply both sides by $A_{t+1}$, apply our bound on $\gamma_{t+1}A'_{t+1} \left( f(\widetilde{\boldsymbol{x}}_{t+1}) - f(\boldsymbol{x}^\star) \right)$, and we get

$$A_{t+1}E_{t+1} \leq (1-\gamma_{t+1})A_t E_t + \gamma_{t+1}A'_{t+1} \left( f(\widetilde{\boldsymbol{x}}_{t+1}) - f(\boldsymbol{x}^\star) \right)$$

$$\leq A_t E_t + a_{t+1} \left\langle \mathbf{M}^{-1}\nabla f(\widetilde{\boldsymbol{x}}_{t+1}), \boldsymbol{v}_t - \boldsymbol{x}^\star \right\rangle_{\mathbf{M}}$$

$$- A'_{t+1}\widehat{\lambda}_{t+1}(1-\sigma^2)N_{t+1} - \frac{\gamma_{t+1}A'_{t+1}}{2\lambda_{t+1}} \left\| \mathbf{M}^{-1}\nabla f(\widetilde{\boldsymbol{x}}_{t+1}) \right\|_{\mathbf{M}}^2$$

After shifting terms around, we see that it remains to show

$$a_{t+1} \left\langle \mathbf{M}^{-1}\nabla f(\widetilde{\boldsymbol{x}}_{t+1}), \boldsymbol{v}_t - \boldsymbol{x}^\star \right\rangle_{\mathbf{M}} - \frac{\gamma_{t+1}A'_{t+1}}{2\lambda_{t+1}} \left\| \mathbf{M}^{-1}\nabla f(\widetilde{\boldsymbol{x}}_{t+1}) \right\|_{\mathbf{M}}^2 \overset{?}{\leq} D_t - D_{t+1}.$$

In fact, by the choice of $a'_{t+1}$ and the definition of $A'_{t+1}$, we have

$$\lambda'_{t+1}(a'_{t+1})^2 = a'_{t+1} + A_t = A'_{t+1}.$$

Multiply both sides by $\gamma_{t+1}^2/(2\lambda'_{t+1})$ and we get

$$\frac{a_{t+1}^2}{2} = \frac{\gamma_{t+1}^2 A'_{t+1}}{2\lambda'_{t+1}} = \frac{\min\left\{1, \frac{\lambda'_{t+1}}{\lambda_{t+1}}\right\}\gamma_{t+1}A'_{t+1}}{2\lambda'_{t+1}} \leq \frac{\gamma_{t+1}A'_{t+1}}{2\lambda_{t+1}}.$$

We recycle an earlier computation and know that

$$D_t - D_{t+1} = a_{t+1} \left\langle \mathbf{M}^{-1}\nabla f(\widetilde{\boldsymbol{x}}_{t+1}), \boldsymbol{v}_t - \boldsymbol{x}^\star \right\rangle_{\mathbf{M}} - \frac{a_{t+1}^2}{2} \left\| \mathbf{M}^{-1}\nabla f(\widetilde{\boldsymbol{x}}_{t+1}) \right\|_{\mathbf{M}}^2$$

$$\geq a_{t+1} \left\langle \mathbf{M}^{-1}\nabla f(\widetilde{\boldsymbol{x}}_{t+1}), \boldsymbol{v}_t - \boldsymbol{x}^\star \right\rangle_{\mathbf{M}} - \frac{\gamma_{t+1}A'_{t+1}}{2\lambda_{t+1}} \left\| \mathbf{M}^{-1}\nabla f(\widetilde{\boldsymbol{x}}_{t+1}) \right\|_{\mathbf{M}}^2,$$

which completes the proof of the potential decrease.

The remaining statements follow as written in (Carmon et al., 2022, Proof of Proposition 1), and we conclude the proof of Lemma B.4. $\qquad\square$

Now that we have shown Lemma B.4, we refer the reader to (Carmon et al., 2022, Appendix A) for the proof of Theorem B.3, as it now follows exactly as written there.

We also give additional bounds on the movement of the iterates in $\|\cdot\|_{\mathbf{M}}$, which is a straightforward adaptation of (Carmon et al., 2020, Lemma 31) to the improved framework from Carmon et al. (2022).

**Lemma B.5.** *For all $t \geq 1$, we have both*

$$\|v_t - x^\star\|_{\mathbf{M}} \leq \sqrt{2}\, \|x_0 - x^\star\|_{\mathbf{M}}$$

$$\|x_t - x^\star\|_{\mathbf{M}} \leq \left( \sqrt{2} + \max_{1 \leq i \leq t} \frac{\lambda_i'}{\lambda_i} \cdot \sqrt{\frac{2}{1 - \sigma^2}} \right) \|x_0 - x^\star\|_{\mathbf{M}}.$$

In the statement of Lemma B.5, the cost of overshooting the guess $\lambda_i'$ becomes evident – without an additional strong convexity guarantee, it is challenging to ensure that each iterate remains in a small ball around $x^\star$. This is the main reason we are unable to apply the framework of Carmon et al. (2022) to the $p = \infty$ case.

*Proof of Lemma B.5.* Using the same notation as in Lemma B.4 and in that proof, we define

$$P_t := A_t E_t + D_t$$

$$\widehat{\lambda}_t := \min\{\lambda_t, \lambda_t'\}.$$

By induction on the conclusion of Lemma B.4, for $t \geq 1$ we have

$$\frac{1}{2}\, \|v_t - x^\star\|_{\mathbf{M}}^2 = D_t \leq P_t + (1 - \sigma^2) \sum_{k=1}^{t} A_k' \widehat{\lambda}_k N_k \leq P_0 = \|x_0 - x^\star\|_{\mathbf{M}}^2.$$

Thus,

$$\|v_t - x^\star\|_{\mathbf{M}} \leq \sqrt{2}\, \|x_0 - x^\star\|_{\mathbf{M}}.$$

For the second conclusion, we introduce the following notation.

$$\alpha_{t+1} := \frac{(1 - \gamma_{t+1}) A_t}{A_{t+1}}$$

$$\beta_{t+1} := \frac{A_t}{A_{t+1}'}$$

$$\delta_{t+1} := 1 - (1 - \alpha_{t+1})(1 - \beta_{t+1}) = 1 - \frac{\gamma_{t+1} A_{t+1}'}{A_{t+1}} \cdot \frac{a_{t+1}'}{A_{t+1}'} = \frac{A_t}{A_{t+1}}$$

We also establish for any $i$,

$$\frac{\gamma_i A_i'}{\lambda_i a_i^2} = \frac{A_i'}{\lambda_i \gamma_i (a_i')^2} = \frac{1}{\gamma_i} \cdot \frac{\lambda_i'}{\lambda_i} = \max\left\{ \frac{\lambda_i'}{\lambda_i}, 1 \right\},$$

which implies

$$\frac{\gamma_i A_i'}{\lambda_i} = a_i^2 \max\left\{ \frac{\lambda_i'}{\lambda_i}, 1 \right\}.$$

Notice that

$$\begin{aligned}
\|x_{t+1} - x^\star\|_{\mathbf{M}} &\leq \alpha_{t+1} \|x_t - x^\star\|_{\mathbf{M}} + (1 - \alpha_{t+1}) \|\widetilde{x}_{t+1} - x^\star\|_{\mathbf{M}} \\
&\leq \alpha_{t+1} \|x_t - x^\star\|_{\mathbf{M}} + (1 - \alpha_{t+1}) (\|q_t - x^\star\|_{\mathbf{M}} + \|\widetilde{x}_{t+1} - q_t\|_{\mathbf{M}}) \\
&\leq \alpha_{t+1} \|x_t - x^\star\|_{\mathbf{M}} \\
&\quad + (1 - \alpha_{t+1}) (\beta_{t+1} \|x_t - x^\star\|_{\mathbf{M}} + (1 - \beta_{t+1}) \|v_t - x^\star\|_{\mathbf{M}} + \|\widetilde{x}_{t+1} - q_t\|_{\mathbf{M}}) \\
&= (\beta_{t+1} + \alpha_{t+1} - \alpha_{t+1}\beta_{t+1}) \|x_t - x^\star\|_{\mathbf{M}} \\
&\quad + (1 - \alpha_{t+1}) (1 - \beta_{t+1}) \|v_t - x^\star\|_{\mathbf{M}} + (1 - \alpha_{t+1}) \|\widetilde{x}_{t+1} - q_t\|_{\mathbf{M}} \\
&= \delta_{t+1} \|x_t - x^\star\|_{\mathbf{M}} + (1 - \delta_{t+1}) \|v_t - x^\star\|_{\mathbf{M}} + (1 - \alpha_{t+1}) \|\widetilde{x}_{t+1} - q_t\|_{\mathbf{M}}
\end{aligned}$$

$$\leq \prod_{i=0}^{t} \delta_{i+1} \left\| \boldsymbol{x}_0 - \boldsymbol{x}^\star \right\|_{\mathbf{M}} + \left( 1 - \prod_{i=0}^{t} \delta_{i+1} \right) \left\| \boldsymbol{v}_t - \boldsymbol{x}^\star \right\|_{\mathbf{M}}$$

$$+ \sum_{i=1}^{t+1} \prod_{j=i+1}^{t+1} \delta_j (1 - \alpha_i) \left\| \widetilde{\boldsymbol{x}}_i - \boldsymbol{q}_{i-1} \right\|_{\mathbf{M}}$$

$$\leq \sqrt{2} \left\| \boldsymbol{x}_0 - \boldsymbol{x}^\star \right\|_{\mathbf{M}} + \sum_{i=1}^{t+1} \prod_{j=i+1}^{t+1} \delta_j (1 - \alpha_i) \left\| \widetilde{\boldsymbol{x}}_i - \boldsymbol{q}_{i-1} \right\|_{\mathbf{M}}$$

$$= \sqrt{2} \left\| \boldsymbol{x}_0 - \boldsymbol{x}^\star \right\|_{\mathbf{M}} + \sum_{i=1}^{t+1} \frac{A_i}{A_{t+1}} (1 - \alpha_i) \left\| \widetilde{\boldsymbol{x}}_i - \boldsymbol{q}_{i-1} \right\|_{\mathbf{M}}$$

$$= \sqrt{2} \left\| \boldsymbol{x}_0 - \boldsymbol{x}^\star \right\|_{\mathbf{M}} + \sum_{i=1}^{t+1} \frac{A_i}{A_{t+1}} \cdot \frac{\gamma_i A_i'}{A_i} \left\| \widetilde{\boldsymbol{x}}_i - \boldsymbol{q}_{i-1} \right\|_{\mathbf{M}}$$

$$= \sqrt{2} \left\| \boldsymbol{x}_0 - \boldsymbol{x}^\star \right\|_{\mathbf{M}} + \frac{1}{A_{t+1}} \sum_{i=1}^{t+1} \sqrt{\frac{\gamma_i A_i'}{\lambda_i}} \cdot \sqrt{\lambda_i \gamma_i A_i'} \left\| \widetilde{\boldsymbol{x}}_i - \boldsymbol{q}_{i-1} \right\|_{\mathbf{M}}$$

$$\leq \sqrt{2} \left\| \boldsymbol{x}_0 - \boldsymbol{x}^\star \right\|_{\mathbf{M}} + \frac{\left( \sum_{i=1}^{t+1} \frac{\gamma_i A_i'}{\lambda_i} \right)^{1/2}}{A_{t+1}} \cdot \left( \sum_{i=1}^{t+1} \lambda_i \gamma_i A_i' \left\| \widetilde{\boldsymbol{x}}_i - \boldsymbol{q}_{i-1} \right\|_{\mathbf{M}}^2 \right)^{1/2}$$

$$\leq \sqrt{2} \left\| \boldsymbol{x}_0 - \boldsymbol{x}^\star \right\|_{\mathbf{M}} + \frac{\left( \sum_{i=1}^{t+1} \frac{\gamma_i A_i'}{\lambda_i} \right)^{1/2}}{A_{t+1}} \cdot \sqrt{\frac{2}{1 - \sigma^2}} \left\| \boldsymbol{x}_0 - \boldsymbol{x}^\star \right\|_{\mathbf{M}}$$

$$\leq \sqrt{2} \left\| \boldsymbol{x}_0 - \boldsymbol{x}^\star \right\|_{\mathbf{M}} + \frac{\sum_{i=1}^{t+1} a_i \max\left\{ 1, \frac{\lambda_i'}{\lambda_i} \right\}}{A_{t+1}} \cdot \sqrt{\frac{2}{1 - \sigma^2}} \left\| \boldsymbol{x}_0 - \boldsymbol{x}^\star \right\|_{\mathbf{M}}$$

$$\leq \sqrt{2} \left\| \boldsymbol{x}_0 - \boldsymbol{x}^\star \right\|_{\mathbf{M}} + \max_{1 \leq i \leq t+1} \frac{\lambda_i'}{\lambda_i} \cdot \sqrt{\frac{2}{1 - \sigma^2}} \left\| \boldsymbol{x}_0 - \boldsymbol{x}^\star \right\|_{\mathbf{M}}$$

$$= \left( \sqrt{2} + \max_{1 \leq i \leq t+1} \frac{\lambda_i'}{\lambda_i} \cdot \sqrt{\frac{2}{1 - \sigma^2}} \right) \left\| \boldsymbol{x}_0 - \boldsymbol{x}^\star \right\|_{\mathbf{M}},$$

completing the proof of Lemma B.5. $\qquad\square$

## C  MINIMIZING THE DISTRIBUTIONALLY ROBUST LOSS

The goal of this section is to prove Theorem 1. We break up the proof into parts as described in Section 2. We structure the section as follows. In the rest of this subsection, we present Algorithm 1, our algorithm that minimizes the distributionally robust loss. In Appendix C.1, we introduce our smooth approximation for the objective equation 2 and show that it is a good additive approximation (this is a standard argument, but we include it as it provides crucial intuition).

As the main difficulty of the proof in Theorem 1 is to establish a Hessian stability for our surrogate loss, we devote the bulk of this section to proving this. Recall that in Section 2.1.1, we claimed that a higher-order smoothness condition called *quasi-self-concordance* gives us the needed Hessian stability – in fact, this follows from (Carmon et al., 2020, Lemma 11). In light of this, it suffices to demonstrate that our surrogate loss is quasi-self-concordant.

In Appendix C.2, we work out some calculus facts related to the softmax function. In particular, it is in Appendix C.2 that we prove the general composition result Lemma C.3 that states that if we take the softmax of several quasi-self-concordant functions, then the resulting function is also quasi-self-concordant. In Appendix C.3, we apply this composition fact to prove that our surrogate objective is quasi-self-concordant. Finally, in Appendix C.4, we combine these building blocks with the acceleration framework in Carmon et al. (2020) and complete the proof of Theorem 1.

### C.1 SMOOTHLY APPROXIMATING THE OBJECTIVE

Recall that for $\boldsymbol{y} \in \mathbb{R}^n$, let $\|\boldsymbol{y}\|_{\mathcal{G}_\infty} := \max_{1 \le i \le m} \|\boldsymbol{y}_{S_i}\|_2$, where for $\boldsymbol{y} \in \mathbb{R}^n$ we let $\boldsymbol{y}_{S_i}$ refer to the vector in $\mathbb{R}^{n_i}$ indexed by the indices in $S_i$. Also, for $\boldsymbol{y} \in \mathbb{R}^m$, let $\mathsf{lse}_\beta(\boldsymbol{y})$ refer to the function

$$\mathsf{lse}_\beta(\boldsymbol{y}) := \beta \log \left( \sum_{i=1}^m \exp \left( \frac{y_i}{\beta} \right) \right).$$

At a high level, our algorithm will minimize the function

$$\widetilde{f}_{\beta,\delta}(\boldsymbol{x}) := \beta \log \left( \sum_{i=1}^m \exp \left( \frac{\sqrt{\delta^2 + \|\mathbf{A}_{S_i}\boldsymbol{x} - \boldsymbol{b}_{S_i}\|_2^2} - \delta}{\beta} \right) \right)$$

for appropriate choices of the parameters $\beta$ and $\delta$. This choice of smoothening is natural because of the following approximation statement – see Lemma C.1.

**Lemma C.1.** *For all* $\boldsymbol{x} \in \mathbb{R}^d$, *we have*

$$\left| \widetilde{f}_{\beta,\delta}(\boldsymbol{x}) - \|\mathbf{A}\boldsymbol{x} - \boldsymbol{b}\|_{\mathcal{G}_\infty} \right| \le \beta \log m + \delta.$$

*Proof of Lemma C.1.* These guarantees are well-known, but we prove them anyway for the sake of self-containment. We first prove that for any $\boldsymbol{v} \in \mathbb{R}^m$, we have

$$\max_{1 \le i \le m} v_i \le \mathsf{lse}_\beta(\boldsymbol{v}) \le \max_{1 \le i \le m} v_i + \beta \log m.$$

In one direction, we have

$$\mathsf{lse}_\beta(\boldsymbol{v}) \le \beta \log \left( \sum_{i=1}^m \exp \left( \frac{\max_{1 \le i \le m} v_i}{\beta} \right) \right) = \beta \log m + \max_{1 \le i \le m} v_i,$$

and in the other, we have

$$\mathsf{lse}_\beta(\boldsymbol{v}) \ge \beta \log \left( \exp \left( \frac{\max_{1 \le i \le m} v_i}{\beta} \right) \right) = \max_{1 \le i \le m} v_i.$$

Next, for $\boldsymbol{v} \in \mathbb{R}^m$, we will show that

$$\|\boldsymbol{v}\|_2 - \delta \le \sqrt{\delta^2 + \|\boldsymbol{v}\|_2^2} - \delta \le \|\boldsymbol{v}\|_2.$$

Indeed, we have

$$\sqrt{\delta^2 + \|\boldsymbol{v}\|_2^2} - \delta \le \sqrt{\delta^2} + \sqrt{\|\boldsymbol{v}\|_2^2} - \delta = \|\boldsymbol{v}\|_2,$$

and

$$\sqrt{\delta^2 + \|\boldsymbol{v}\|_2^2} - \delta \ge \sqrt{\|\boldsymbol{v}\|_2^2} - \delta = \|\boldsymbol{v}\|_2 - \delta.$$

From this, we get

$$\widetilde{f}_{\beta,\delta}(\boldsymbol{x}) \le \max_{1 \le i \le m} \left( \sqrt{\delta^2 + \|\mathbf{A}_{S_i}\boldsymbol{x} - \boldsymbol{b}_{S_i}\|_2^2} - \delta \right) + \beta \log m \le \|\mathbf{A}\boldsymbol{x} - \boldsymbol{b}\|_{\mathcal{G}_\infty} + \beta \log m$$

and

$$\widetilde{f}_{\beta,\delta}(\boldsymbol{x}) \ge \beta \log \left( \sum_{i=1}^m \exp \left( \frac{\|\mathbf{A}_{S_i}\boldsymbol{x} - \boldsymbol{b}_{S_i}\|_2 - \delta}{\beta} \right) \right) \ge \|\mathbf{A}\boldsymbol{x} - \boldsymbol{b}\|_{\mathcal{G}_\infty} - \delta.$$

Putting these together gives

$$\left| \widetilde{f}_{\beta,\delta}(\boldsymbol{x}) - \|\mathbf{A}\boldsymbol{x} - \boldsymbol{b}\|_{\mathcal{G}_\infty} \right| \le \max(\beta \log m, \delta) \le \beta \log m + \delta,$$

completing the proof of Lemma C.1. $\square$

Eventually, we will choose $\beta = \varepsilon/(4 \log m)$ and $\delta = \varepsilon/4$ and then minimize $\widetilde{f}_{\beta,\delta}$ to $\varepsilon/2$ additive error. In light of Lemma C.1, this will be enough to get an $\varepsilon$-additive approximation to the optimum for $\|\mathbf{A}\boldsymbol{x} - \boldsymbol{b}\|_{\mathcal{G}_\infty}$.

## C.2 CALCULUS FOR LOGSUMEXP

We investigate certain properties of $\mathsf{lse}_\beta(\boldsymbol{y})$ when each entry $[\boldsymbol{y}]_i$ is a function $h_i(t)$ for $t \in \mathbb{R}$ for all $i \in [m]$. Let $h(t) \in \mathbb{R}^m$ denote the vector where its $i$th entry is given by $h_i(t)$. We treat each $h_i$ as a one-dimensional restriction of a function $g_i \colon \mathbb{R}^m \to \mathbb{R}$, so $h_i(t) = g_i(\boldsymbol{y} + t\boldsymbol{d})$ for center $\boldsymbol{y}$ and direction $\boldsymbol{d}$ (we omit the parameters $\boldsymbol{y}, \boldsymbol{d}$ in the notation $h_i$ as it will be clear from context). Finally, recall the definition of quasi-self-concordance (Definition 2.1).

We begin with calculating the first two derivatives of $\mathsf{lse}_\beta(h(t))$ with respect to $t$ in Lemma C.2.

**Lemma C.2.** *Let* $\lambda_i(t) := \exp\left(h_i(t)/\beta\right)$. *Then, we have*

$$\left(\frac{d}{dt}\right) \mathsf{lse}_\beta(h(t)) = \frac{\sum_{i=1}^m \left(\lambda_i(t) \cdot h_i'(t)\right)}{\sum_{i=1}^m \lambda_i(t)}$$

$$\left(\frac{d}{dt}\right)^2 \mathsf{lse}_\beta(h(t)) = \frac{1}{\beta} \left( \frac{\sum_{i=1}^m \lambda_i(t) h_i'(t)^2}{\sum_{i=1}^m \lambda_i(t)} - \left( \frac{\sum_{i=1}^m \lambda_i(t) h_i'(t)}{\sum_{i=1}^m \lambda_i(t)} \right)^2 \right) + \frac{\sum_{i=1}^m \lambda_i(t) h_i''(t)}{\sum_{i=1}^m \lambda_i(t)}.$$

*Proof of Lemma C.2.* The first derivative follows from the chain rule. Indeed, we have

$$\mathsf{lse}_\beta'(h(t)) = \beta \cdot \frac{\sum_{i=1}^m \lambda_i'(t)}{\sum_{i=1}^m \lambda_i(t)} = \beta \cdot \frac{\sum_{i=1}^m \left(\lambda_i(t) \cdot \frac{h_i'(t)}{\beta}\right)}{\sum_{i=1}^m \lambda_i(t)} = \frac{\sum_{i=1}^m \left(\lambda_i(t) \cdot h_i'(t)\right)}{\sum_{i=1}^m \lambda_i(t)} \leq \max_i h_i'(t).$$

For the second derivative, we use the differentiation rule for multiplication and division and the chain rule, giving

$$\mathsf{lse}_\beta''(h(t)) = \frac{\left[\left(\sum_{i=1}^m \lambda_i'(t) h_i'(t) + \lambda_i(t) h_i''(t)\right)\left(\sum_{i=1}^m \lambda_i(t)\right)\right] - \frac{1}{\beta}\left(\sum_{i=1}^m \lambda_i(t) h_i'(t)\right)^2}{\left(\sum_{i=1}^m \lambda_i(t)\right)^2}$$

$$= \frac{\left[\frac{1}{\beta}\left(\sum_{i=1}^m \lambda_i(t) h_i'(t)^2 + \beta \lambda_i(t) h_i''(t)\right)\left(\sum_{i=1}^m \lambda_i(t)\right)\right] - \frac{1}{\beta}\left(\sum_{i=1}^m \lambda_i(t) h_i'(t)\right)^2}{\left(\sum_{i=1}^m \lambda_i(t)\right)^2}$$

$$= \frac{1}{\beta} \left( \frac{\sum_{i=1}^m \lambda_i(t) h_i'(t)^2}{\sum_{i=1}^m \lambda_i(t)} - \frac{\left(\sum_{i=1}^m \lambda_i(t) h_i'(t)\right)^2}{\left(\sum_{i=1}^m \lambda_i(t)\right)^2} \right) + \frac{\sum_{i=1}^m \lambda_i(t) h_i''(t)}{\sum_{i=1}^m \lambda_i(t)}.$$

This completes the proof of Lemma C.2. $\square$

Next, we prove a general fact regarding composing $\mathsf{lse}$ with a vector formed by functions that are themselves quasi self concordant. See Lemma C.3.

**Lemma C.3** (Composing softmax with quasi-self-concordant functions). *Let* $\|\cdot\|$ *be an arbitrary norm and* $h_1, \ldots, h_m$ *be such that* $h_i \colon \mathbb{R}^d \to \mathbb{R}$. *Let* $h$ *be the vector formed by concatenating the results of* $h_1, \ldots, h_m$. *Additionally, let* $h_1, \ldots, h_m$ *be such that for all* $1 \leq i \leq m$ *and for all* $\boldsymbol{y}, \boldsymbol{d} \in \mathbb{R}^m$ *and* $t \in \mathbb{R}$,

$$\left(\frac{d}{dt}\right) h_i(\boldsymbol{y} + t\boldsymbol{d}) \leq \|\boldsymbol{d}\| \qquad\qquad\qquad \text{(Lipschitzness)}$$

$$\left| \left(\frac{d}{dt}\right)^3 h_i(\boldsymbol{y} + t\boldsymbol{d}) \right| \leq \nu \|\boldsymbol{d}\| \left(\frac{d}{dt}\right)^2 h_i(\boldsymbol{y} + t\boldsymbol{d}) \qquad \text{(quasi-self-concordance)}.$$

*Then, for all* $\boldsymbol{y}, \boldsymbol{d} \in \mathbb{R}^m$ *and all* $t \in \mathbb{R}$, *we have*

$$\left| \left(\frac{d}{dt}\right)^3 \beta \log \left( \sum_{i=1}^m \exp \left( \frac{h_i(\boldsymbol{y} + t\boldsymbol{d})}{\beta} \right) \right) \right| \leq \left( \frac{16}{\beta} + \nu \right) \|\boldsymbol{d}\| \left(\frac{d}{dt}\right)^2 \mathsf{lse}_\beta(h(\boldsymbol{y} + t\boldsymbol{d})).$$

As far as we are aware, this type of composition result was not previously known and may be of independent interest.

To prove Lemma C.3, we need Lemma C.4.

**Lemma C.4.** *For any two random variables $X, Y$, we have*

$$\text{Var}\,[XY] \leq 2\,\|Y\|_\infty^2\,\text{Var}\,[X] + 2\,\|X\|_\infty^2\,\text{Var}\,[Y]\,.$$

*Proof of Lemma C.4.* The proof follows that of Giraudo (2014), but we reproduce it here for completeness. First, notice that for random variables $U, V$, we have

$$2\text{Var}\,[U] + 2\text{Var}\,[V] - \text{Var}\,[U+V] = \text{Var}\,[U] + \text{Var}\,[V] - 2\text{Cov}\,[U,V] = \text{Var}\,[U-V] \geq 0.$$

Let $U = (X - \mathbb{E}\,[X])Y$ and $V = \mathbb{E}\,[X]\,Y$. Then, $U + V = XY$, and we have

$$\text{Var}\,[XY] \leq 2\text{Var}\,[(X - \mathbb{E}\,[X])Y] + 2\text{Var}\,[\mathbb{E}\,[X]\,Y] = 2\text{Var}\,[(X - \mathbb{E}\,[X])Y] + 2\mathbb{E}\,[X]^2\,\text{Var}\,[Y]\,.$$

It remains to bound $\text{Var}\,[(X - \mathbb{E}\,[X])Y]$. By Hölder's inequality, we have

$$\text{Var}\,[(X - \mathbb{E}\,[X])Y] \leq \mathbb{E}\,[((X - \mathbb{E}\,[X])Y)^2] \leq \mathbb{E}\,[(X - \mathbb{E}\,[X])^2]\,\|Y\|_\infty^2 = \text{Var}\,[X]\,\|Y\|_\infty^2\,.$$

Combining everything gives us the conclusion of Lemma C.4. $\qquad\square$

We are now ready to prove Lemma C.3.

*Proof of Lemma C.3.* Let $\lambda_i(t) := \exp(h_i(t)/\beta)$.

In this proof, we will encounter many weighted averages of vectors $\boldsymbol{z} \in \mathbb{R}^m$ of the form

$$\frac{\sum_{i=1}^m \lambda_i(t) z_i}{\sum_{i=1}^m \lambda_i(t)}\,.$$

Let $\mathcal{D}$ be the distribution over $[m]$ whose entries are given by $\mathcal{D}_j = \lambda_j(t)/\sum_{i=1}^m \lambda_i(t)$. In the rest of this proof, all expected values, variances, and covariances will be taken with respect to this distribution. In an abuse of notation, let $h(t)$ denote the "random" variable that is $h_i(t)$ with probability $\mathcal{D}_i$. Define $h'(t), h''(t), h'''(t)$ analogously.

To find the third derivative of $\text{lse}_\beta(h(t))$, we start with its second derivative. By Lemma C.2, it is given by

$$\text{lse}_\beta''(h(t)) = \underbrace{\frac{1}{\beta}\left(\frac{\sum_{i=1}^m \lambda_i(t) h_i'(t)^2}{\sum_{i=1}^m \lambda_i(t)} - \left(\frac{\sum_{i=1}^m \lambda_i(t) h_i'(t)}{\sum_{i=1}^m \lambda_i(t)}\right)^2\right)}_{T_1} + \underbrace{\frac{\sum_{i=1}^m \lambda_i(t) h_i''(t)}{\sum_{i=1}^m \lambda_i(t)}}_{T_2}$$

$$= \frac{1}{\beta}\text{Var}\,[h'(t)] + \mathbb{E}\,[h''(t)]\,.$$

We now differentiate the above term by term. First, we have

$$T_2'(t) = \frac{\sum_{i=1}^m \lambda_i(t)\left(\left(\frac{h_i'(t) h_i''(t)}{\beta}\right) + h_i'''(t)\right)}{\sum_{i=1}^m \lambda_i(t)} - \frac{1}{\beta} \cdot \frac{\left(\sum_{i=1}^m \lambda_i(t) h_i'(t)\right)\left(\sum_{i=1}^m \lambda_i(t) h_i''(t)\right)}{\left(\sum_{i=1}^m \lambda_i(t)\right)^2}$$

$$= \frac{1}{\beta}\left(\frac{\sum_{i=1}^m \lambda_i(t) h_i'(t) h_i''(t)}{\sum_{i=1}^m \lambda_i(t)} - \frac{\left(\sum_{i=1}^m \lambda_i(t) h_i'(t)\right)\left(\sum_{i=1}^m \lambda_i(t) h_i''(t)\right)}{\left(\sum_{i=1}^m \lambda_i(t)\right)^2}\right) + \frac{\sum_{i=1}^m \lambda_i(t) h_i'''(t)}{\sum_{i=1}^m \lambda_i(t)}$$

$$= \frac{1}{\beta}\text{Cov}\,[h'(t), h''(t)] + \mathbb{E}\,[h'''(t)]\,.$$

Next, we have

$$\frac{d}{dt}\mathbb{E}\,[h'(t)]^2 = 2\mathbb{E}\,[h'(t)] \cdot \frac{d}{dt}\mathbb{E}\,[h'(t)] = 2\mathbb{E}\,[h'(t)]\left(\frac{1}{\beta}\text{Var}\,[h'(t)] + \mathbb{E}\,[h''(t)]\right)$$

and

$$\frac{d}{dt}\mathbb{E}\,[h'(t)^2]$$

$$
= \frac{\left(\sum_{i=1}^m \lambda_i'(t) h_i'(t)^2 + 2 h_i'(t) h_i''(t) \lambda_i(t)\right)\left(\sum_{i=1}^m \lambda_i(t)\right) - \frac{1}{\beta}\left(\sum_{i=1}^m \lambda_i(t) h_i'(t)\right)\left(\sum_{i=1}^m \lambda_i(t) h_i'(t)^2\right)}{\left(\sum_{i=1}^m \lambda_i(t)\right)^2}
$$

$$
= \frac{\left(\sum_{i=1}^m \lambda_i'(t) h_i'(t)^2 + 2 h_i'(t) h_i''(t) \lambda_i(t)\right)}{\sum_{i=1}^m \lambda_i(t)} - \frac{1}{\beta} \cdot \frac{\left(\sum_{i=1}^m \lambda_i(t) h_i'(t)\right)\left(\sum_{i=1}^m \lambda_i(t) h_i'(t)^2\right)}{\left(\sum_{i=1}^m \lambda_i(t)\right)^2}
$$

$$
= \frac{\sum_{i=1}^m \lambda_i(t)\left(\frac{h_i'(t)^3}{\beta} + 2 h_i'(t) h_i''(t)\right)}{\sum_{i=1}^m \lambda_i(t)} - \frac{1}{\beta} \cdot \frac{\left(\sum_{i=1}^m \lambda_i(t) h_i'(t)\right)\left(\sum_{i=1}^m \lambda_i(t) h_i'(t)^2\right)}{\left(\sum_{i=1}^m \lambda_i(t)\right)^2}
$$

$$
= \frac{1}{\beta}\mathsf{Cov}\left[h'(t), h'(t)^2\right] + 2\mathbb{E}\left[h'(t) h''(t)\right].
$$

Combining everything gives us

$$
\mathsf{lse}_\beta'''(h(t))
$$
$$
= \frac{1}{\beta}\left(\frac{1}{\beta}\mathsf{Cov}\left[h'(t), h'(t)^2\right] + 2\mathbb{E}\left[h'(t) h''(t)\right] - 2\mathbb{E}\left[h'(t)\right]\left(\frac{1}{\beta}\mathsf{Var}\left[h'(t)\right] + \mathbb{E}\left[h''(t)\right]\right)\right)
$$
$$
\quad + \frac{1}{\beta}\mathsf{Cov}\left[h'(t), h''(t)\right] + \mathbb{E}\left[h'''(t)\right]
$$
$$
= \frac{1}{\beta^2}\mathsf{Cov}\left[h'(t), h'(t)^2\right] - \frac{2}{\beta^2}\mathbb{E}\left[h'(t)\right]\mathsf{Var}\left[h'(t)\right] + \frac{3}{\beta}\mathsf{Cov}\left[h'(t), h''(t)\right] + \mathbb{E}\left[h'''(t)\right].
$$

We first analyze the terms that only depend on $h'(t)$. To do so, we use Lemma C.4 to write

$$
\left|\mathsf{Cov}\left[h'(t), h'(t)^2\right]\right| \le \sqrt{\mathsf{Var}\left[h'(t)\right]}\sqrt{\mathsf{Var}\left[h'(t)^2\right]} \le 2\left\|\boldsymbol{d}\right\|\mathsf{Var}\left[h'(t)\right].
$$

Now, we have

$$
\frac{1}{\beta^2}\left|\mathsf{Cov}\left[h'(t), h'(t)^2\right] - 2\mathbb{E}\left[h'(t)\right]\mathsf{Var}\left[h'(t)\right]\right|
$$
$$
\le \frac{1}{\beta^2}\left|\mathsf{Cov}\left[h'(t), h'(t)^2\right]\right| + \frac{2}{\beta^2}\left|\mathbb{E}\left[h'(t)\right]\mathsf{Var}\left[h'(t)\right]\right|
$$
$$
\le \frac{4}{\beta^2}\left\|\boldsymbol{d}\right\|\mathsf{Var}\left[h'(t)\right] \le \frac{4}{\beta}\left\|\boldsymbol{d}\right\|\mathsf{lse}_\beta''(h(t)).
$$

Next, we take care of the remaining terms. We have

$$
\frac{3}{\beta}\left|\mathsf{Cov}\left[h'(t), h''(t)\right]\right| + \left|\mathbb{E}\left[h'''(t)\right]\right| \le \frac{6}{\beta}\left(\max_i h_i'(t)\right)\mathbb{E}\left[\left|h''(t) - \mathbb{E}\left[h''(t)\right]\right|\right] + \left|\mathbb{E}\left[h'''(t)\right]\right|
$$
$$
\le \frac{12}{\beta}\left\|\boldsymbol{d}\right\|\mathsf{lse}_\beta''(h(t)) + \mathbb{E}\left[\left|h'''(t)\right|\right]
$$
$$
\le \frac{12}{\beta}\left\|\boldsymbol{d}\right\|\mathsf{lse}_\beta''(h(t)) + \nu\left\|\boldsymbol{d}\right\|\mathbb{E}\left[h''(t)\right]
$$
$$
\le \left(\frac{12}{\beta} + \nu\right)\left\|\boldsymbol{d}\right\|\mathsf{lse}_\beta''(h(t)),
$$

where the penultimate line follows from Lemma C.7. Combining these conclusions yields

$$
\left|\mathsf{lse}_\beta'''(h(t))\right| \le \left(\frac{16}{\beta} + \nu\right)\left\|\boldsymbol{d}\right\|\mathsf{lse}_\beta''(h(t)),
$$

completing the proof of Lemma C.3. $\qquad\square$

## C.3 SMOOTHNESS AND QUASI-SELF-CONCORDANCE OF THE MODIFIED OBJECTIVE

The main result of this subsection is Lemma C.5.

**Lemma C.5.** *Let* $\mathbf{W}$ *be such that for all* $\boldsymbol{z} \in \mathbb{R}^d$, *we have* $\left\|\mathbf{A}\boldsymbol{z}\right\|_{\mathcal{G}_\infty} \le \left\|\mathbf{W}^{1/2}\mathbf{A}\boldsymbol{z}\right\|_2$. *For all* $\boldsymbol{x}, \boldsymbol{z} \in \mathbb{R}^d$ *and* $t \in \mathbb{R}$, *we have*

$$
\left(\frac{d}{dt}\right)^2 \widetilde{f}_{\beta,\delta}(\boldsymbol{x} + t\boldsymbol{z}) \le \left(\frac{1}{\delta} + \frac{1}{\beta}\right)\left\|\mathbf{W}^{1/2}\mathbf{A}\boldsymbol{z}\right\|_2^2 \qquad\text{(smoothness)}
$$

$$\left| \left( \frac{d}{dt} \right)^3 \widetilde{f}_{\beta,\delta}(\boldsymbol{x} + t\boldsymbol{z}) \right| \leq \left( \frac{16}{\delta} + \frac{3}{\beta} \right) \left\| \mathbf{W}^{1/2} \mathbf{A} \boldsymbol{z} \right\|_2 \left( \frac{d}{dt} \right)^2 \widetilde{f}_{\beta,\delta}(\boldsymbol{x} + t\boldsymbol{z}) \quad \text{(quasi-self-concordance)}.$$

Our goal in the rest of this section is to prove Lemma C.5.

We begin with defining $h_i(t)$ as (absorb the $\delta, \boldsymbol{y}, \boldsymbol{d}$ parameters into the definition of $h_i$)

$$h_i(t) := \sqrt{\delta^2 + \|\boldsymbol{y}_{S_i} + t\boldsymbol{d}_{S_i}\|_2^2}.$$

Let $h(t)$ denote the vector whose $i$th entry is $h_i(t)$. Then, observe that

$$\mathsf{lse}_\beta(h(t)) = \beta \log \left( \sum_{i=1}^m \exp \left( \frac{h_i(t)}{\beta} \right) \right) = \beta \log \left( \sum_{i=1}^m \exp \left( \frac{\sqrt{\delta^2 + \|\boldsymbol{y}_{S_i} + t\boldsymbol{d}_{S_i}\|_2^2}}{\beta} \right) \right).$$

It is easy to see that every one-dimensional restriction of $\widetilde{f}_{\beta,\delta}$ can be obtained by an affine transformation of $\mathsf{lse}_\beta(h(t))$ after appropriate choices of $\boldsymbol{y}, \boldsymbol{d} \in \mathbb{R}^m$. Hence, we first analyze $\mathsf{lse}_\beta(h(t))$ for all $\boldsymbol{y}, \boldsymbol{d} \in \mathbb{R}^m$.

We begin with proving the smoothness of $\mathsf{lse}_\beta(h(t))$ with respect to $\|\cdot\|_{\mathcal{G}_\infty}$.

**Lemma C.6.** *For all $\boldsymbol{y}, \boldsymbol{d} \in \mathbb{R}^m$ and all $t \in \mathbb{R}$, we have*

$$\left( \frac{d}{dt} \right)^2 \mathsf{lse}_\beta(h(t)) \leq \left( \frac{1}{\delta} + \frac{1}{\beta} \right) \|\boldsymbol{d}\|_{\mathcal{G}_\infty}^2 .$$

*Proof of Lemma C.6.* By direct calculation, it is easy to see that

$$h_i'(t) = \frac{\langle \boldsymbol{y}_{S_i} + t\boldsymbol{d}_{S_i}, \boldsymbol{d}_{S_i} \rangle}{h_i(t)}$$

$$h_i''(t) = \frac{\|\boldsymbol{d}_{S_i}\|_2^2 \, h_i(t) - h_i'(t)^2 h_i(t)}{h_i(t)^2} = \frac{\|\boldsymbol{d}_{S_i}\|_2^2 - h_i'(t)^2}{h_i(t)}. \tag{9}$$

We plug this into the result of Lemma C.2 and get

$$\mathsf{lse}_\beta''(h(t)) \leq \frac{1}{\beta} \max_i h_i'(t)^2 + \max_i h_i''(t)$$

$$= \frac{1}{\beta} \max_i \left( \frac{\langle \boldsymbol{y}_{S_i} + t\boldsymbol{d}_{S_i}, \boldsymbol{d}_{S_i} \rangle}{\sqrt{\delta^2 + \|\boldsymbol{y}_{S_i} + t\boldsymbol{d}_{S_i}\|_2^2}} \right)^2 + \max_i \frac{\|\boldsymbol{d}_{S_i}\|_2^2 - h_i'(t)^2}{\sqrt{\delta^2 + \|\boldsymbol{y}_{S_i} + t\boldsymbol{d}_{S_i}\|_2^2}}$$

$$\leq \frac{1}{\beta} \max_i \|\boldsymbol{d}_{S_i}\|_2^2 + \frac{1}{\delta} \max_i \|\boldsymbol{d}_{S_i}\|_2^2 = \left( \frac{1}{\beta} + \frac{1}{\delta} \right) \|\boldsymbol{d}\|_{\mathcal{G}_\infty}^2 ,$$

completing the proof of Lemma C.6. $\qquad\qquad\square$

Our next task is to show that $\mathsf{lse}_\beta(h(t))$ is $O(1/\beta + 1/\delta)$-quasi-self-concordant in $\|\cdot\|_{\mathcal{G}_\infty}$. To do so, we will appeal to Lemma C.3. To be able to do this, we first have to prove the quasi-self-concordance of each component function in $\mathsf{lse}_\beta(h(t))$.

**Lemma C.7.** *For all $\boldsymbol{y}, \boldsymbol{d} \in \mathbb{R}^m$ and all $t \in \mathbb{R}$, we have*

$$\left| \left( \frac{d}{dt} \right)^3 \sqrt{\delta^2 + \|\boldsymbol{y}_{S_i} + t\boldsymbol{d}_{S_i}\|_2^2} \right| \leq \frac{3}{\delta} \|\boldsymbol{d}_{S_i}\|_2 \left( \left( \frac{d}{dt} \right)^2 \sqrt{\delta^2 + \|\boldsymbol{y}_{S_i} + t\boldsymbol{d}_{S_i}\|_2^2} \right).$$

*Proof of Lemma C.7.* Although a similar fact appears in (Ostrovskii & Bach, 2020, Section 2.1.2), it is not in the exact form we need. So, we prove the required statement here.

Recycling the computation from equation 9, recall

$$h_i''(t) = \frac{\|\boldsymbol{d}_{S_i}\|_2^2 - h_i'(t)^2}{h_i(t)},$$

which gives

$$h_i'''(t) = \frac{-2h_i'(t)h_i''(t)h_i(t) - h_i'(t)(h_i(t)h_i''(t))}{h_i(t)^2} = -\frac{3h_i'(t)h_i''(t)}{h_i(t)}.$$

Finally, again recalling equation 9, notice that

$$\left|\frac{h_i'(t)}{h_i(t)}\right| = \left|\frac{\langle \boldsymbol{y}_{S_i} + t\boldsymbol{d}_{S_i}, \boldsymbol{d}_{S_i}\rangle}{h_i(t)^2}\right| = \left|\left\langle \frac{\boldsymbol{y}_{S_i} + t\boldsymbol{d}_{S_i}}{\sqrt{\delta^2 + \|\boldsymbol{y}_{S_i} + t\boldsymbol{d}_{S_i}\|_2^2}}, \frac{\boldsymbol{d}_{S_i}}{\sqrt{\delta^2 + \|\boldsymbol{y}_{S_i} + t\boldsymbol{d}_{S_i}\|_2^2}}\right\rangle\right| \le \frac{\|\boldsymbol{d}_{S_i}\|_2}{\delta}.$$

Combining everything completes the proof of Lemma C.7. $\qquad\square$

We are now ready to prove the quasi-self-concordance of $\mathsf{lse}_\beta(h(t))$ in $\|\cdot\|_{\mathcal{G}_\infty}$.

**Lemma C.8.** *For all $\boldsymbol{y}, \boldsymbol{d} \in \mathbb{R}^m$ and $t \in \mathbb{R}$, we have*

$$\left|\left(\frac{d}{dt}\right)^3 \mathsf{lse}_\beta(h(t))\right| \le \left(\frac{16}{\beta} + \frac{3}{\delta}\right)\|\boldsymbol{d}\|_{\mathcal{G}_\infty}\left(\frac{d}{dt}\right)^2 \mathsf{lse}_\beta(h(t)).$$

*Proof of Lemma C.8.* In the statement of Lemma C.3, let $\|\cdot\| = \|\cdot\|_{\mathcal{G}_\infty}$. By the definition of $\|\cdot\|_{\mathcal{G}_\infty}$ and $h_i$, we have for all $i$ and $t$ that $h_i'(t) \le \|\boldsymbol{d}\|_{\mathcal{G}_\infty}$. Additionally, from Lemma C.7, we have that the $h_i(t)$ are $3/\delta$-quasi-self-concordant in the norm $\|\boldsymbol{d}\|_{\mathcal{G}_\infty}$ for all $i$. Lemma C.8 now follows immediately from Lemma C.3. $\qquad\square$

Finally, we can prove Lemma C.5.

*Proof of Lemma C.5.* By the conclusion of Lemma C.6, we know that for all $\boldsymbol{y}, \boldsymbol{d} \in \mathbb{R}^m$ and $t \in \mathbb{R}$ that

$$\left(\frac{d}{dt}\right)^2 \mathsf{lse}_\beta(h(t)) \le \left(\frac{1}{\delta} + \frac{1}{\beta}\right)\|\boldsymbol{z}\|_{\mathcal{G}_\infty}^2.$$

Let $\boldsymbol{y} = \mathbf{A}\boldsymbol{x} - \boldsymbol{b}$ for some $\boldsymbol{x}$ and $\boldsymbol{d} = \mathbf{A}\boldsymbol{z}$ for some $\boldsymbol{z}$. Let

$$g(\boldsymbol{y}) := \beta \log\left(\sum_{i=1}^m \exp\left(\frac{\sqrt{\delta^2 + \|\boldsymbol{y}_{S_i}\|_2^2} - \delta}{\beta}\right)\right).$$

Then,

$$\left(\frac{d}{dt}\right)^2 \widetilde{f}_{\beta,\delta}(\boldsymbol{x} + t\boldsymbol{z}) = \left(\frac{d}{dt}\right)^2 g(\mathbf{A}\boldsymbol{x} - \boldsymbol{b} + t\mathbf{A}\boldsymbol{z}) \le \left(\frac{1}{\delta} + \frac{1}{\beta}\right)\|\mathbf{A}\boldsymbol{z}\|_{\mathcal{G}_\infty}^2.$$

With the exact same reasoning applied to the conclusion of Lemma C.8, we also see that

$$\left|\left(\frac{d}{dt}\right)^3 \widetilde{f}_{\beta,\delta}(\boldsymbol{x} + t\boldsymbol{z})\right| \le \left(\frac{16}{\delta} + \frac{3}{\beta}\right)\|\mathbf{A}\boldsymbol{z}\|_{\mathcal{G}_\infty}\left(\frac{d}{dt}\right)^2 \widetilde{f}_{\beta,\delta}(\boldsymbol{x} + t\boldsymbol{z}).$$

The conclusion of Lemma C.5 then follows from remembering that we have $\mathbf{W}$ such that for all $\boldsymbol{z} \in \mathbb{R}^d$, $\|\mathbf{A}\boldsymbol{z}\|_{\mathcal{G}_\infty} \le \|\mathbf{W}^{1/2}\mathbf{A}\boldsymbol{z}\|_2$ (following from Theorem 2.3). $\qquad\square$

## C.4    ANALYSIS OF ALGORITHM 1

In this subsection, we use the calculus facts from the previous two subsections to analyze Algorithm 1. The outline of this proof follows that of (Jambulapati et al., 2022, Theorem 2), which in turn builds up to using the proof used in (Carmon et al., 2020, Corollary 12). The main idea is to define the algorithm based on the norm given by a good choice of positive semidefinite $\mathbf{M}$, given by Theorem 2.3.

In the rest of this section, let $\mathbf{W}$ be factor-2 block Lewis weight overestimates for $[\mathbf{A}|\boldsymbol{b}]$. As in Line 1 of Algorithm 1 and from the corresponding guarantee given in (Manoj & Ovsiankin, 2025,

Lemmas 5.6, 5.8), this means that within $2 \log m$ linear-system-solves in $\mathbf{A}^\top \mathbf{D} \mathbf{A}$ for diagonal $\mathbf{D}$, we can find $\mathbf{W}$ such that for all $\boldsymbol{x} \in \mathbb{R}^d$ and $c \in \mathbb{R}$ we have

$$\|\mathbf{A}\boldsymbol{x} - c\boldsymbol{b}\|_{\mathcal{G}_\infty} \leq \left\|\mathbf{W}^{1/2}\mathbf{A}\boldsymbol{x} - c\mathbf{W}^{1/2}\boldsymbol{b}\right\|_2 \leq \sqrt{2(\operatorname{rank}(\mathbf{A}) + 1)} \|\mathbf{A}\boldsymbol{x} - c\boldsymbol{b}\|_{\mathcal{G}_\infty}.$$

Note that choosing $c = 1$ yields our original objective on either side of the above inequality. Motivated by the above, it is natural to use the norm given by $\mathbf{M} := \mathbf{A}^\top \mathbf{W} \mathbf{A}$ to give the geometry for the ball optimization oracle and for the analysis. Additionally, without loss of generality and for the sake of the analysis, let us rescale the problem so that

$$1 = \mathsf{OPT} := \|\mathbf{A}\boldsymbol{x}^\star - \boldsymbol{b}\|_{\mathcal{G}_\infty}.$$

Also, as mentioned earlier, assume without loss of generality that $\operatorname{rank}(\mathbf{A}) = d$.

We begin with Lemma C.9, which bounds our initial suboptimality in $\widetilde{f}$ and in $\|\cdot\|_{\mathbf{M}}$.

**Lemma C.9.** *Let $\widetilde{\boldsymbol{x}}_{\beta,\delta} := \underset{\boldsymbol{x} \in \mathbb{R}^d}{\operatorname{argmin}} \, \widetilde{f}_{\beta,\delta}(\boldsymbol{x})$. Then,*

$$\|\widetilde{\boldsymbol{x}}_{\beta,\delta} - \boldsymbol{x}_0\|_{\mathbf{M}} \leq (2 + 2(\beta \log m + \delta))\sqrt{2(d+1)}$$
$$\widetilde{f}_{\beta,\delta}(\boldsymbol{x}_0) - \widetilde{f}_{\beta,\delta}(\widetilde{\boldsymbol{x}}_{\beta,\delta}) \leq \sqrt{2(d+1)} - 1 + 2(\beta \log m + \delta).$$

*Proof of Lemma C.9.* It is easy to check that

$$\boldsymbol{x}_0 := \left(\mathbf{A}^\top \mathbf{W} \mathbf{A}\right)^{-1} \mathbf{A}^\top \mathbf{W} \boldsymbol{b} = \underset{\boldsymbol{x} \in \mathbb{R}^d}{\operatorname{argmin}} \, \left\|\mathbf{W}^{1/2}\mathbf{A}\boldsymbol{x} - \mathbf{W}^{1/2}\boldsymbol{b}\right\|_2.$$

By Lemma C.1, for all $\boldsymbol{x} \in \mathbb{R}^d$,

$$\left|\widetilde{f}_{\beta,\delta}(\boldsymbol{x}) - \|\mathbf{A}\boldsymbol{x} - \boldsymbol{b}\|_{\mathcal{G}_\infty}\right| \leq \beta \log m + \delta,$$

implying

$$\left|\|\mathbf{A}\boldsymbol{x}^\star - \boldsymbol{b}\|_{\mathcal{G}_\infty} - \widetilde{f}_{\beta,\delta}(\widetilde{\boldsymbol{x}}_{\beta,\delta})\right| \leq \beta \log m + \delta.$$

Combining this with Theorem E.3, we get

$$1 \leq \|\mathbf{A}\boldsymbol{x}^\star - \boldsymbol{b}\|_{\mathcal{G}_\infty} \leq \|\mathbf{A}\boldsymbol{x}_0 - \boldsymbol{b}\|_{\mathcal{G}_\infty} \leq \left\|\mathbf{W}^{1/2}\mathbf{A}\boldsymbol{x}_0 - \mathbf{W}^{1/2}\boldsymbol{b}\right\|_2$$

and

$$\frac{\left\|\mathbf{W}^{1/2}\mathbf{A}\boldsymbol{x}_0 - \mathbf{W}^{1/2}\boldsymbol{b}\right\|_2}{\sqrt{2(d+1)}} \leq \frac{\left\|\mathbf{W}^{1/2}\mathbf{A}\boldsymbol{x}^\star - \mathbf{W}^{1/2}\boldsymbol{b}\right\|_2}{\sqrt{2(d+1)}} \leq \|\mathbf{A}\boldsymbol{x}^\star - \boldsymbol{b}\|_{\mathcal{G}_\infty} = 1.$$

Combining these gives

$$1 \leq \left\|\mathbf{W}^{1/2}\mathbf{A}\boldsymbol{x}_0 - \mathbf{W}^{1/2}\boldsymbol{b}\right\|_2 \leq \sqrt{2(d+1)}.$$

Additionally,

$$\left\|\mathbf{W}^{1/2}\mathbf{A}\widetilde{\boldsymbol{x}}_{\beta,\delta} - \mathbf{W}^{1/2}\boldsymbol{b}\right\|_2 \leq \sqrt{2(d+1)} \|\mathbf{A}\widetilde{\boldsymbol{x}}_{\beta,\delta} - \boldsymbol{b}\|_{\mathcal{G}_\infty}$$
$$\leq \sqrt{2(d+1)} \left(\widetilde{f}_{\beta,\delta}(\widetilde{\boldsymbol{x}}_{\beta,\delta}) + \beta \log m + \delta\right)$$
$$\leq \sqrt{2(d+1)} \left(\|\mathbf{A}\boldsymbol{x}^\star - \boldsymbol{b}\|_{\mathcal{G}_\infty} + 2(\beta \log m + \delta)\right)$$
$$= \sqrt{2(d+1)}(1 + 2(\beta \log m + \delta)).$$

Then,

$$\|\widetilde{\boldsymbol{x}} - \boldsymbol{x}_0\|_{\mathbf{M}} = \left\|\left(\mathbf{W}^{1/2}\mathbf{A}\widetilde{\boldsymbol{x}}_{\beta,\delta} - \mathbf{W}^{1/2}\boldsymbol{b}\right) - \left(\mathbf{W}^{1/2}\mathbf{A}\boldsymbol{x}_0 - \mathbf{W}^{1/2}\boldsymbol{b}\right)\right\|_2$$
$$\leq \left\|\mathbf{W}^{1/2}\mathbf{A}\widetilde{\boldsymbol{x}}_{\beta,\delta} - \mathbf{W}^{1/2}\boldsymbol{b}\right\|_2 + \left\|\mathbf{W}^{1/2}\mathbf{A}\boldsymbol{x}_0 - \mathbf{W}^{1/2}\boldsymbol{b}\right\|_2$$

$$\leq (2 + 2(\beta \log m + \delta)) \sqrt{2(d+1)},$$

and

$$
\begin{aligned}
\widetilde{f}_{\beta,\delta}(\boldsymbol{x}_0) - \widetilde{f}_{\beta,\delta}(\widetilde{\boldsymbol{x}}_{\beta,\delta}) &\leq \|\mathbf{A}\boldsymbol{x}_0 - \boldsymbol{b}\|_{\mathcal{G}_\infty} - \|\mathbf{A}\boldsymbol{x}^\star - \boldsymbol{b}\|_{\mathcal{G}_\infty} + 2(\beta \log m + \delta) \\
&\leq \left\| \mathbf{W}^{1/2}\mathbf{A}\boldsymbol{x}_0 - \mathbf{W}^{1/2}\boldsymbol{b} \right\|_2 - \mathsf{OPT} + 2(\beta \log m + \delta) \\
&\leq \sqrt{2(d+1)} - 1 + 2(\beta \log m + \delta).
\end{aligned}
$$

This completes the proof of Lemma C.9. $\qquad\square$

We are now ready to prove Theorem 1.

*Proof of Theorem 1.* Algorithm 1 optimizes the regularization of $\widetilde{f}$ given by

$$\widehat{f}(\boldsymbol{x}) := \widetilde{f}_{\beta,\delta}(\boldsymbol{x}) + \frac{\varepsilon}{110R^2} \left\| \mathbf{W}^{1/2}\mathbf{A}(\boldsymbol{x} - \boldsymbol{x}_0) \right\|_2^2,$$

where $R$ is such that $\|\boldsymbol{x}_0 - \widetilde{\boldsymbol{x}}_{\beta,\delta}\|_\mathbf{M} \leq R$. Let $\widehat{\boldsymbol{x}} := \underset{\boldsymbol{x} \in \mathbb{R}^d}{\operatorname{argmin}} \widehat{f}(\boldsymbol{x})$. Using (Carmon et al., 2020, Proof of Corollary 12), we know that for every iterate $\boldsymbol{x}$ of Algorithm 1,

$$\left| \widehat{f}(\boldsymbol{x}) - \widetilde{f}_{\beta,\delta}(\boldsymbol{x}) \right| \leq \frac{\varepsilon}{4}.$$

We now choose $\beta = \varepsilon/(4 \log m)$ and $\delta = \varepsilon/4$, so that $\widetilde{f}_{\beta,\delta}$ approximates $f$ up to error $\varepsilon/2$ on every point. Using Lemma C.9, this gives $R = (2 + \varepsilon)\sqrt{2(d+1)}$. It is therefore sufficient to optimize $\widehat{f}$ up to $\varepsilon/4$ additive error.

Next, using Lemma C.5 and (Carmon et al., 2020, Lemmas 11, 43), we have that $\widehat{f}$ is $(1/\nu, e)$-Hessian stable in $\|\cdot\|_\mathbf{M}$ for $\nu = \Omega(1/(\varepsilon \log m))$. We now invoke (Carmon et al., 2020, Theorem 9), which tells us that we can implement a $(C/\sqrt{d}, C/\varepsilon)$-ball optimization oracle for $f$ with $O\left(\log\left(\frac{d}{\varepsilon}\right)^2\right)$ linear-system-solves.

The next step is to turn the ball optimization oracle into a $\frac{1}{2}$-MS oracle (Definition B.1). Using (Carmon et al., 2020, Proposition 5), we get a ball oracle complexity of $O\left(\log\left(\frac{d}{\varepsilon}\right)\right)$ to implement the MS oracle. In total, our linear-system-solve complexity for implementing the MS oracle for iteration $t$ is $O\left(\log\left(\frac{d}{\varepsilon}\right)^3\right)$.

Finally, using (Carmon et al., 2020, Theorem 6), we get that Algorithm 1 has a Newton iteration complexity of

$$
O\left( \left( \frac{(1+\varepsilon)\sqrt{d}\log m}{\varepsilon} \right)^{2/3} \log\left( \frac{\sqrt{d}+\varepsilon}{\varepsilon} \right) \left( \log\left( \frac{(\log m/\varepsilon)d(1+(1+\varepsilon)\sqrt{d}\log m/\varepsilon)}{\varepsilon} \right) \right)^3 \right)
$$

$$
= O\left( \frac{d^{1/3}}{\varepsilon^{2/3}} \log\left( \frac{d\log m}{\varepsilon} \right)^{14/3} \right),
$$

as promised.

Next, we analyze what happens if we fall in the case where $\mathbf{W} = \mathbf{I}_m$. Here, by using the $\sqrt{m}$ distortion from approximating $\ell_\infty^m$ with $\ell_2^m$, we have for all $\boldsymbol{x} \in \mathbb{R}^d$,

$$\frac{\|\mathbf{A}\boldsymbol{x} - \boldsymbol{b}\|_2}{\sqrt{m}} \leq \|\mathbf{A}\boldsymbol{x} - \boldsymbol{b}\|_{\mathcal{G}_\infty} \leq \|\mathbf{A}\boldsymbol{x} - \boldsymbol{b}\|_2.$$

Using this and repeating the previous analysis with this choice of $\mathbf{M}$ gives us a rate of

$$O\left( \frac{m^{1/3}}{\varepsilon^{2/3}} \log\left( \frac{m\log m}{\varepsilon} \right)^{14/3} \right),$$

as required.

It remains to determine the form of the Newton steps. For this, it is sufficient to understand the Hessian of $\widehat{f}$. A straightforward calculation shows that it is of the form $\mathbf{A}^\top \mathbf{B} \mathbf{A}$ where $\mathbf{B}$ is a block-diagonal matrix where each block has size $|S_i| \times |S_i|$. Thus, each Newton step solves a linear system of the form $\mathbf{A}^\top \mathbf{B} \mathbf{A} \boldsymbol{z} = \boldsymbol{v}$.

Combining this with the iteration complexity guarantee to find $\mathbf{W}$ (see Theorem 2.3) completes the proof of Theorem 1. $\qquad\square$

## D  INTERPOLATING BETWEEN AVERAGE AND ROBUST LOSSES

In this section, we prove Theorem 2. As before, our proof follows the outline in Section 2. The main technical challenges are to establish a form of strong convexity for our objective $f$ and then to build a solver for the proximal problem equation 8.

The rest of this section is organized as follows. In Appendix D.1, we derive calculus facts about our objective $f$, including bounds on its Hessian and the promised strong convexity (particularly Lemma D.2 and the more general result it builds on, Lemma D.3). In Appendix D.2, we prove some facts about the iterates of Algorithm 3 when applied to our setting. In Appendix D.3, we more precisely define and analyze our solver for proximal sub-problems. This section is fairly technical and we give a more detailed outline there. Finally, in Appendix D.4, we assemble all these components and analyze Algorithm 5, thereby proving Theorem 2.

Throughout this analysis, we rescale the problem so that $f(\boldsymbol{x}^\star) = 1$. It is now sufficient to solve for an $\varepsilon$-additive error solution.

### D.1  CALCULUS FOR THE OBJECTIVE

In this section, we work out some calculus facts related to our objective $\|\mathbf{A}\boldsymbol{x} - \boldsymbol{b}\|_{\mathcal{G}_p}^p$. Throughout this discussion, let $f(\boldsymbol{x}) := \|\mathbf{A}\boldsymbol{x} - \boldsymbol{b}\|_{\mathcal{G}_p}^p$.

**Lemma D.1.** *For any $\boldsymbol{z} \in \mathbb{R}^d$, we have*

$$p \sum_{i=1}^m \|\mathbf{A}_{S_i}\boldsymbol{x} - \boldsymbol{b}_{S_i}\|_2^{p-2} \|\mathbf{A}_{S_i}\boldsymbol{z}\|_2^2 \le \boldsymbol{z}^\top \left( \nabla^2 f(\boldsymbol{x}) \right) \boldsymbol{z} \le p(p-1) \sum_{i=1}^m \|\mathbf{A}_{S_i}\boldsymbol{x} - \boldsymbol{b}_{S_i}\|_2^{p-2} \|\mathbf{A}_{S_i}\boldsymbol{z}\|_2^2.$$

*Proof of Lemma D.1.* Let us first calculate the derivative and hessian for $f(\cdot)$ using the chain rule and usual matrix differentiation rules:

$$f(\boldsymbol{x}) = \sum_{i=1}^m \|\mathbf{A}_{S_i}\boldsymbol{x} - \boldsymbol{b}_{S_i}\|_2^p \ ,$$

$$\nabla f(\boldsymbol{x}) = p \sum_{i=1}^m \|\mathbf{A}_{S_i}\boldsymbol{x} - \boldsymbol{b}_{S_i}\|_2^{p-2} \mathbf{A}_{S_i}^\top (\mathbf{A}_{S_i}\boldsymbol{x} - \boldsymbol{b}_{S_i}) \ , \tag{10}$$

$$\nabla^2 f(\boldsymbol{x}) = p \sum_{i=1}^m \|\mathbf{A}_{S_i}\boldsymbol{x} - \boldsymbol{b}_{S_i}\|_2^{p-2} \mathbf{A}_{S_i}^\top \mathbf{A}_{S_i}$$

$$+ p(p-2) \sum_{i=1}^m \|\mathbf{A}_{S_i}\boldsymbol{x} - \boldsymbol{b}_{S_i}\|_2^{p-4} \left( \mathbf{A}_{S_i}^\top (\mathbf{A}_{S_i}\boldsymbol{x} - \boldsymbol{b}_{S_i})(\mathbf{A}_{S_i}\boldsymbol{x} - \boldsymbol{b}_{S_i})^\top \mathbf{A}_{S_i} \right) \ . \tag{11}$$

Using this formula, we take the quadratic form with respect to a vector $\boldsymbol{z}$. By Cauchy-Schwarz, notice that

$$\boldsymbol{z}^\top \|\mathbf{A}_{S_i}\boldsymbol{x} - \boldsymbol{b}_{S_i}\|_2^{p-4} \left( \mathbf{A}_{S_i}^\top (\mathbf{A}_{S_i}\boldsymbol{x} - \boldsymbol{b}_{S_i})(\mathbf{A}_{S_i}\boldsymbol{x} - \boldsymbol{b}_{S_i})^\top \mathbf{A}_{S_i} \right) \boldsymbol{z}$$

$$= \|\mathbf{A}_{S_i}\boldsymbol{x} - \boldsymbol{b}_{S_i}\|_2^{p-4} \langle \mathbf{A}_{S_i}\boldsymbol{z}, \mathbf{A}_{S_i}\boldsymbol{x} - \boldsymbol{b}_{S_i} \rangle^2 \le \|\mathbf{A}_{S_i}\boldsymbol{x} - \boldsymbol{b}_{S_i}\|_2^{p-2} \|\mathbf{A}_{S_i}\boldsymbol{z}\|_2^2.$$

With that, we have

$$\boldsymbol{z}^\top \left( \nabla^2 f(\boldsymbol{x}) \right) \boldsymbol{z} \le p \sum_{i=1}^m \|\mathbf{A}_{S_i}\boldsymbol{x} - \boldsymbol{b}_{S_i}\|^{p-2} \|\mathbf{A}_{S_i}\boldsymbol{z}\|_2^2 + (p-2) \|\mathbf{A}_{S_i}\boldsymbol{x} - \boldsymbol{b}_{S_i}\|^{p-2} \|\mathbf{A}_{S_i}\boldsymbol{z}\|_2^2 \ ,$$

$$= p(p-1) \sum_{i=1}^{m} \|\mathbf{A}_{S_i} \boldsymbol{x} - \boldsymbol{b}_{S_i}\|_2^{p-2} \|\mathbf{A}_{S_i} \boldsymbol{z}\|_2^2 \quad . \tag{12}$$

For the lower bound, we use our calculation for $\nabla^2 f(\boldsymbol{x})$ to write

$$\boldsymbol{z}^\top \left( \nabla^2 f(\boldsymbol{x}) \right) \boldsymbol{z} \geq p \sum_{i=1}^{m} \|\mathbf{A}_{S_i} \boldsymbol{x} - \boldsymbol{b}_{S_i}\|_2^{p-2} \|\mathbf{A}_{S_i} \boldsymbol{z}\|_2^2,$$

completing the proof of Lemma D.1. $\qquad \square$

### D.1.1 Strong convexity of the objective

The main pair of results of this section are Lemma D.2 and Lemma D.3. We can think of Lemma D.2 as a form of strong convexity for our objective.

**Lemma D.2** (Strong convexity of $f$). *Let $f(\boldsymbol{x}) := \|\mathbf{A}\boldsymbol{x} - \boldsymbol{b}\|_{\mathcal{G}_p}^p$. For all $\boldsymbol{d} \in \mathbb{R}^d$, we have*

$$f(\boldsymbol{x} + \boldsymbol{d}) \geq f(\boldsymbol{x}) + \langle \nabla f(\boldsymbol{x}), \boldsymbol{d} \rangle + \frac{4}{2^p} \|\mathbf{A}\boldsymbol{d}\|_{\mathcal{G}_p}^p,$$

*and therefore*

$$\|\boldsymbol{x} - \boldsymbol{x}^\star\|_{\mathbf{M}} \leq 2^{3/2 - 3/p} d^{1/2 - 1/p} (f(\boldsymbol{x}) - f(\boldsymbol{x}^\star))^{1/p} \quad .$$

**Lemma D.3** (Strong convexity of $\|\boldsymbol{y}\|_2^p$). *Let $\boldsymbol{v} \in \mathbb{R}^k$ for $k \geq 1$. For any $\triangle \in \mathbb{R}^k$, we have*

$$\|\boldsymbol{v} + \triangle\|_2^p \geq \|\boldsymbol{v}\|_2^p + p \|\boldsymbol{v}\|_2^{p-2} \langle \boldsymbol{v}, \triangle \rangle + \frac{4}{2^p} \|\triangle\|_2^p \quad .$$

To motivate Lemma D.3, let us see how Lemma D.3 implies Lemma D.2.

*Proof of Lemma D.2.* Note that

$$\nabla f(\boldsymbol{x}) = \sum_{i=1}^{m} p \|\mathbf{A}_{S_i} \boldsymbol{x} - \boldsymbol{b}_{S_i}\|_2^{p-2} \mathbf{A}_{S_i}^\top (\mathbf{A}_{S_i} \boldsymbol{x} - \boldsymbol{b}_{S_i}) \quad .$$

This implies

$$\sum_{i=1}^{m} p \|\mathbf{A}_{S_i} \boldsymbol{x} - \boldsymbol{b}_{S_i}\|_2^{p-2} \langle \mathbf{A}_{S_i} \boldsymbol{x} - \boldsymbol{b}_{S_i}, \mathbf{A}_{S_i} \boldsymbol{d} \rangle = \langle \nabla f(\boldsymbol{x}), \boldsymbol{d} \rangle \quad .$$

Combining this and applying Lemma D.3 (which is a strong convexity lemma for $\|\cdot\|_2^p$ that we prove subsequently in this section), we get

$$f(\boldsymbol{x} + \boldsymbol{d}) = \|\mathbf{A}(\boldsymbol{x} + \boldsymbol{d}) - \boldsymbol{b}\|_{\mathcal{G}_p}^p = \|\mathbf{A}\boldsymbol{d} + (\mathbf{A}\boldsymbol{x} - \boldsymbol{b})\|_{\mathcal{G}_p}^p \quad ,$$

$$= \sum_{i=1}^{m} \|\mathbf{A}_{S_i} \boldsymbol{d} + (\mathbf{A}_{S_i} \boldsymbol{x} - \boldsymbol{b}_{S_i})\|_2^p \quad ,$$

$$\geq^{\text{(Lemma D.3)}} \sum_{i=1}^{m} \|\mathbf{A}_{S_i} \boldsymbol{x} - \boldsymbol{b}_{S_i}\|_2^p + p \|\mathbf{A}_{S_i} \boldsymbol{x} - \boldsymbol{b}_{S_i}\|_2^{p-2} \langle (\mathbf{A}_{S_i} \boldsymbol{x} - \boldsymbol{b}_{S_i}), \mathbf{A}_{S_i} \boldsymbol{d} \rangle + \frac{4}{2^p} \|\mathbf{A}_{S_i} \boldsymbol{d}\|_2^p \quad ,$$

$$= \sum_{i=1}^{m} \|\mathbf{A}_{S_i} \boldsymbol{x} - \boldsymbol{b}_{S_i}\|_2^p + \left\langle p \|\mathbf{A}_{S_i} \boldsymbol{x} - \boldsymbol{b}_{S_i}\|_2^{p-2} \mathbf{A}_{S_i}^\top (\mathbf{A}_{S_i} \boldsymbol{x} - \boldsymbol{b}_{S_i}), \boldsymbol{d} \right\rangle + \frac{4}{2^p} \|\mathbf{A}_{S_i} \boldsymbol{d}\|_2^p \quad ,$$

$$=^{\text{equation } 10} \|\mathbf{A}\boldsymbol{x} - \boldsymbol{b}\|_{\mathcal{G}_p}^p + \langle \nabla f(\boldsymbol{x}), \boldsymbol{d} \rangle + \frac{4}{2^p} \|\mathbf{A}\boldsymbol{d}\|_{\mathcal{G}_p}^p = f(\boldsymbol{x}) + \langle \nabla f(\boldsymbol{x}), \boldsymbol{d} \rangle + \frac{4}{2^p} \|\mathbf{A}\boldsymbol{d}\|_{\mathcal{G}_p}^p \quad .$$

We now take care of the second statement. Observe that at optimality, we have $\nabla f(\boldsymbol{x}^\star) = 0$. Plugging this in (replace $\boldsymbol{x}$ by $\boldsymbol{x}^\star$ and $\boldsymbol{d}$ by $\boldsymbol{x} - \boldsymbol{x}^\star$ above), rearranging, and taking $p$th roots gives

$$\|\mathbf{A}(\boldsymbol{x} - \boldsymbol{x}^\star)\|_{\mathcal{G}_p} \leq \left( \frac{4}{2^p} \right)^{-1/p} (f(\boldsymbol{x}) - f(\boldsymbol{x}^\star))^{1/p} = \frac{2}{4^{1/p}} (f(\boldsymbol{x}) - f(\boldsymbol{x}^\star))^{1/p} \quad .$$

Next, recall that by Theorem 2.3,

$$\|\boldsymbol{x} - \boldsymbol{x}^\star\|_{\mathbf{M}} = \left\| \mathbf{W}^{1/2 - 1/p} \mathbf{A}(\boldsymbol{x} - \boldsymbol{x}^\star) \right\|_2 \leq (2d)^{1/2 - 1/p} \|\mathbf{A}(\boldsymbol{x} - \boldsymbol{x}^\star)\|_{\mathcal{G}_p} \quad .$$

Stitching the inequalities together completes the proof of Lemma D.2. $\qquad \square$

In the rest of this subsection, we prove Lemma D.3. We begin with a few numerical inequalities.

**Lemma D.4.** *For $\alpha \leq -1/2$ and $p \geq 2$, $g(\alpha) := \frac{1+p\alpha}{(-(2\alpha+1))^{p/2}}$ is nonincreasing in $\alpha$.*

*Proof of Lemma D.4.* We first take the derivative of $g$ with respect to $\alpha$,

$$
\begin{aligned}
g'(\alpha) &= \frac{p(-(2\alpha+1))^{p/2} - \left((-2)\frac{p}{2}\left(-(2\alpha+1)\right)^{p/2-1}\right)(1+p\alpha)}{(-(2\alpha+1))^p} \quad , \\
&= \frac{p(-(2\alpha+1)^{p/2}) + p\left(-(2\alpha+1)\right)^{p/2-1}(1+p\alpha)}{(-(2\alpha+1))^p} \quad , \\
&= p \cdot \frac{(-(2\alpha+1)) + (1+p\alpha)}{(-(2\alpha+1))^{p/2+1}} \quad , \\
&= p \cdot \frac{(p-2)\alpha}{(-(2\alpha+1))^{p/2+1}} \leq 0 \quad ,
\end{aligned}
$$

where in the final inequality we used that $p \geq 2$ and $\alpha \leq -1/2$. This completes the proof of the lemma. $\qquad\square$

We also need the following lemma, which is similar to a result due to Adil et al. (Adil et al., 2019, Lemma 4.5). It amounts to proving Lemma D.3 when the dimension $k = 1$.

**Lemma D.5** (Case A. of Lemma D.6). *For any $\alpha \in \mathbb{R}$ and $p \geq 2$,*

$$
|1+\alpha|^p \geq 1 + p\alpha + \frac{4}{2^p}|\alpha|^p \quad .
$$

*Proof of Lemma D.5.* Note that the inequality is true when $p = 2$ and becomes an equality. We consider the case when $p > 2$ and use $h(\alpha)$ to denote the error function,

$$
h(\alpha) := |1+\alpha|^p - \left(1 + p\alpha + \frac{4}{2^p}|\alpha|^p\right) \quad .
$$

We aim to show $h(\alpha) \geq 0$ for all $\alpha \in \mathbb{R}$. Let us first write the derivatives of $h$.

$$
h'(\alpha) = p\left(|1+\alpha|^{p-2}(1+\alpha) - \left(1 + \frac{4}{2^p}|\alpha|^{p-2}\alpha\right)\right) \quad ,
$$

$$
h''(\alpha) = p(p-1)\left(|1+\alpha|^{p-2} - \frac{4}{2^p}|\alpha|^{p-2}\right) = p(p-1)\left(|1+\alpha|^{p-2} - \left|\frac{\alpha}{2}\right|^{p-2}\right) \quad .
$$

It is now easy to verify the following statements about $h$,

    I. $h'(-2) = h''(-2) = 0$ and $h''(\alpha) > 0$ for $\alpha < -2$, $\Rightarrow$ within the range $(-\infty, -2]$ the function $h$ is minimized at $-2$;

    II. $h'(-2) = 0$ and $h''(\alpha) \leq 0$ for $\alpha \in (-2, -2/3] \Rightarrow h'(\alpha) < 0$ in the range $(-2, -2/3]$, i.e., in that range the function $h$ is minimized at $-2/3$;

    III. $h'(-2/3) < 0 = h'(0)$ and $h''(\alpha) > 0$ for $\alpha > -2/3 \Rightarrow$ the function $h$ is decreasing in $(-2/3, 0)$ and increasing in $[0, \infty)$, i.e., within the range $(-2/3, \infty)$ the function $h$ is minimized at $0$.

As a result of the above observations, it is enough to check the inequality at the inputs $\alpha \in \{-2, -2/3, 0\}$. We have for $p > 2$,

$$
h(-2) = 1 - (1 - 2p + 4) = 2p - 4 > 0 \quad ,
$$

$$
h\left(-\frac{2}{3}\right) = \frac{1}{3^p} - \left(1 - \frac{2p}{3} + \frac{4}{2^p}\left|\frac{2}{3}\right|^p\right) = \frac{1}{3^p} - 1 + \frac{2p}{3} - \frac{4}{3^p} = -1 + \frac{2p}{3} - \frac{3}{3^p} > 0
$$

$$
h(0) = 1 - 1 = 0 \quad .
$$

This implies that $h(\alpha) \geq 0$ for all values of $\alpha$, concluding the proof of Lemma D.5. $\qquad\square$

Next, we prove a special case of Lemma D.3.

**Lemma D.6.** *For any $\alpha \in \mathbb{R}$, $\beta \geq 0$, and $p \geq 2$, we have*

$$\left((1+\alpha)^2 + \beta^2\right)^{p/2} \geq 1 + p\alpha + \frac{4}{2^p}\left(\alpha^2 + \beta^2\right)^{p/2} \ .$$

*Proof of Lemma D.6.* Let us study the difference of both sides of the inequality using the following function,

$$h(\alpha, \beta) := \left((1+\alpha)^2 + \beta^2\right)^{p/2} - \left(1 + p\alpha + \frac{4}{2^p}\left(\alpha^2 + \beta^2\right)^{p/2}\right) \ .$$

We want to show that for $\alpha \in \mathbb{R}$, $\beta \geq 0$, and $p \geq 2$, $h(\alpha, \beta) \geq 0$. We will break this proof into three cases: **A.** $\alpha \in \mathbb{R}$ and $\beta = 0$; **B.** $\alpha \in (-\infty, -2] \cup [-2/3, \infty)$ and $\beta > 0$; and **C.** $\alpha \in (-2, -2/3)$ and $\beta > 0$. These cases together cover of the entire range of $\alpha \in \mathbb{R}$ and $\beta \geq 0$.

**Case A.** When $\beta = 0$, the proof simply follows from the statement of Lemma D.5 by noting $|\alpha|^p = (\sqrt{\alpha^2})^p = (\alpha^2)^{p/2}$.

In the remaining two cases we will show that for any $\alpha \in \mathbb{R}$, increasing the value of $\beta$ still maintains $h(\alpha, \beta) \geq 0$. To see this, we first note that the derivative of $h(\alpha, \beta)$ w.r.t. $\beta$ is given by,

$$\nabla_\beta h(\alpha, \beta) = p\beta \left(\left((1+\alpha)^2 + \beta^2\right)^{p/2-1} - \frac{4}{2^p}\left(\alpha^2 + \beta^2\right)^{p/2-1}\right) \ .$$

For $\beta > 0$, ensuring this derivative is positive is equivalent to the following,

$$\nabla_\beta h(\alpha, \beta) > 0 \equiv p\beta\left((1+\alpha)^2 + \beta^2\right)^{p/2-1} > p\beta \cdot \frac{4}{2^p}\left(\alpha^2 + \beta^2\right)^{p/2-1} \ ,$$

$$\equiv^{(p\beta > 0)} (1+\alpha)^2 + \beta^2 > \left(\frac{1}{2^{p-2}}\right)^{2/(p-2)} \cdot \left(\alpha^2 + \beta^2\right) \ ,$$

$$\equiv (1+\alpha)^2 + \beta^2 > \frac{1}{4} \cdot \left(\alpha^2 + \beta^2\right) \ ,$$

$$\equiv (3\alpha^2 + 8\alpha + 4) + 3\beta^2 > 0 \ ,$$

$$\equiv \beta^2 > -\left(\alpha^2 + \frac{8}{3}\alpha + \frac{4}{3}\right) \ . \tag{13}$$

**Case B.** Note that the roots of the quadratic function $3\alpha^2 + 8\alpha + 4$ are given by $\alpha_1 = -2$ and $\alpha_2 = -2/3$. This means that for $\alpha \in (-\infty, -2] \cup [-2/3, \infty)$ we have $3\alpha^2 + 8\alpha + 4 \geq 0$ which is **sufficient** to ensure using equation 13 that $\nabla_\beta h(\alpha, \beta) > 0$, and hence $h(\alpha, \beta) > 0$. This takes care of Case B.

**Case C.** Now we only need to consider the range $\alpha \in (-2, -2/3)$ with $\beta > 0$. In this range, the recall the equivalence equation 13,

$$\nabla_\beta h(\alpha, \beta) > 0 \equiv \beta > \sqrt{-\left(\alpha^2 + \frac{8}{3}\alpha + \frac{4}{3}\right)} =: \beta_0(\alpha) \ .$$

Thus for all $\beta > \beta_0(\alpha)$ we know that $h(\alpha, \beta)$ is increasing in $\beta$ and vice-versa. This allows us for any given $\alpha \in (-2, -2/3)$ to further break Case C into two sub-cases:

**Case C.I** For $\beta \in [0, \beta_0)$, since $h(\alpha, \beta)$ is decreasing in $\beta$ its lowest value is attained at $\beta = 0$ and we only need to verify that $h(\alpha, 0) \geq 0$. We get this directly from Lemma D.5.

**Case C.II** For $\beta \in [\beta_0, \infty)$, since $h(\alpha, \beta)$ is increasing in $\beta$ its lowest value is attained at $\beta = \beta_0$ and we only need to verify that $h(\alpha, \beta_0(\alpha)) \geq 0$. We first simplify the expression for $h(\alpha, \beta_0(\alpha))$,

$$h(\alpha, \beta_0(\alpha)) = \left((1+\alpha)^2 + \beta_0^2\right)^{p/2} - \left(1 + p\alpha + K_p\left(\alpha^2 + \beta_0^2\right)^{p/2}\right) \ ,$$

$$= \left(-\frac{1}{3} - \frac{2}{3}\alpha\right)^{p/2} - \left(1 + p\alpha + \frac{4}{2^p}\left(-\frac{8}{3}\alpha - \frac{4}{3}\right)^{p/2}\right) \ ,$$

$$= \left(-\frac{1}{3} - \frac{2}{3}\alpha\right)^{p/2} - \left(1 + p\alpha + 4\left(-\frac{2}{3}\alpha - \frac{1}{3}\right)^{p/2}\right) ,$$

$$= -1 - p\alpha - 3\left(-\frac{2}{3}\alpha - \frac{1}{3}\right)^{p/2} ,$$

$$= -1 - p\alpha - \frac{1}{3^{p/2-1}}(-2\alpha - 1)^{p/2} ,$$

$$= -(-2\alpha - 1)^{p/2}\left(\frac{1 + p\alpha}{(-2\alpha - 1)^{p/2}} + \frac{1}{3^{p/2-1}}\right) .$$

Now since $\alpha \in (-2, -2/3) < -1/2$ we can use Lemma D.4 to note that the first term is non-decreasing in $\alpha$ which means that its lowest value in this range can be lower bounded by its value at $\alpha = -2$, i.e., for $\alpha \in (-2, -2/3)$,

$$h(\alpha, \beta_0(\alpha)) \geq h(-2, \beta_0(-2)) ,$$

$$= -3^{p/2}\left(\frac{1 - 2p}{3^{p/2}} + \frac{1}{3^{p/2-1}}\right) ,$$

$$= 2p - 1 - 3 = 2(p - 2) > 0 ,$$

which finishes the proof of Case C.II and also Case C. Together Cases A, B and C complete the proof of Lemma D.6. $\square$

We are now ready to prove Lemma D.3.

*Proof of Lemma D.3.* First, assume that $\|\boldsymbol{v}\|_2 = 1$. We will later extend the result to all $\boldsymbol{v}$.

Since $\|\boldsymbol{v}\|_2 = 1$, we can write $\triangle = \alpha\boldsymbol{v} + \beta\boldsymbol{w}$ where $\langle\boldsymbol{v}, \boldsymbol{w}\rangle = 0$ and $\|\boldsymbol{w}\|_2 = 1$, so that we have $\|\triangle\|_2^2 = \alpha^2 + \beta^2$. Without loss of generality, we have $\beta \geq 0$. Fixing $\boldsymbol{w}$ and $\alpha$ for now, it is enough to show that for all $\beta \geq 0$, we have

$$\|(1 + \alpha)\boldsymbol{v} + \beta\boldsymbol{w}\|_2^p = \left((1 + \alpha)^2 + \beta^2\right)^{p/2} \overset{?}{\geq} 1 + p\alpha + \frac{4}{2^p}\|\triangle\|_2^p = 1 + p\alpha + \frac{4}{2^p}\left(\alpha^2 + \beta^2\right)^{p/2}.$$

This follows immediately by Lemma D.6.

We now extend the result for all $\boldsymbol{v}$. Let $\overline{\boldsymbol{v}} := \boldsymbol{v}/\|\boldsymbol{v}\|_2$ and note that

$$\|\boldsymbol{v} + \triangle\|_2^p = \|\boldsymbol{v}\|_2^p\left\|\overline{\boldsymbol{v}} + \frac{\triangle}{\|\boldsymbol{v}\|_2}\right\|_2^p \geq \|\boldsymbol{v}\|_2^p\left(1 + \left\langle\overline{\boldsymbol{v}}, \frac{\triangle}{\|\boldsymbol{v}\|_2}\right\rangle + \frac{4}{2^p}\left\|\frac{\triangle}{\|\boldsymbol{v}\|_2}\right\|_2^p\right)$$

$$= \|\boldsymbol{v}\|_2^p + p\|\boldsymbol{v}\|_2^{p-2}\langle\boldsymbol{v}, \triangle\rangle + \frac{4}{2^p}\|\triangle\|_2^p,$$

completing the proof of Lemma D.3. $\square$

### D.1.2 SMOOTHNESS OF THE OBJECTIVE

The main result of this subsection is Lemma D.7.

**Lemma D.7.** *For all $\boldsymbol{x} \in \mathbb{R}^d$, we have*

$$f(\boldsymbol{x}) - f(\boldsymbol{x}^\star) \leq \frac{p(p - 1)}{2}f(\boldsymbol{x})^{1 - \frac{2}{p}}\|\mathbf{A}(\boldsymbol{x} - \boldsymbol{x}^\star)\|_{\mathcal{G}_p}^2 .$$

*Proof of Lemma D.7.* By Taylor's/mean-value theorem, we can write for some $\boldsymbol{y}$ on the line connecting $\boldsymbol{x}^\star$ and $\boldsymbol{x}$,

$$f(\boldsymbol{x}) = f(\boldsymbol{x}^\star) + \langle\nabla f(\boldsymbol{x}^\star), \boldsymbol{x} - \boldsymbol{x}^\star\rangle + \frac{1}{2}(\boldsymbol{x} - \boldsymbol{x}^\star)^\top\nabla^2 f(\boldsymbol{y})(\boldsymbol{x} - \boldsymbol{x}^\star)$$

$$\overset{equation\ 12}{\leq} f(\boldsymbol{x}^\star) + \frac{p(p - 1)}{2}\sum_{i=1}^m\|\mathbf{A}_{S_i}\boldsymbol{y} - \boldsymbol{b}_{S_i}\|_2^{p-2}\|\mathbf{A}_{S_i}(\boldsymbol{x} - \boldsymbol{x}^\star)\|_2^2$$

$$\leq f(\boldsymbol{x}^\star) + \frac{p(p-1)}{2} \left( \sum_{i=1}^{m} \|\mathbf{A}_{S_i} \boldsymbol{y} - \boldsymbol{b}_{S_i}\|_2^p \right)^{\frac{p-2}{p}} \left( \sum_{i=1}^{m} \|\mathbf{A}_{S_i}(\boldsymbol{x} - \boldsymbol{x}^\star)\|_2^p \right)^{\frac{2}{p}}$$

$$\leq f(\boldsymbol{x}^\star) + \frac{p(p-1)}{2} f(\boldsymbol{x})^{1-\frac{2}{p}} \|\mathbf{A}(\boldsymbol{x} - \boldsymbol{x}^\star)\|_{\mathcal{G}_p}^2 \,,$$

completing the proof of Lemma D.7. $\qquad\square$

## D.2 FACTS ABOUT THE ITERATES

The main result of this section is Lemma D.8. In words, Lemma D.8 tells us that each proximal query we make in Algorithm 3 (see Line 7 of Algorithm 3) has bounded objective value. We will need this later when we argue about the convergence rates for the algorithms used to solve the proximal subproblems.

**Lemma D.8.** *For all queries $\boldsymbol{q}_t$, we have*

$$f(\boldsymbol{q}_t) \leq f(\boldsymbol{x}_t) + (9p(p-1))^{\frac{p}{2}} d^{\frac{p}{2}-1}.$$

*Proof of Lemma D.8.* We establish the following upper bound on $f(\boldsymbol{v}_t) - f(\boldsymbol{x}^\star)$ using the ingredients developed so far:

$$f(\boldsymbol{v}_t) - f(\boldsymbol{x}^\star) \leq \frac{p(p-1)}{2} f(\boldsymbol{v}_t)^{1-\frac{2}{p}} \|\mathbf{A}(\boldsymbol{v}_t - \boldsymbol{x}^\star)\|_{\mathcal{G}_p}^2 \qquad \text{(Lemma D.7)}$$

$$\leq \frac{p(p-1)}{2} f(\boldsymbol{v}_t)^{1-\frac{2}{p}} \|\boldsymbol{v}_t - \boldsymbol{x}^\star\|_{\mathbf{M}}^2 \qquad \text{(Theorem 2.3)}$$

$$\leq p(p-1) f(\boldsymbol{v}_t)^{1-\frac{2}{p}} \|\boldsymbol{x}_0 - \boldsymbol{x}^\star\|_{\mathbf{M}}^2 \qquad \text{(Lemma B.5)}$$

$$\leq p(p-1) f(\boldsymbol{v}_t)^{1-\frac{2}{p}} 2^2 (2d)^{1-\frac{2}{p}} \qquad \text{(Lemma E.5)}$$

$$\leq 8 d^{1-\frac{2}{p}} p(p-1) f(\boldsymbol{v}_t)^{1-\frac{2}{p}} \,.$$

Now, recall that we assume by rescaling that $f(\boldsymbol{x}^\star) = 1$. From this, it trivially follows that $1 \leq d^{1-\frac{2}{p}} p(p-1) f(\boldsymbol{v}_t)^{1-\frac{2}{p}}$. Combining these and re-arranging the above inequality leads to the following polynomial inequality in $f(\boldsymbol{v}_t)$,

$$0 \geq f(\boldsymbol{v}_t) - 8 d^{1-\frac{2}{p}} p(p-1) f(\boldsymbol{v}_t)^{1-\frac{2}{p}} - 1 \,,$$

$$= f(\boldsymbol{v}_t) - 9 d^{1-\frac{2}{p}} p(p-1) f(\boldsymbol{v}_t)^{1-\frac{2}{p}} + d^{1-\frac{2}{p}} p(p-1) f(\boldsymbol{v}_t)^{1-\frac{2}{p}} - 1 \,,$$

$$\geq f(\boldsymbol{v}_t) - 9 d^{1-\frac{2}{p}} p(p-1) f(\boldsymbol{v}_t)^{1-\frac{2}{p}} \,, \tag{14}$$

where in the last inequality we used the fact that the optimal value $f(\boldsymbol{x}^\star) = 1$ (due to our rescaling), which implies that for $p \geq 2$,

$$1 \leq f(\boldsymbol{v}_t) \leq d^{1-\frac{2}{p}} p(p-1) f(\boldsymbol{v}_t)^{1-\frac{2}{p}} \,.$$

Solving for $f(\boldsymbol{v}_t)$ in equation 14, we get

$$f(\boldsymbol{v}_t) \leq (9p(p-1))^{\frac{p}{2}} d^{\frac{p}{2}-1} \,.$$

Using the definition of $\boldsymbol{q}_t$ from Algorithm 3 (Line 6) along with the convexity of $f$ (Jensen's inequality), and using our bound on $f(\boldsymbol{v}_t)$ we note that,

$$f(\boldsymbol{q}_t) \leq f(\boldsymbol{x}_t) + f(\boldsymbol{v}_t) \,,$$

$$\leq f(\boldsymbol{x}_t) + (9p(p-1))^{\frac{p}{2}} d^{\frac{p}{2}-1} \,,$$

which completes the proof of Lemma D.8. $\qquad\square$

## D.3 Proximal subproblems – calculus, algorithms, proofs

Let

$$f_{\boldsymbol{q}_t}(\widetilde{\boldsymbol{x}}) := f(\widetilde{\boldsymbol{x}}) + ep^p \left\| \widetilde{\boldsymbol{x}} - \boldsymbol{q}_t \right\|_{\mathbf{M}}^p \quad .$$

In this subsection, we design and analyze an algorithm (Algorithm 4) that approximately solves the subproblem

$$\underset{\widetilde{\boldsymbol{x}} \in \mathbb{R}^d}{\operatorname{argmin}} \ f_{\boldsymbol{q}_t}(\widetilde{\boldsymbol{x}}).$$

Specifically, we will output $(\widetilde{\boldsymbol{x}}_{t+1}, \lambda_{t+1})$ that satisfy the $\frac{1}{2}$-MS oracle condition (Definition B.1) and an appropriate movement bound (Definition B.2).

This subproblem is the workhorse of Algorithm 5, and once we implement and analyze the solver, it is very straightforward to plug this into Algorithm 3 and Theorem B.3 to get our final iteration complexity.

---

**Algorithm 4** GpRegressionProxOracle: Implements $\frac{1}{2}$-MS oracle for $\|\cdot\|_{\mathcal{G}_p}$ regression (see Lemma D.20 and Algorithm 2.

---

**Require:** Query $\boldsymbol{q}_t$, previous iterate $\boldsymbol{x}_t$, intended parameter distance $\gamma$.
1: Define
$$f_{\boldsymbol{q}_t}(\widetilde{\boldsymbol{x}}) := f(\widetilde{\boldsymbol{x}}) + ep^p \left\| \widetilde{\boldsymbol{x}} - \boldsymbol{q}_t \right\|_{\mathbf{M}}^p$$
$$h_{\boldsymbol{q}_t}(\widetilde{\boldsymbol{x}}) := \left\| \widetilde{\boldsymbol{x}} - \boldsymbol{q}_t \right\|_{\nabla^2 f(\boldsymbol{q}_t)}^2 + ep^p \left\| \widetilde{\boldsymbol{x}} - \boldsymbol{q}_t \right\|_{\mathbf{M}}^p$$
$$D_{h_{\boldsymbol{q}_t}}(\boldsymbol{x}, \boldsymbol{y}) := h_{\boldsymbol{q}_t}(\boldsymbol{x}) - h_{\boldsymbol{q}_t}(\boldsymbol{y}) - \langle \nabla h_{\boldsymbol{q}_t}(\boldsymbol{y}), \boldsymbol{x} - \boldsymbol{y} \rangle^{\cdot}$$
$$\widetilde{\boldsymbol{x}}_{\boldsymbol{q}_t} := \underset{\widetilde{\boldsymbol{x}} \in \mathbb{R}^d}{\operatorname{argmin}} \ f_{\boldsymbol{q}_t}(\widetilde{\boldsymbol{x}})$$
2: Let $T \geq Cp^{O(1)} e \log \left( dp e h_{\boldsymbol{q}_t}(\widetilde{\boldsymbol{x}}_{\boldsymbol{q}_t}) \left( \frac{4}{p\gamma} \right)^p \right).$
3: Run Algorithm 2 with input iteration count $T$, base function $f_{\boldsymbol{q}_t}$, reference function $h_{\boldsymbol{q}_t}$, and initialization $\boldsymbol{q}_t$.

---

The goal of the rest of this section is to analyze Algorithm 4. The analysis follows several steps:

1. We find a reference function $h_{\boldsymbol{q}_t}$ that depends on the query point $\boldsymbol{q}_t$ for which the proximal objective $f_{\boldsymbol{q}_t}$ is relatively smooth and relatively strongly convex with $O(p^{O(1)})$ condition number (see Appendix A for a sense of why this is useful). The main result here is Lemma D.9.

2. We show that $f_{\boldsymbol{q}_t}$ is strongly convex, following from Lemma D.3. This will help us understand the argument suboptimality for any point that approximately optimizes $f_{\boldsymbol{q}_t}$ in function value. We also show that the reference function $h_{\boldsymbol{q}_t}$ is strongly convex, using the same tools, for the same reason.

3. We show a form of smoothness for $f_{\boldsymbol{q}_t}$. This helps us bound the gradient of any point that approximately optimizes $f_{\boldsymbol{q}_t}$. Combining these later will tell us that an approximate solution to $f_{\boldsymbol{q}_t}$ in argument value is also an approximate stationary point, i.e., it satisfies the $\frac{1}{2}$-MS condition (Definition B.1).

4. We solve the proximal subproblems. This solution itself follows a few steps:

   (a) We apply Theorem A.1. This tells us that as long as we can approximately solve the Bregman proximal problems (approximately implementing Line 3 in Algorithm 2), we will be in good shape.

   (b) This means we have to figure out how to approximately solve problems of the form $\underset{\boldsymbol{x} \in \mathbb{R}^d}{\operatorname{argmin}} \ \langle \boldsymbol{g}, \boldsymbol{x} \rangle + L h_{\boldsymbol{q}_t}(\boldsymbol{x})$, where $L$ is the smoothness constant derived for $f_{\boldsymbol{q}_t}$ with respect to $h_{\boldsymbol{q}_t}$. We do this up to an accuracy that approximate mirror descent can handle (see Theorem A.1 for details on what we want this approximation to look like). For the approximation to work, we need to approximately solve this problem up

to both argument accuracy and approximate stationarity. The main technical result of interest here is Lemma D.18.

5. We use the smoothness and strong convexity guarantees to show that our solution from the previous step satisfies the $\frac{1}{2}$-MS oracle (Definition B.1), which means we can plug-and-play into Theorem B.3.

### D.3.1 HESSIAN STABILITY

Throughout this section, we adopt the following notation:

$$C_p := ep^p$$

$$f(\boldsymbol{x}) := \sum_{i=1}^{m} \|\mathbf{A}_{S_i}\boldsymbol{x} - \boldsymbol{b}_{S_i}\|_2^p$$

$$f_{\boldsymbol{q}}(\boldsymbol{x}) := f(\boldsymbol{x}) + C_p \|\boldsymbol{x} - \boldsymbol{q}\|_{\mathbf{M}}^p$$

$$h_{\boldsymbol{q}}(\boldsymbol{x}) := \|\boldsymbol{x} - \boldsymbol{q}\|_{\nabla^2 f(\boldsymbol{q})}^2 + C_p \|\boldsymbol{x} - \boldsymbol{q}\|_{\mathbf{M}}^p$$

We begin with proving our Hessian stability fact, which should also be equivalently viewed as showing that $f_{\boldsymbol{q}_t}$ is relatively smooth and relatively strongly convex in $h_{\boldsymbol{q}_t}$ with $O(p^{O(1)})$ condition number. Our main result is Lemma D.9 which relies on analytical results Lemma D.10 and Lemma D.11 that we prove later.

**Lemma D.9.** *For all $\boldsymbol{x} \in \mathbb{R}^d$ and $p \geq 2$, we have*

$$\frac{1}{2p \cdot e} \nabla^2 h_{\boldsymbol{q}}(\boldsymbol{x}) \preceq \nabla^2 f_{\boldsymbol{q}}(\boldsymbol{x}) \preceq p \cdot e \nabla^2 h_{\boldsymbol{q}}(\boldsymbol{x}) \ .$$

*Proof of Lemma D.9.* Using an arbitrary $\boldsymbol{z} \in \mathbb{R}^d$ we can write the following quadratic form of the hessian of $f$,

$$\boldsymbol{z}^\top \nabla^2 f(\boldsymbol{x})\boldsymbol{z} \leq^{(a)} p \cdot (p-1) \sum_{i=1}^{m} \|\mathbf{A}_{S_i}\boldsymbol{x} - \boldsymbol{b}_{S_i}\|_2^{p-2} \|\mathbf{A}_{S_i}\boldsymbol{z}\|_2^2 \ ,$$

$$= p \cdot (p-1) \sum_{i=1}^{m} \|\mathbf{A}_{S_i}(\boldsymbol{x} - \boldsymbol{q}) + \mathbf{A}_{S_i}\boldsymbol{q} - \boldsymbol{b}_{S_i}\|_2^{p-2} \|\mathbf{A}_{S_i}\boldsymbol{z}\|_2^2 \ ,$$

$$\leq^{(b)} p \cdot (p-1) \sum_{i=1}^{m} \left( \alpha_p^{p-2} \|\mathbf{A}_{S_i}(\boldsymbol{x} - \boldsymbol{q})\|_2^{p-2} \|\mathbf{A}_{S_i}\boldsymbol{z}\|_2^2 + \beta_p^{p-2} \|\mathbf{A}_{S_i}\boldsymbol{q} - \boldsymbol{b}_{S_i}\|_2^{p-2} \|\mathbf{A}_{S_i}\boldsymbol{z}\|_2^2 \right) \ ,$$

$$\leq^{(c)} p \cdot (p-1) \cdot \alpha_p^{p-2} \sum_{i=1}^{m} \|\mathbf{A}_{S_i}(\boldsymbol{x} - \boldsymbol{q})\|_2^{p-2} \|\mathbf{A}_{S_i}\boldsymbol{z}\|_2^2 + (p-1) \cdot \beta_p^{p-2} \boldsymbol{z}^\top \nabla^2 f(\boldsymbol{q})\boldsymbol{z} \ ,$$

$$\leq^{(d)} p \cdot (p-1) \cdot \alpha_p^{p-2} \left( \|\boldsymbol{x} - \boldsymbol{q}\|_{\mathbf{M}}^p \right)^{(p-2)/p} \left( \|\boldsymbol{z}\|_{\mathbf{M}}^p \right)^{2/p} + (p-1) \cdot \beta_p^{p-2} \boldsymbol{z}^\top \nabla^2 f(\boldsymbol{q})\boldsymbol{z} \ ,$$

$$= p \cdot (p-1) \cdot \alpha_p^{p-2} \|\boldsymbol{x} - \boldsymbol{q}\|_{\mathbf{M}}^{p-2} \|\boldsymbol{z}\|_{\mathbf{M}}^2 + (p-1) \cdot \beta_p^{p-2} \boldsymbol{z}^\top \nabla^2 f(\boldsymbol{q})\boldsymbol{z} \ ,$$

$$\leq^{(e)} \frac{(p-1) \cdot \alpha_p^{p-2}}{C_p} \boldsymbol{z}^\top \nabla^2 g_{\boldsymbol{q}}(\boldsymbol{x})\boldsymbol{z} + (p-1) \cdot \beta_p^{p-2} \boldsymbol{z}^\top \nabla^2 f(\boldsymbol{q})\boldsymbol{z} \ , \tag{15}$$

where in (a) we apply the upper bound from Lemma D.1, in (b) we pick $\alpha_p, \beta_p \geq 1$ such that $1/\alpha_p + 1/\beta_p = 1$ (we will choose them later), in (c) we apply the lower bound from Lemma D.1, in (d) we use the choice of our weights in designing $\mathbf{M}$ and Theorem 2.3 and finally in (e) we use the following calculations for the regularizer term for some $\boldsymbol{z} \in \mathbb{R}^d$,

$$g_{\boldsymbol{q}}(\boldsymbol{x}) := C_p \|\boldsymbol{x} - \boldsymbol{q}\|_{\mathbf{M}}^p \ ,$$

$$\nabla g_{\boldsymbol{q}}(\boldsymbol{x}) = pC_p \|\boldsymbol{x} - \boldsymbol{q}\|_{\mathbf{M}}^{p-2} \mathbf{M}(\boldsymbol{x} - \boldsymbol{q}) \ ,$$

$$\nabla^2 g_{\boldsymbol{q}}(\boldsymbol{x}) = pC_p \|\boldsymbol{x} - \boldsymbol{q}\|_{\mathbf{M}}^{p-2} \mathbf{M} + p(p-2)C_p \|\boldsymbol{x} - \boldsymbol{q}\|_{\mathbf{M}}^{p-4} \mathbf{M}(\boldsymbol{x} - \boldsymbol{q})(\boldsymbol{x} - \boldsymbol{q})^\top \mathbf{M} \ ,$$

$$\boldsymbol{z}^\top \nabla^2 g_{\boldsymbol{q}}(\boldsymbol{x})\boldsymbol{z} = pC_p \|\boldsymbol{x} - \boldsymbol{q}\|_{\mathbf{M}}^{p-2} \|\boldsymbol{z}\|_{\mathbf{M}}^2 + p(p-2)C_p \|\boldsymbol{x} - \boldsymbol{q}\|_{\mathbf{M}}^{p-4} \left( (\boldsymbol{x} - \boldsymbol{q})^\top \mathbf{M}\boldsymbol{z} \right)^2 \geq^{(p \geq 2)} 0 \ .$$

Combining equation 15 with the definition of $f_{\boldsymbol{q}}$ gives us,

$$\boldsymbol{z}^\top \nabla^2 f_{\boldsymbol{q}}(\boldsymbol{x})\boldsymbol{z} = \boldsymbol{z}^\top \nabla^2 f(\boldsymbol{x})\boldsymbol{z} + \boldsymbol{z}^\top \nabla^2 g_{\boldsymbol{q}}(\boldsymbol{x})\boldsymbol{z} \ ,$$

$$\leq^{\text{using } equation \ 15} (p-1) \cdot \beta_p^{p-2} \boldsymbol{z}^\top \nabla^2 f(\boldsymbol{q}) \boldsymbol{z} + \left( 1 + \frac{(p-1) \cdot \alpha_p^{p-2}}{C_p} \right) \boldsymbol{z}^\top \nabla^2 g_{\boldsymbol{q}}(\boldsymbol{x}) \boldsymbol{z} \ .$$

Thus, in order to finish the proof for the upper bound we need to pick $\alpha_p, \beta_p$. We split the analysis here into two cases: **A.** $p > 2$ and **B.** $p = 2$.

**Case A.** ($p > 2$)   For simplicity we will just pick $\alpha_p = p - 1$ and $\beta_p = \frac{p-1}{p-2}$ which implies,

$$\boldsymbol{z}^\top \nabla^2 f_{\boldsymbol{q}}(\boldsymbol{x}) \boldsymbol{z} \leq (p-1) \cdot \left( 1 + \frac{1}{p-2} \right)^{p-2} \boldsymbol{z}^\top \nabla^2 f(\boldsymbol{q}) \boldsymbol{z} + \left( 1 + \frac{(p-1) \cdot (p-1)^{p-2}}{C_p} \right) \boldsymbol{z}^\top \nabla^2 g_{\boldsymbol{q}}(\boldsymbol{x}) \boldsymbol{z} \ ,$$

$$\leq (p-1) \cdot e \boldsymbol{z}^\top \nabla^2 f(\boldsymbol{q}) \boldsymbol{z} + \left( 1 + \frac{(p-1)^{p-1}}{C_p} \right) \boldsymbol{z}^\top \nabla^2 g_{\boldsymbol{q}}(\boldsymbol{x}) \boldsymbol{z} \ ,$$

$$= \frac{(p-1) \cdot e}{2} \boldsymbol{z}^\top \left( \nabla^2 h_{\boldsymbol{q}}(\boldsymbol{x}) - \nabla^2 g_{\boldsymbol{q}}(\boldsymbol{x}) \right) \boldsymbol{z} + \left( 1 + \frac{(p-1)^{p-1}}{C_p} \right) \boldsymbol{z}^\top \nabla^2 g_{\boldsymbol{q}}(\boldsymbol{x}) \boldsymbol{z} \ ,$$

$$\leq^{(p \geq 2)} p \cdot e \boldsymbol{z}^\top \nabla^2 h_{\boldsymbol{q}}(\boldsymbol{x}) \boldsymbol{z} + \left( 1 + \frac{(p-1)^{p-1}}{C_p} - \frac{(p-1) \cdot e}{2} \right) \boldsymbol{z}^\top \nabla^2 g_{\boldsymbol{q}}(\boldsymbol{x}) \boldsymbol{z} \ ,$$

$$= p \cdot e \boldsymbol{z}^\top \nabla^2 h_{\boldsymbol{q}}(\boldsymbol{x}) \boldsymbol{z} + \left( 1 + \frac{(p-1)^{p-1}}{ep^p} - \frac{(p-1) \cdot e}{2} \right) \boldsymbol{z}^\top \nabla^2 g_{\boldsymbol{q}}(\boldsymbol{x}) \boldsymbol{z} \ ,$$

$$\leq^{\text{(Lemma D.10)}} p \cdot e \boldsymbol{z}^\top \nabla^2 h_{\boldsymbol{q}}(\boldsymbol{x}) \boldsymbol{z} \ ,$$

where in the final inequality we use Lemma D.10 which tell us that for $p \geq 2$ the constant in front of $\boldsymbol{z}^\top \nabla^2 g_{\boldsymbol{q}}(\boldsymbol{x}) \boldsymbol{z}$ is negative along with the fact that $\boldsymbol{z}^\top \nabla^2 g_{\boldsymbol{q}}(\boldsymbol{x}) \boldsymbol{z}$ is non-negative. To get the lower bound we first exchange $\boldsymbol{x}, \boldsymbol{q}$ in equation 15 (and use the values of $\alpha_p$ and $\beta_p$) to get,

$$\boldsymbol{z}^\top \nabla^2 f(\boldsymbol{q}) \boldsymbol{z} \leq \frac{(p-1) \cdot (p-1)p - 2}{ep^p} \boldsymbol{z}^\top \nabla^2 g_{\boldsymbol{x}}(\boldsymbol{q}) \boldsymbol{z} + (p-1) \left( 1 + \frac{1}{p-2} \right)^{p-2} \boldsymbol{z}^\top \nabla^2 f(\boldsymbol{x}) \boldsymbol{z} \ ,$$

$$\Rightarrow \boldsymbol{z}^\top \nabla^2 f(\boldsymbol{q}) \boldsymbol{z} \leq \frac{(p-1)^{p-1}}{ep^p} \boldsymbol{z}^\top \nabla^2 g_{\boldsymbol{x}}(\boldsymbol{q}) \boldsymbol{z} + (p-1) e \boldsymbol{z}^\top \nabla^2 f(\boldsymbol{x}) \boldsymbol{z} \ ,$$

$$\Rightarrow \frac{1}{(p-1)e} \boldsymbol{z}^\top \nabla^2 f(\boldsymbol{q}) \boldsymbol{z} - \frac{(p-1)^{p-2}}{e^2 p^p} \boldsymbol{z}^\top \nabla^2 g_{\boldsymbol{x}}(\boldsymbol{q}) \boldsymbol{z} \leq \boldsymbol{z}^\top \nabla^2 f(\boldsymbol{x}) \boldsymbol{z} \ .$$

We can finally lower bound,

$$\boldsymbol{z}^\top \nabla^2 f_{\boldsymbol{q}}(\boldsymbol{x}) \boldsymbol{z} = \boldsymbol{z}^\top \nabla^2 f(\boldsymbol{x}) \boldsymbol{z} + \boldsymbol{z}^\top \nabla^2 g_{\boldsymbol{q}}(\boldsymbol{x}) \boldsymbol{z} \ ,$$

$$\geq \frac{1}{(p-1)e} \boldsymbol{z}^\top \nabla^2 f(\boldsymbol{q}) \boldsymbol{z} - \frac{(p-1)^{p-2}}{e^2 p^p} \boldsymbol{z}^\top \nabla^2 g_{\boldsymbol{x}}(\boldsymbol{q}) \boldsymbol{z} + \boldsymbol{z}^\top \nabla^2 g_{\boldsymbol{q}}(\boldsymbol{x}) \boldsymbol{z} \ ,$$

$$= \frac{1}{2(p-1)e} \boldsymbol{z}^\top \left( \nabla^2 h_{\boldsymbol{q}}(\boldsymbol{x}) - \nabla^2 g_{\boldsymbol{q}}(\boldsymbol{x}) \right) \boldsymbol{z} - \frac{(p-1)^{p-2}}{e^2 p^p} \boldsymbol{z}^\top \nabla^2 g_{\boldsymbol{x}}(\boldsymbol{q}) \boldsymbol{z} + \boldsymbol{z}^\top \nabla^2 g_{\boldsymbol{q}}(\boldsymbol{x}) \boldsymbol{z} \ ,$$

$$\geq^{(g_{\boldsymbol{q}}(\boldsymbol{x}) = g_{\boldsymbol{x}}(\boldsymbol{q}))} \frac{1}{2pe} \boldsymbol{z}^\top \nabla^2 h_{\boldsymbol{q}}(\boldsymbol{x}) \boldsymbol{z} + \left( 1 - \frac{1}{2(p-1)e} - \frac{(p-1)^{p-2}}{e^2 p^p} \right) \boldsymbol{z}^\top \nabla^2 g_{\boldsymbol{q}}(\boldsymbol{x}) \boldsymbol{z} \ ,$$

$$\geq^{\text{(Lemma D.11)}} \frac{1}{2pe} \boldsymbol{z}^\top \nabla^2 h_{\boldsymbol{q}}(\boldsymbol{x}) \boldsymbol{z} \ ,$$

where in the final inequality we use Lemma D.11 and the fact that $\boldsymbol{z}^\top \nabla^2 g_{\boldsymbol{q}}(\boldsymbol{x}) \boldsymbol{z}$ is non-negative. This finishes the proof for Case A.

We finally consider the corner case with $p = 2$.

**Case B.** ($p = 2$)   In this case the proof is trivial, and follows from simply writing the quadratic forms for $f_{\boldsymbol{q}}$ and $h_{\boldsymbol{q}}$. We do so below,

$$\boldsymbol{z}^\top \nabla^2 f_{\boldsymbol{q}}(\boldsymbol{x}) \boldsymbol{z} = \boldsymbol{z}^\top \nabla^2 f(\boldsymbol{x}) \boldsymbol{z} + \boldsymbol{z}^\top \nabla^2 g_{\boldsymbol{q}}(\boldsymbol{x}) \boldsymbol{z} \ ,$$

$$= \boldsymbol{z}^\top \nabla^2 f(\boldsymbol{x}) \boldsymbol{z} + 2C_2 \|\boldsymbol{z}\|_{\mathbf{M}}^2 \ ,$$

$$\leq 2\boldsymbol{z}^\top \nabla^2 f(\boldsymbol{x}) \boldsymbol{z} + 2C_2 \|\boldsymbol{z}\|_{\mathbf{M}}^2 = \boldsymbol{z}^\top \nabla^2 h_{\boldsymbol{q}}(\boldsymbol{x}) \boldsymbol{z} \ ,$$

which shows the relative smoothness with a constant of $1$ which is smaller (and hence better) than the claimed constant (for $p = 2$) of $2e$ in the lemma. Now for the relative strong convexity we do the same,

$$
\begin{aligned}
\boldsymbol{z}^\top \nabla^2 f_{\boldsymbol{q}}(\boldsymbol{x}) \boldsymbol{z} &= \boldsymbol{z}^\top \nabla^2 f(\boldsymbol{x}) \boldsymbol{z} + 2C_2 \|\boldsymbol{z}\|_{\mathbf{M}}^2 \;\;, \\
&\geq \frac{1}{2} \cdot \left( 2\boldsymbol{z}^\top \nabla^2 f(\boldsymbol{x}) \boldsymbol{z} + 2C_2 \|\boldsymbol{z}\|_{\mathbf{M}}^2 \right) \;\;, \\
&= \frac{1}{2} \boldsymbol{z}^\top \nabla^2 h_{\boldsymbol{q}}(\boldsymbol{x}) \boldsymbol{z} \;\;,
\end{aligned}
$$

which shows relative strong-convexity with a constant of $\frac{1}{2}$ which is larger (and hence better) than the claimed constant (for $p = 2$) of $\frac{1}{4e}$ in the lemma. This finishes the proof for Case B.

This completes the proof of Lemma D.9. $\qquad\square$

We prove two small technical lemmas that we used in the above proof now.

**Lemma D.10.** *For all $p \geq 2$, $g(p) = 1 + \frac{(p-1)^{p-1}}{ep^p} - \frac{(p-1) \cdot e}{2} \leq 0$.*

*Proof.* First note that at $p = 2$ the function takes a strictly negative value,

$$
g(2) = 1 + \frac{(1}{e2^2} - \frac{e}{2} = \frac{4e + 1 - 2e^2}{4e} < 0 \;\;.
$$

We will now show that the function is increasing in $p$ for $p \geq 2$,

$$
\begin{aligned}
g'(p) &= -\frac{(p-1)^{p-1} p^p (\ln(p) + 1)}{p^2 p} + \frac{(p-1)^{p-1}(\ln(p-1) + 1)}{p^p} - \frac{e}{2} \;\;, \\
&= -\frac{(p-1)^{p-1} \ln(p/(p-1))}{p^p} - \frac{e}{2} < 0 \;\;.
\end{aligned}
$$

Thus, the function attains its maximum value at $p = 2$ in the range $p \geq 2$, implying it is strictly negative in that range. $\qquad\square$

**Lemma D.11.** *For all $p \geq 2$, $g(p) = 1 - \frac{1}{2(p-1)e} - \frac{(p-1)^{p-2}}{e^2 p^p} \geq 0$.*

*Proof.* First note that at $p = 2$ the function takes a strictly positive value,

$$
g(2) = 1 - \frac{1}{2e} - \frac{1^0}{e^2 2^2} = 1 - \frac{1}{2e} - \frac{1}{4e^2} = \frac{4e^2 - 2e - 1}{4e^2} > 0 \;\;.
$$

We will now show that the function is increasing in $p$ for $p \geq 2$,

$$
\begin{aligned}
g'(p) &= \frac{1}{2(p-1)^2 e} + \frac{(p-1)^{p-2} p^p (\ln(p) + 1)}{e^2 p^{2p}} - \frac{(p-1)^{p-2}(\ln(p-1) + (p-2)/(p-1))}{e^2 p^p} \;\;, \\
&= \frac{1}{2(p-1)^2 e} + \frac{(p-1)^{p-2}(\ln(p) + 1)}{e^2 p^p} - \frac{(p-1)^{p-2}(\ln(p-1) + 1 - 1/(p-1))}{e^2 p^p} \;\;, \\
&= \frac{1}{2(p-1)^2 e} + \frac{(p-1)^{p-2}\left(\ln(p/(p-1)) + 1/(p-1)\right)}{e^2 p^p} > 0 \;\;.
\end{aligned}
$$

Thus, the function $g$ attains its minimum value at $p = 2$ in the range $p \geq 2$, implying that it is strictly positive in that range. $\qquad\square$

### D.3.2 STRONG CONVEXITY OF THE PROXIMAL OBJECTIVE AND FRIENDS

We begin with showing that the proximal objective enjoys a form of strong convexity.

**Lemma D.12.** *For all $\boldsymbol{x}, \boldsymbol{d} \in \mathbb{R}^d$, we have*

$$
f_{\boldsymbol{q}}(\boldsymbol{x} + \boldsymbol{d}) \geq f_{\boldsymbol{q}}(\boldsymbol{x}) + \langle \nabla f_{\boldsymbol{q}}(\boldsymbol{x}), \boldsymbol{d} \rangle + \frac{4}{2^p} \left( \|\mathbf{A}\boldsymbol{d}\|_{\mathcal{G}_p}^p + C_p \|\boldsymbol{d}\|_{\mathbf{M}}^p \right).
$$

*Proof of Lemma D.12.* Let $K_p := \frac{4}{2^p}$.

The plan is to apply Lemma D.3 to $f_{\boldsymbol{q}}(\boldsymbol{x} + \boldsymbol{d})$. We start with the regularizer. Notice that

$$
\begin{aligned}
\|\boldsymbol{x} + \boldsymbol{d} - \boldsymbol{q}\|_{\mathbf{M}}^p = \left\|\mathbf{M}^{1/2}(\boldsymbol{x} + \boldsymbol{d} - \boldsymbol{q})\right\|_2^p &= \left\|\mathbf{M}^{1/2}(\boldsymbol{x} - \boldsymbol{q}) + \mathbf{M}^{1/2}\boldsymbol{d}\right\|_2^p \ , \\
&\geq^{(\text{Lemma D.3})} \left\|\mathbf{M}^{1/2}(\boldsymbol{x} - \boldsymbol{q})\right\|_2^p \\
&\quad + \left\langle p\left\|\mathbf{M}^{1/2}(\boldsymbol{x} - \boldsymbol{q})\right\|_2^{p-2}\mathbf{M}^{1/2}(\boldsymbol{x} - \boldsymbol{q}), \mathbf{M}^{1/2}\boldsymbol{d}\right\rangle + K_p\left\|\mathbf{M}^{1/2}\boldsymbol{d}\right\|_2^p \ , \\
&= \|\boldsymbol{x} - \boldsymbol{q}\|_{\mathbf{M}}^p + \left\langle p\|\boldsymbol{x} - \boldsymbol{q}\|_{\mathbf{M}}^{p-2}\mathbf{M}(\boldsymbol{x} - \boldsymbol{q}), \boldsymbol{d}\right\rangle + K_p\|\boldsymbol{d}\|_{\mathbf{M}}^p \ , \\
&= \|\boldsymbol{x} - \boldsymbol{q}\|_{\mathbf{M}}^p + \left\langle \nabla_{\boldsymbol{x}}\left(\|\boldsymbol{x} - \boldsymbol{q}\|_{\mathbf{M}}^p\right), \boldsymbol{d}\right\rangle + K_p\|\boldsymbol{d}\|_{\mathbf{M}}^p \ .
\end{aligned}
$$

(16)

(17)

We combine this with the conclusion of Lemma D.2, giving

$$
\begin{aligned}
f_{\boldsymbol{q}}(\boldsymbol{x} + \boldsymbol{d}) &= f(\boldsymbol{x} + \boldsymbol{d}) + C_p\|\boldsymbol{x} + \boldsymbol{d} - \boldsymbol{q}\|_{\mathbf{M}}^p \ , \\
&\geq^{(\text{Lemma D.2})} f(\boldsymbol{x}) + \langle\nabla f(\boldsymbol{x}), \boldsymbol{d}\rangle + K_p\|\mathbf{A}\boldsymbol{d}\|_{\mathcal{G}_p}^p + C_p\|\boldsymbol{x} + \boldsymbol{d} - \boldsymbol{q}\|_{\mathbf{M}}^p \ , \\
&\geq^{\text{equation 17}} f(\boldsymbol{x}) + \langle\nabla f(\boldsymbol{x}), \boldsymbol{d}\rangle + K_p\|\mathbf{A}\boldsymbol{d}\|_{\mathcal{G}_p}^p + C_p\|\boldsymbol{x} - \boldsymbol{q}\|_{\mathbf{M}}^p \\
&\quad + C_p\left\langle\nabla_{\boldsymbol{x}}\left(\|\boldsymbol{x} - \boldsymbol{q}\|_{\mathbf{M}}^p\right), \boldsymbol{d}\right\rangle + K_p C_p\|\boldsymbol{d}\|_{\mathbf{M}}^p \ , \\
&= {\color{red}f(\boldsymbol{x}) + C_p\|\boldsymbol{x} - \boldsymbol{q}\|_{\mathbf{M}}^p} + \left\langle\nabla_{\boldsymbol{x}}\left({\color{red}f(\boldsymbol{x}) + C_p\|\boldsymbol{x} - \boldsymbol{q}\|_{\mathbf{M}}^p}\right), \boldsymbol{d}\right\rangle \\
&\quad + K_p\|\mathbf{A}\boldsymbol{d}\|_{\mathcal{G}_p}^p + K_p C_p\|\boldsymbol{d}\|_{\mathbf{M}}^p \ , \\
&= {\color{red}f_{\boldsymbol{q}}(\boldsymbol{x})} + \langle\nabla f_{\boldsymbol{q}}(\boldsymbol{x}), \boldsymbol{d}\rangle + K_p\left(\|\mathbf{A}\boldsymbol{d}\|_{\mathcal{G}_p}^p + C_p\|\boldsymbol{d}\|_{\mathbf{M}}^p\right) \ .
\end{aligned}
$$

completing the proof of Lemma D.12. $\qquad\square$

We also show that the subproblems we solve in Line 3 of Algorithm 2 are strongly convex.

**Lemma D.13.** *Fix $\boldsymbol{z}, \boldsymbol{q}, \boldsymbol{d} \in \mathbb{R}^d$ and let $L > 0$. Consider the function*

$$
g(\boldsymbol{x}) := \langle\boldsymbol{z}, \boldsymbol{x}\rangle + L\left(\|\boldsymbol{x} - \boldsymbol{q}\|_{\nabla^2 f(\boldsymbol{q})}^2 + C_p\|\boldsymbol{x} - \boldsymbol{q}\|_{\mathbf{M}}^p\right) \ .
$$

*Then,*

$$
g(\boldsymbol{x} + \boldsymbol{d}) \geq g(\boldsymbol{x}) + \langle\nabla g(\boldsymbol{x}), \boldsymbol{d}\rangle + L\left(\|\boldsymbol{d}\|_{\nabla^2 f(\boldsymbol{q})}^2 + \frac{4C_p}{2^p}\|\boldsymbol{d}\|_{\mathbf{M}}^p\right) \ .
$$

*In particular, if $\boldsymbol{z}$ is the minimizer for $g$, then for any $\boldsymbol{d} \in \mathbb{R}^d$, we have*

$$
\|\boldsymbol{d}\|_{\mathbf{M}} \leq \frac{2}{p \cdot (4e)^{1/p}}\left(\frac{g(\boldsymbol{z} + \boldsymbol{d}) - g(\boldsymbol{z})}{L}\right)^{1/p} \ .
$$

*Proof of Lemma D.13.* This is pretty much the same proof as Lemma D.12. It is easy to check that

$$
\|(\boldsymbol{x} + \boldsymbol{d}) - \boldsymbol{q}\|_{\nabla^2 f(\boldsymbol{q})}^2 = \|\boldsymbol{x} - \boldsymbol{q}\|_{\nabla^2 f(\boldsymbol{q})}^2 + \left\langle 2\nabla^2 f(\boldsymbol{q})(\boldsymbol{x} - \boldsymbol{q}), \boldsymbol{d}\right\rangle + \|\boldsymbol{d}\|_{\nabla^2 f(\boldsymbol{q})}^2 \ ,
$$

(18)

and using Lemma D.3 in the same way as in the proof of Lemma D.12, we have

$$
\|(\boldsymbol{x} + \boldsymbol{d}) - \boldsymbol{q}\|_{\mathbf{M}}^p \geq^{\text{equation 17}} \|\boldsymbol{x} - \boldsymbol{q}\|_{\mathbf{M}}^p + \left\langle p\|\boldsymbol{x} - \boldsymbol{q}\|_{\mathbf{M}}^{p-2}\mathbf{M}(\boldsymbol{x} - \boldsymbol{q}), \boldsymbol{d}\right\rangle + \frac{4}{2^p}\|\boldsymbol{d}\|_{\mathbf{M}}^p \ .
$$

Combining this with the definition of $g$ gives the following,

$$
\begin{aligned}
g(\boldsymbol{x} + \boldsymbol{d}) &= \langle\boldsymbol{z}, \boldsymbol{x} + \boldsymbol{d}\rangle + L\left(\|\boldsymbol{x} + \boldsymbol{d} - \boldsymbol{q}\|_{\nabla^2 f(\boldsymbol{q})}^2 + C_p\|\boldsymbol{x} + \boldsymbol{d} - \boldsymbol{q}\|_{\mathbf{M}}^p\right) \ , \\
&\geq^{\text{equation 18, equation 17}} {\color{red}\langle\boldsymbol{z}, \boldsymbol{x}\rangle} + \langle\boldsymbol{z}, \boldsymbol{d}\rangle + L\|\boldsymbol{x} - \boldsymbol{q}\|_{\nabla^2 f(\boldsymbol{q})}^2 + L\left\langle 2\nabla^2 f(\boldsymbol{q})(\boldsymbol{x} - \boldsymbol{q}), \boldsymbol{d}\right\rangle \\
&\quad + L\|\boldsymbol{d}\|_{\nabla^2 f(\boldsymbol{q})}^2 + LC_p\left(\|\boldsymbol{x} - \boldsymbol{q}\|_{\mathbf{M}}^p + \left\langle p\|\boldsymbol{x} - \boldsymbol{q}\|_{\mathbf{M}}^{p-2}\mathbf{M}(\boldsymbol{x} - \boldsymbol{q}), \boldsymbol{d}\right\rangle + \frac{4}{2^p}\|\boldsymbol{d}\|_{\mathbf{M}}^p\right) \ ,
\end{aligned}
$$

$$= g(\boldsymbol{x}) + \left\langle \boldsymbol{z} + 2L\nabla^2 f(\boldsymbol{q})(\boldsymbol{x} - \boldsymbol{q}) + LC_p p \|\boldsymbol{x} - \boldsymbol{q}\|_{\mathbf{M}}^{p-2} \mathbf{M}(\boldsymbol{x} - \boldsymbol{q}), \boldsymbol{d} \right\rangle$$

$$+ L\left( \|\boldsymbol{d}\|_{\nabla^2 f(\boldsymbol{q})}^2 + \frac{4C_p}{2^p} \|\boldsymbol{d}\|_{\mathbf{M}}^p \right) ,$$

$$= g(\boldsymbol{x}) + \langle \nabla g(\boldsymbol{x}), \boldsymbol{d} \rangle + L\left( \|\boldsymbol{d}\|_{\nabla^2 f(\boldsymbol{q})}^2 + \frac{4C_p}{2^p} \|\boldsymbol{d}\|_{\mathbf{M}}^p \right) ,$$

which proves the first result of the lemma.

To get the second result, we observe that $\nabla g(\boldsymbol{z}) = 0$ by the optimality of $\boldsymbol{z}$. Ignoring the $\|\boldsymbol{d}\|_{\nabla^2 f(\boldsymbol{q})}$ terms and rearranging gives the conclusion of Lemma D.13. $\qquad \square$

### D.3.3 SMOOTHNESS OF THE PROXIMAL OBJECTIVE

We first bound the operator norm of a matrix related to the Hessian of the proximal objective.

**Lemma D.14.** *For all $\boldsymbol{q}, \boldsymbol{y} \in \mathbb{R}^d$, we have*

$$\left\| \mathbf{M}^{-1/2} \left( \nabla^2 f_{\boldsymbol{q}}(\boldsymbol{y}) \right) \mathbf{M}^{-1/2} \right\|_{\mathrm{op}} \le ep^2(p-1) \left( 2f(\boldsymbol{q})^{1-\frac{2}{p}} + C_p \|\boldsymbol{y} - \boldsymbol{q}\|_{\mathbf{M}}^{p-2} \right) .$$

*Proof of Lemma D.14.* Recall from the proof of Lemma D.9 the definition of the regularization term $g_{\boldsymbol{q}}(\boldsymbol{y}) := C_p \|\boldsymbol{y} - \boldsymbol{q}\|_{\mathbf{M}}^p$ for $C_p = ep^p$ as well as the following calculations,

$$g_{\boldsymbol{q}}(\boldsymbol{y}) := C_p \|\boldsymbol{y} - \boldsymbol{q}\|_{\mathbf{M}}^p ,$$

$$\nabla g_{\boldsymbol{q}}(\boldsymbol{y}) = pC_p \|\boldsymbol{y} - \boldsymbol{q}\|_{\mathbf{M}}^{p-2} \mathbf{M}(\boldsymbol{y} - \boldsymbol{q}) ,$$

$$\nabla^2 g_{\boldsymbol{q}}(\boldsymbol{y}) = pC_p \|\boldsymbol{y} - \boldsymbol{q}\|_{\mathbf{M}}^{p-2} \mathbf{M} + p(p-2)C_p \|\boldsymbol{y} - \boldsymbol{q}\|_{\mathbf{M}}^{p-4} \mathbf{M}(\boldsymbol{y} - \boldsymbol{q})(\boldsymbol{y} - \boldsymbol{q})^\top \mathbf{M} .$$

By Lemma D.9, we know that

$$\nabla^2 f_{\boldsymbol{q}}(\boldsymbol{y}) \preceq ep \left( 2\nabla^2 f(\boldsymbol{q}) + \nabla^2 g_{\boldsymbol{q}}(\boldsymbol{y}) \right) .$$

Observe that

$$\mathbf{M}^{-1/2} \left( \nabla^2 g_{\boldsymbol{q}}(\boldsymbol{y}) \right) \mathbf{M}^{-1/2} = pC_p \left( \|\boldsymbol{y} - \boldsymbol{q}\|_{\mathbf{M}}^{p-2} + (p-2) \|\boldsymbol{y} - \boldsymbol{q}\|_{\mathbf{M}}^{p-4} \mathbf{M}^{1/2}(\boldsymbol{y} - \boldsymbol{q})(\boldsymbol{y} - \boldsymbol{q})^\top \mathbf{M}^{1/2} \right) ,$$

$$\preceq pC_p \|\boldsymbol{y} - \boldsymbol{q}\|_{\mathbf{M}}^{p-2} \mathbf{I} + (p-2) \|\boldsymbol{y} - \boldsymbol{q}\|_{\mathbf{M}}^{p-4} \left\| \mathbf{M}^{1/2}(\boldsymbol{y} - \boldsymbol{q})(\boldsymbol{y} - \boldsymbol{q})^\top \mathbf{M}^{1/2} \right\|_{\mathrm{op}} \mathbf{I} ,$$

$$\preceq pC_p \|\boldsymbol{y} - \boldsymbol{q}\|_{\mathbf{M}}^{p-2} \mathbf{I} + (p-2) \|\boldsymbol{y} - \boldsymbol{q}\|_{\mathbf{M}}^{p-4} \left\| \mathbf{M}^{1/2}(\boldsymbol{y} - \boldsymbol{q}) \right\|_2^2 \mathbf{I} ,$$

$$\preceq p(p-1)C_p \|\boldsymbol{y} - \boldsymbol{q}\|_{\mathbf{M}}^{p-2} \mathbf{I} ,$$

and, applying Lemma D.1 (with $\mathbf{M}^{-1/2}\boldsymbol{z}$ as the vectors in the quadratic form) and Hölder inequality with norms $\|\cdot\|_{p/(p-2)}, \|\cdot\|_{p/2}$, for $\boldsymbol{z} \in \mathbb{R}^d$ we have

$$\boldsymbol{z}^\top \mathbf{M}^{-1/2} \left( \nabla^2 f(\boldsymbol{q}) \right) \mathbf{M}^{-1/2} \boldsymbol{z} \le p(p-1) \sum_{i=1}^m \|\mathbf{A}_{S_i}\boldsymbol{q} - \boldsymbol{b}_{S_i}\|_2^{p-2} \left\| \mathbf{A}_{S_i}\mathbf{M}^{-1/2}\boldsymbol{z} \right\|_2^2$$

$$\le p(p-1) \left( \sum_{i=1}^m \|\mathbf{A}_{S_i}\boldsymbol{q} - \boldsymbol{b}_{S_i}\|_2^p \right)^{\frac{p-2}{p}} \left( \sum_{i=1}^m \left\| \mathbf{A}_{S_i}\mathbf{M}^{-1/2}\boldsymbol{z} \right\|_2^p \right)^{\frac{2}{p}}$$

$$\le p(p-1)f(\boldsymbol{q})^{1-\frac{2}{p}} \left\| \mathbf{M}^{-1/2}\boldsymbol{z} \right\|_{\mathbf{M}}^2 = p(p-1)f(\boldsymbol{q})^{1-\frac{2}{p}} \|\boldsymbol{z}\|_2^2 .$$

Combining gives

$$\mathbf{M}^{-1/2} \left( \nabla^2 f_{\boldsymbol{q}}(\boldsymbol{y}) \right) \mathbf{M}^{-1/2} \preceq ep\mathbf{M}^{-1/2} \left( 2\nabla^2 f(\boldsymbol{q}) + \nabla^2 g_{\boldsymbol{q}(\boldsymbol{y})} \right) \mathbf{M}^{-1/2} ,$$

$$\preceq 2ep^2(p-1)f(\boldsymbol{q})^{1-\frac{2}{p}} + ep^2(p-1)C_p \|\boldsymbol{y} - \boldsymbol{q}\|_{\mathbf{M}}^{p-2} ,$$

$$\preceq ep^2(p-1) \left( 2f(\boldsymbol{q})^{1-\frac{2}{p}} + C_p \|\boldsymbol{y} - \boldsymbol{q}\|_{\mathbf{M}}^{p-2} \right) ,$$

completing the proof of Lemma D.14. $\qquad \square$

Next, we show a bound on the norm of the gradient of any solution $x$ that is approximately optimal for $f_q$.

**Lemma D.15.** *For all $q, x \in \mathbb{R}^d$, we have*

$$\left\|\mathbf{M}^{-1}\nabla f_q(x)\right\|_{\mathbf{M}} \leq ep^2(p-1)\left(f(q)^{1-\frac{2}{p}} + C_p \max\left\{\|x-q\|_{\mathbf{M}}, \|x_q-q\|_{\mathbf{M}}\right\}^{p-2}\right)\|x-x_q\|_{\mathbf{M}} .$$

*Proof of Lemma D.15.* We use a continuity argument. By Taylor's theorem, we know for some $y$ along the line connecting $x$ and $x_q$ (minimizer of $f_q$) that

$$\nabla f_q(x) = \nabla f_q(x_q) + \nabla^2 f_q(y)(x - x_q) = \nabla^2 f_q(y)(x - x_q) .$$

Taking $\mathbf{M}^{-1}$-norm of both sides gives,

$$\begin{aligned}
\|\nabla f_q(x)\|_{\mathbf{M}^{-1}} &= \left\|\mathbf{M}^{-1/2}\nabla f_q(x)\right\|_2 , \\
&= \left\|\mathbf{M}^{-1/2}\nabla^2 f_q(y)(x - x_q)\right\|_2 , \\
&= \left\|\mathbf{M}^{-1/2}\nabla^2 f_q(y)\mathbf{M}^{-1/2}\mathbf{M}^{1/2}(x - x_q)\right\|_2 , \\
&\leq \left\|\mathbf{M}^{-1/2}\left(\nabla^2 f_q(y)\right)\mathbf{M}^{-1/2}\right\|_{\text{op}} \cdot \|x - x_q\|_{\mathbf{M}} .
\end{aligned}$$

The rest of the proof involves bounding the operator norm term. This follows directly from Lemma D.14, from which we get (using convexity of $\|\cdot\|_{\mathbf{M}}$),

$$\begin{aligned}
\left\|\mathbf{M}^{-1/2}\nabla^2 f_q(y)\mathbf{M}^{-1/2}\right\|_{\text{op}} &\leq ep^2(p-1)\left(2f(q)^{1-\frac{2}{p}} + C_p\|y-q\|_{\mathbf{M}}^{p-2}\right) \\
&\leq ep^2(p-1)\left(2f(q)^{1-\frac{2}{p}} + C_p \max\left\{\|x-q\|_{\mathbf{M}}, \|x_q-q\|_{\mathbf{M}}\right\}^{p-2}\right).
\end{aligned}$$

Putting everything together, we get

$$\begin{aligned}
\left\|\mathbf{M}^{-1}\nabla f_q(x)\right\|_{\mathbf{M}} &= \|\nabla f_q(x)\|_{\mathbf{M}^{-1}} , \\
&\leq ep^2(p-1)\left(2f(q)^{1-\frac{2}{p}} + C_p \max\left\{\|x-q\|_{\mathbf{M}}, \|x_q-q\|_{\mathbf{M}}\right\}^{p-2}\right)\|x-x_q\|_{\mathbf{M}} ,
\end{aligned}$$

completing the proof of Lemma D.15. $\qquad\square$

### D.3.4 SOLVING THE PROXIMAL SUBPROBLEMS

We begin by showing that the optimal solution to the proximal problem $x_{q_t} := \operatorname*{argmin}_{x \in \mathbb{R}^d} f_{q_t}(x)$ is not too far from $x^\star$.

**Lemma D.16.** *For all proximal queries $q_t$, we have*

$$\|x_{q_t} - x^\star\|_{\mathbf{M}} \leq d^{\frac{1}{2}-\frac{1}{p}}\left(2^{\frac{3}{2}}f(x_t) + 4\right).$$

*Proof.* In the rest of this proof, we omit the subscript $t$ wherever it is clear which iterates we are working with.

We first show that

$$\|x_q - q\|_{\mathbf{M}} \leq \|x^\star - q\|_{\mathbf{M}} .$$

To see this, suppose this is not the case. Then, we have

$$f(x^\star) + C_p\|x^\star - q\|_{\mathbf{M}}^p < f(x_q) + C_p\|x_q - q\|_{\mathbf{M}}^p ,$$

which contradicts the optimality of $x_q$ for $f_q$.

We now write

$$\begin{aligned}
\|x_{q_t} - x^\star\|_{\mathbf{M}} &\leq \|x_{q_t} - q_t\|_{\mathbf{M}} + \|x^\star - q_t\|_{\mathbf{M}} , \\
&\leq 2\|x^\star - q_t\|_{\mathbf{M}} ,
\end{aligned}$$

$$\leq 2\left(\|\boldsymbol{x}_t - \boldsymbol{x}^\star\|_{\mathbf{M}} + \|\boldsymbol{v}_t - \boldsymbol{x}^\star\|_{\mathbf{M}}\right) ,$$

where in the last inequality, we used the definition of $\boldsymbol{q}_t$ from Line 6 in Algorithm 3 and the convexity of $\|\cdot\|_{\mathbf{M}}$. The required control on $\|\boldsymbol{v}_t - \boldsymbol{x}^\star\|_{\mathbf{M}}$ comes from Lemma B.5 and Lemma E.5 (along with re-scaling assumption to make the optimal value 1) – we have

$$\|\boldsymbol{v}_t - \boldsymbol{x}^\star\|_{\mathbf{M}} \leq \sqrt{2}\,\|\boldsymbol{x}_0 - \boldsymbol{x}^\star\|_{\mathbf{M}} \leq 4d^{\frac{1}{2}-\frac{1}{p}} .$$

For the other term, we apply Lemma D.2 and get

$$\|\boldsymbol{x}_t - \boldsymbol{x}^\star\|_{\mathbf{M}} \leq 2^{\frac{3}{2}} d^{\frac{1}{2}-\frac{1}{p}} \left(f(\boldsymbol{x}_t) - f(\boldsymbol{x}^\star)\right)^{\frac{1}{p}} < 2^{\frac{3}{2}} d^{\frac{1}{2}-\frac{1}{p}} f(\boldsymbol{x}_t)^{\frac{1}{p}}.$$

Adding gives us the conclusion of Lemma D.16. $\qquad\square$

The next few lemmas are targeted at solving the proximal subproblems. We begin with a calculation that we will use in showing that the initial Bregman divergence between our initialization and the optimum is small.

**Lemma D.17.** *In the same setting as Lemma D.9, for all $\boldsymbol{x}, \boldsymbol{y} \in \mathbb{R}^d$, we have*

$$h_{\boldsymbol{q}}(\boldsymbol{x}_{\boldsymbol{q}}) \leq p(p-1)f(\boldsymbol{q})^{1-\frac{2}{p}} \|\boldsymbol{x}_{\boldsymbol{q}} - \boldsymbol{q}\|_{\mathbf{M}}^2 + C_p \|\boldsymbol{x}_{\boldsymbol{q}} - \boldsymbol{q}\|_{\mathbf{M}}^p < f(\boldsymbol{q}) + C_p \|\boldsymbol{x}_{\boldsymbol{q}} - \boldsymbol{q}\|_{\mathbf{M}}^p \leq 2f(\boldsymbol{q}).$$

*Proof of Lemma D.17.* By optimality of $\boldsymbol{x}_{\boldsymbol{q}}$ for the subproblem, we have

$$f(\boldsymbol{x}_{\boldsymbol{q}}) + C_p \|\boldsymbol{x}_{\boldsymbol{q}} - \boldsymbol{q}\|_{\mathbf{M}}^p \leq f(\boldsymbol{q}) + C_p \|\boldsymbol{q} - \boldsymbol{q}\|_{\mathbf{M}}^p = f(\boldsymbol{q}).$$

Rearranging gives,

$$\|\boldsymbol{x}_{\boldsymbol{q}} - \boldsymbol{q}\|_{\mathbf{M}}^p \leq \frac{f(\boldsymbol{q}) - f(\boldsymbol{x}_{\boldsymbol{q}})}{C_p} \leq \frac{f(\boldsymbol{q})}{C_p} . \tag{19}$$

We now use the definition of $h_{\boldsymbol{q}}$ and Lemma D.1 to write

$$h_{\boldsymbol{q}}(\boldsymbol{x}_{\boldsymbol{q}}) = \|\boldsymbol{x}_{\boldsymbol{q}} - \boldsymbol{q}\|_{\nabla^2 f(\boldsymbol{q})}^2 + C_p \|\boldsymbol{x}_{\boldsymbol{q}} - \boldsymbol{q}\|_{\mathbf{M}}^p ,$$

$$\leq^{\text{Lemma D.1}} p(p-1) \sum_{i=1}^m \|\mathbf{A}_{S_i}\boldsymbol{q} - \boldsymbol{b}_{S_i}\|_2^{p-2} \|\mathbf{A}_{S_i}(\boldsymbol{x}_{\boldsymbol{q}} - \boldsymbol{q})\|_2^2 + C_p \|\boldsymbol{x}_{\boldsymbol{q}} - \boldsymbol{q}\|_{\mathbf{M}}^p ,$$

$$\leq^{(a)} p(p-1)\left(\sum_{i=1}^m \|\mathbf{A}_{S_i}\boldsymbol{q} - \boldsymbol{b}_{S_i}\|_2^p\right)^{1-\frac{2}{p}} \left(\sum_{i=1}^m \|\mathbf{A}_{S_i}(\boldsymbol{x}_{\boldsymbol{q}} - \boldsymbol{q})\|_2^p\right)^{\frac{2}{p}} + C_p \|\boldsymbol{x}_{\boldsymbol{q}} - \boldsymbol{q}\|_{\mathbf{M}}^p ,$$

$$\leq^{(b)} p(p-1)f(\boldsymbol{q})^{1-\frac{2}{p}} \|\boldsymbol{x}_{\boldsymbol{q}} - \boldsymbol{q}\|_{\mathbf{M}}^2 + C_p \|\boldsymbol{x}_{\boldsymbol{q}} - \boldsymbol{q}\|_{\mathbf{M}}^p ,$$

$$\leq^{\text{equation 19}} p(p-1)f(\boldsymbol{q})^{1-\frac{2}{p}} \left(\frac{f(\boldsymbol{q})}{C_p}\right)^{\frac{2}{p}} + C_p \|\boldsymbol{x}_{\boldsymbol{q}} - \boldsymbol{q}\|_{\mathbf{M}}^p ,$$

$$=^{(C_p = ep^p)} \frac{(p-1)}{ep} f(\boldsymbol{q}) + C_p \|\boldsymbol{x}_{\boldsymbol{q}} - \boldsymbol{q}\|_{\mathbf{M}}^p ,$$

$$< f(\boldsymbol{q}) + C_p \|\boldsymbol{x}_{\boldsymbol{q}} - \boldsymbol{q}\|_{\mathbf{M}}^p ,$$

$$<^{\text{equation 19}} 2f(\boldsymbol{q}) ,$$

where in (a) we used Hölder inequality with norms $\|\cdot\|_{p/(p-2)}$, $\|\cdot\|_{p/2}$ and in (b) we used Theorem 2.3 .

This completes the proof for the series of inequalities in Lemma D.17. $\qquad\square$

We now have the tools to show how to approximately solve problems in Line 3 of Algorithm 2 when applied in our setting. Although this and future complexity bounds depend on $f(\boldsymbol{x}_t)$, we will later be able to use Theorem B.3 to "bootstrap" and get an unconditional upper bound below.

**Lemma D.18.** *Let $\alpha \leq 1/2$. In the context of Algorithm 5, there exists an algorithm that approximately solves subproblems of the form (for $p \geq 2$ and $L = pe$),*

$$z := \underset{\boldsymbol{x} \in \mathbb{R}^d}{\arg\min} \ \langle \boldsymbol{g}, \boldsymbol{x} \rangle + L \left( \|\boldsymbol{x} - \boldsymbol{q}\|_{\nabla^2 f(\boldsymbol{q})}^2 + C_p \|\boldsymbol{x} - \boldsymbol{q}\|_{\mathbf{M}}^p \right) \ ,$$

*in the sense that we output $\boldsymbol{x}$ for which,*

$$\max \left\{ \|\boldsymbol{x} - \boldsymbol{z}\|_{\mathbf{M}} , \left\| \mathbf{M}^{-1} \boldsymbol{g} + 2L \left( \mathbf{M}^{-1} \nabla^2 f(\boldsymbol{q})(\boldsymbol{x} - \boldsymbol{q}) + C_p \|\boldsymbol{x} - \boldsymbol{q}\|_{\mathbf{M}}^{p-2} (\boldsymbol{x} - \boldsymbol{q}) \right) \right\|_{\mathbf{M}} \right\} \leq \alpha \ .$$

*The algorithm takes $p^{O(1)} \log \left( \frac{pd \cdot f(\boldsymbol{q})}{\alpha} \right)$ linear-system-solves in matrices of the form $\mathbf{A}^\top \mathbf{B} \mathbf{A}$ for block-diagonal $\mathbf{B}$, where each block in $\mathbf{B}$ has size $|S_i| \times |S_i|$.*

*Proof of Lemma D.18.* This proof is lengthy, and splitting it into lemmas would disrupt the intended reading flow. So we break it up into several key components here.

**Motivation for the lemma.** First, let us see why this lemma is even useful. In each iteration of Algorithm 4, which in turn calls Algorithm 2, the main primitive is computing

$$\widetilde{\boldsymbol{x}}_i = \underset{\widetilde{\boldsymbol{x}} \in \mathbb{R}^d}{\arg\min} \ f_{\boldsymbol{q}_t}(\widetilde{\boldsymbol{x}}_{i-1}) + \langle \nabla f_{\boldsymbol{q}_t}(\widetilde{\boldsymbol{x}}_{i-1}), \widetilde{\boldsymbol{x}} - \widetilde{\boldsymbol{x}}_{i-1} \rangle + pe D_{h_{\boldsymbol{q}_t}}(\widetilde{\boldsymbol{x}}, \widetilde{\boldsymbol{x}}_{i-1}) \ ,$$

$$= \underset{\widetilde{\boldsymbol{x}} \in \mathbb{R}^d}{\arg\min} \ f_{\boldsymbol{q}_t}(\widetilde{\boldsymbol{x}}_{i-1}) + \langle \nabla f_{\boldsymbol{q}_t}(\widetilde{\boldsymbol{x}}_{i-1}), \widetilde{\boldsymbol{x}} - \widetilde{\boldsymbol{x}}_{i-1} \rangle + pe \left( h_{\boldsymbol{q}_t}(\widetilde{\boldsymbol{x}}) - h_{\boldsymbol{q}_t}(\widetilde{\boldsymbol{x}}_{i-1}) - \langle \nabla h_{\boldsymbol{q}_t}(\widetilde{\boldsymbol{x}}_{i-1}), \widetilde{\boldsymbol{x}} - \widetilde{\boldsymbol{x}}_{i-1} \rangle \right) \ ,$$

$$= \underset{\widetilde{\boldsymbol{x}} \in \mathbb{R}^d}{\arg\min} \ f_{\boldsymbol{q}_t}(\widetilde{\boldsymbol{x}}_{i-1}) - pe h_{\boldsymbol{q}_t}(\widetilde{\boldsymbol{x}}_{i-1}) + \langle \nabla f_{\boldsymbol{q}_t}(\widetilde{\boldsymbol{x}}_{i-1}) - pe \nabla h_{\boldsymbol{q}_t}(\widetilde{\boldsymbol{x}}_{i-1}), \widetilde{\boldsymbol{x}} - \widetilde{\boldsymbol{x}}_{i-1} \rangle + pe h_{\boldsymbol{q}_t}(\widetilde{\boldsymbol{x}}) \ ,$$

$$= \underset{\widetilde{\boldsymbol{x}} \in \mathbb{R}^d}{\arg\min} \ \langle \nabla f_{\boldsymbol{q}_t}(\widetilde{\boldsymbol{x}}_{i-1}) - pe \nabla h_{\boldsymbol{q}_t}(\widetilde{\boldsymbol{x}}_{i-1}), \widetilde{\boldsymbol{x}} \rangle + pe h_{\boldsymbol{q}_t}(\widetilde{\boldsymbol{x}}) \ .$$

Observe that the subproblem is of the form

$$\boldsymbol{z} = \underset{\boldsymbol{x} \in \mathbb{R}^d}{\arg\min} \ \langle \boldsymbol{g}, \boldsymbol{x} \rangle + pe h_{\boldsymbol{q}}(\boldsymbol{x}) \ ,$$

$$= \underset{\boldsymbol{x} \in \mathbb{R}^d}{\arg\min} \ \langle \boldsymbol{g}, \boldsymbol{x} \rangle + pe \left( \|\boldsymbol{x} - \boldsymbol{q}\|_{\nabla^2 f(\boldsymbol{q})}^2 + C_p \|\boldsymbol{x} - \boldsymbol{q}\|_{\mathbf{M}}^p \right) \ , \tag{20}$$

and so our goal is to show how to solve these types of problems.

**The general algorithm.** Consider solving the related subproblem (instead of equation 20),

$$\underset{\boldsymbol{x} \in \mathbb{R}^d}{\arg\min} \ \langle \boldsymbol{g}, \boldsymbol{x} \rangle + L \left( \|\boldsymbol{x} - \boldsymbol{q}\|_{\nabla^2 f(\boldsymbol{q})}^2 + C_p \tau \|\boldsymbol{x} - \boldsymbol{q}\|_{\mathbf{M}}^2 \right)$$

for some fixed $\tau \geq 0$. This is a quadratic problem, and we can therefore solve it in 1 linear-system-solve. It is easy to check that at optimality, we have

$$\boldsymbol{g} + 2pe \left( \nabla^2 f(\boldsymbol{q})(\boldsymbol{x} - \boldsymbol{q}) + C_p \tau \mathbf{M}(\boldsymbol{x} - \boldsymbol{q}) \right) = 0 \ ,$$

which rearranges to[†]

$$\boldsymbol{x} - \boldsymbol{q} = -\frac{1}{2pe} \left( \nabla^2 f(\boldsymbol{q}) + C_p \tau \mathbf{M} \right)^{-1} \boldsymbol{g} \ .$$

Note that at optimality for our original subproblem equation 20, we have $\tau^\star := \|\boldsymbol{z} - \boldsymbol{q}\|_{\mathbf{M}}^{p-2}$ where $\boldsymbol{z}$ is the solution of subproblem equation 20. Also note that $\|\boldsymbol{x} - \boldsymbol{q}\|_{\mathbf{M}}$ is a decreasing function in $\tau$ because,

$$\|\boldsymbol{x} - \boldsymbol{q}\|_{\mathbf{M}}^2 = \frac{1}{4p^2 e^2} \|\boldsymbol{g}\|_{(\nabla^2 f(\boldsymbol{q}) + C_p \tau \mathbf{M})^{-1} \mathbf{M} (\nabla^2 f(\boldsymbol{q}) + C_p \tau \mathbf{M})^{-1}}^2 \ ,$$

and for $\tau_1 \leq \tau_2$,

$$\left( \nabla^2 f(\boldsymbol{q}) + C_p \tau_1 \mathbf{M} \right)^{-1} \mathbf{M} \left( \nabla^2 f(\boldsymbol{q}) + C_p \tau_1 \mathbf{M} \right)^{-1} \succeq \left( \nabla^2 f(\boldsymbol{q}) + C_p \tau_2 \mathbf{M} \right)^{-1} \mathbf{M} \left( \nabla^2 f(\boldsymbol{q}) + C_p \tau_2 \mathbf{M} \right)^{-1} \ .$$

---

[†]Recall that $\nabla^2 f(\boldsymbol{q}) = \mathbf{A}^\top \mathbf{B}_1 \mathbf{A}$ for block-diagonal $\mathbf{B}_1$ and by construction, $\mathbf{M} = \mathbf{A}^\top \mathbf{W}^{1-\frac{2}{p}} \mathbf{A}$ where $\mathbf{W}$ consists of the block Lewis weights on the diagonal. Thus, $\nabla^2 f(\boldsymbol{q}) + C_p \tau \mathbf{M} = \mathbf{A}^\top \mathbf{B}_2 \mathbf{A}$ for block-diagonal $\mathbf{B}_2$.

We therefore see that if $\tau > \|x - q\|_{\mathbf{M}}^{p-2}$ — where $x$ is the optimal solution for a fixed $\tau$ — then we are over-regularizing and need to decrease $\tau$ and vice-versa. This means we can binary search for the appropriate value of $\tau$. To execute this, we first need to establish the accuracy up to which we have to identify $\tau$.

**Convergence in Argument.** By Lemma D.13 (setting $d = x - z$), recall that it is enough to solve sub-problem equation 20 up to additive accuracy $(p/2)^p L\alpha^p$ to get $\|x - z\|_{\mathbf{M}} \leq \alpha$. Suppose we find $\tau$ for which $\tau^\star \leq \tau \leq \tau^\star + \delta$. By writing the objectives and comparing, we see that the the $x$ we find from using $\tau$ gives us at most a $\delta \cdot d$-suboptimal solution compared to $z$. Plugging this into the bound from Lemma D.13 tells us that we should choose $\delta = (p/2)^p L\alpha^p/d$, and plugging this into the binary search over $\tau \in [0, d^p(1 + f(q))]$ gives us $p^{O(1)} \log\left(\frac{pd \cdot f(q)}{\alpha}\right)$ steps, as needed.

**First-order stationary point.** We first claim that it is enough to get

$$\left\|\mathbf{M}^{-1}\nabla h_q(x) - \mathbf{M}^{-1}\nabla h_q(z)\right\|_{\mathbf{M}} \leq \frac{\alpha}{L}.$$

Indeed, let $z$ be the optimal solution for the subproblem. This means that it must satisfy the first order stationary condition, namely,

$$g + L\nabla h_q(z) = 0.$$

Multiplying both sides by $\mathbf{M}^{-1}$, subtracting, and dividing both sides by $L$ gives us the expression we are interested in.

Writing first order stationary conditions gives both

$$g + 2L\left(\nabla^2 f(q)(x - q) + C_p\tau\mathbf{M}(x - q)\right) = 0$$
$$g + 2L\left(\nabla^2 f(q)(z - q) + C_p\tau^\star\mathbf{M}(z - q)\right) = 0.$$

Multiplying both sides of both equalities by $\mathbf{M}^{-1}$ and subtracting these gives

$$2L\left(\mathbf{M}^{-1}\nabla^2 f(q)(x - z) + C_p\left(\tau(x - q) - \tau^\star(z - q)\right)\right) = 0.$$

Expanding out $L(\mathbf{M}^{-1}\nabla h_q(x) - \mathbf{M}^{-1}h_q(z))$ and subtracting the above gives the desired condition

$$2L\left|\tau - \|x - q\|_{\mathbf{M}}^{p-2}\right| \cdot \|x - q\|_{\mathbf{M}} \overset{?}{\leq} \alpha.$$

Next, let us run the binary search from above so that we get argument convergence, i.e. $\|x - z\|_{\mathbf{M}} \leq \alpha^C \ll 0.1\alpha$ for some constant $C$. Using the fact that the approximate mirror descent step using $z$ decreases the objective value (Lemma A.4), observe that

$$\|x - q\|_{\mathbf{M}} \leq \|z - q\|_{\mathbf{M}} + \|x - z\|_{\mathbf{M}} \leq \|q - z\|_{\mathbf{M}} + 0.1\alpha \lesssim \sqrt{d}(1 + f(q)).$$

It then follows that binary searching $\tau$ to additive accuracy $\alpha(\sqrt{d}(1 + f(q)))^{-1}/L$ is sufficient. By the same argument as above, this takes $p^{O(1)} \log\left(\frac{pd \cdot f(q_t)}{\alpha}\right)$ steps, completing the proof of Lemma D.18. $\qquad\square$

We now combine Lemma D.18 with Theorem A.1 and Algorithm 2 to obtain approximate argument optimality for each proximal subproblem.

**Lemma D.19.** *Let $\gamma > 0$ and $x_q := \underset{x \in \mathbb{R}^d}{\operatorname{argmin}} f_q(x)$. There exists an algorithm that returns $x$ for which*

$$\|x - x_q\|_{\mathbf{M}} \leq \gamma.$$

*The algorithm takes at most $O\left(p^{O(1)} \log\left(ph_q(x_q)\left(\frac{4}{p\gamma}\right)^p\right)\right)$ iterations of solving subproblems of the form $\underset{x \in \mathbb{R}^d}{\operatorname{argmin}} \langle g, x \rangle + eph_q(x)$ for fixed vectors $g$ and $q$.*

*Proof of Lemma D.19.* This proof resembles (Jambulapati et al., 2022, Lemma 4.5), which uses an exact version of mirror descent arising from Lu et al. (2018). The main difference between our argument and that of (Jambulapati et al., 2022, Lemma 4.5) is that we rigorously identify a concrete upper bound on the complexity needed to satisfy the MS condition and argue that the mirror descent algorithm can handle the inexact Bregman proximal problem solves.

First, we use Lemma D.12 on the approximate solution $\boldsymbol{x}$ and true solution $\boldsymbol{x_q}$ and get,

$$f_{\boldsymbol{q}}(\boldsymbol{x}) \geq f_{\boldsymbol{q}}(\boldsymbol{x_q}) + \frac{4}{2^p} \left( \|\mathbf{A}(\boldsymbol{x} - \boldsymbol{x_q})\|_{\mathcal{G}_p}^p + C_p \|\boldsymbol{x_q} - \boldsymbol{x}\|_{\mathbf{M}}^p \right) ,$$

$$\geq f_{\boldsymbol{q}}(\boldsymbol{x}) + \frac{4C_p}{2^p} \|\boldsymbol{x_q} - \boldsymbol{x}\|_{\mathbf{M}}^p .$$

Rearranging, we get

$$\|\boldsymbol{x_q} - \boldsymbol{x}\|_{\mathbf{M}} \leq \left( \frac{2^p}{4C_p} \right)^{1/p} (f_{\boldsymbol{q}}(\boldsymbol{x}) - f_{\boldsymbol{q}}(\boldsymbol{x_q}))^{1/p} ,$$

$$= \left( \frac{2^p}{4ep^p} \right)^{1/p} (f_{\boldsymbol{q}}(\boldsymbol{x}) - f_{\boldsymbol{q}}(\boldsymbol{x_q}))^{1/p} ,$$

$$< \frac{2}{p} (f_{\boldsymbol{q}}(\boldsymbol{x}) - f_{\boldsymbol{q}}(\boldsymbol{x_q}))^{1/p} .$$

Using the notation from Lu et al. (2018), for convex $h \colon \mathbb{R}^d \to \mathbb{R}$, let

$$D_h(\boldsymbol{x}, \boldsymbol{y}) := h(\boldsymbol{x}) - h(\boldsymbol{y}) - \langle \nabla h(\boldsymbol{y}), \boldsymbol{x} - \boldsymbol{y} \rangle .$$

Recall the conclusion of Lemma D.9 – we have for $\mu = 1/(2pe)$ and $L = pe$ that

$$\mu \nabla^2 h_{\boldsymbol{q}}(\boldsymbol{x}) \preceq \nabla^2 f_{\boldsymbol{q}}(\boldsymbol{x}) \preceq L \nabla^2 h_{\boldsymbol{q}}(\boldsymbol{x}).$$

By Theorem A.1 and Lemma D.9, using the same notation from Lemma D.9, we have for all iterations $t$ of Algorithm 2 (with $f = f_{\boldsymbol{q}}$ and $h = h_{\boldsymbol{q}}$) that,

$$f_{\boldsymbol{q}}(\boldsymbol{x}_t) - f_{\boldsymbol{q}}(\boldsymbol{x_q}) \leq L \left( 1 - \frac{\mu}{L} \right)^t D_{h_{\boldsymbol{q}}}(\boldsymbol{x_q}, \boldsymbol{q}) + \max_{1 \leq i \leq t} \langle \triangle_i, \boldsymbol{x}_t - \boldsymbol{x_q} \rangle ,$$

$$= 2L \left( 1 - \frac{\mu}{L} \right)^t h_{\boldsymbol{q}}(\boldsymbol{x_q}) + \max_{1 \leq i \leq t} \langle \triangle_i, \boldsymbol{x}_t - \boldsymbol{x_q} \rangle .$$

Hence, for $t \geq \frac{L}{\mu} \log \left( L h_{\boldsymbol{q}}(\boldsymbol{x_q}) \left( \frac{4}{p\gamma} \right)^p \right)$, it is easy to check that for $p \geq 2$,

$$f_{\boldsymbol{q}}(\boldsymbol{x}_t) - f_{\boldsymbol{q}}(\boldsymbol{x_q}) \leq 2L \left( \frac{1}{e} \right)^{\log \left( L h_{\boldsymbol{q}}(\boldsymbol{x_q}) \left( \frac{4}{p\gamma} \right)^p \right)} h_{\boldsymbol{q}}(\boldsymbol{x_q}) + \max_{1 \leq i \leq t} \langle \triangle_i, \boldsymbol{x}_t - \boldsymbol{x_q} \rangle ,$$

$$= 2 \left( \frac{p\gamma}{4} \right)^p + \max_{1 \leq i \leq t} \langle \triangle_i, \boldsymbol{x}_t - \boldsymbol{x_q} \rangle ,$$

$$\leq \left( \frac{p\gamma}{2} \right)^p + \max_{1 \leq i \leq t} \langle \triangle_i, \boldsymbol{x}_t - \boldsymbol{x_q} \rangle ,$$

and combining this with Lemma D.18 to make the error term on the order of our accuracy, we get $\|\boldsymbol{x_q} - \boldsymbol{x}\|_{\mathbf{M}} \lesssim \gamma$. We thus conclude the proof of Lemma D.19. $\qquad\square$

The last step is to use our proximal problem solver to build a valid MS oracle.

**Lemma D.20.** *In the context of Algorithm 3, there exists an algorithm $(\widetilde{\boldsymbol{x}}_{t+1}, \lambda_{t+1}) = \mathcal{O}_{\text{prox}}(\boldsymbol{q}_t)$ that approximately solves*

$$\operatorname*{argmin}_{\widetilde{\boldsymbol{x}} \in \mathbb{R}^d} f(\widetilde{\boldsymbol{x}}) + ep^p \|\widetilde{\boldsymbol{x}} - \boldsymbol{q}_t\|_{\mathbf{M}}^p$$

*using $O\left( p^{O(1)} \log \left( \frac{pd \cdot f(\boldsymbol{x}_t)}{\varepsilon} \right) \right)$ linear-system-solves in $\mathbf{A}^\top \mathbf{B} \mathbf{A}$, in the sense that*

$$\left\| \frac{1}{ep^{p+1} \|\widetilde{\boldsymbol{x}}_{t+1} - \boldsymbol{q}_t\|_{\mathbf{M}}^{p-2}} \mathbf{M}^{-1} \nabla f(\widetilde{\boldsymbol{x}}_{t+1}) + (\widetilde{\boldsymbol{x}}_{t+1} - \boldsymbol{q}_t) \right\|_{\mathbf{M}} \leq \frac{1}{2} \|\widetilde{\boldsymbol{x}}_{t+1} - \boldsymbol{q}_t\|_{\mathbf{M}} .$$

*Proof of Lemma D.20.* The point of this proof is to give an analysis of Algorithm 4.

For notational simplicity, let $\boldsymbol{x} = \widetilde{\boldsymbol{x}}_{t+1}$ and $\lambda = \lambda_{t+1}$. We will reintroduce the indices when it is essential to clarify the iterations we are discussing.

First, it is helpful to see why the stated notion of approximation is useful. Let $C_p := ep^p$. Observe that at exact optimality, we have

$$\nabla f(\boldsymbol{x_q}) + \underbrace{ep^{p+1} \|\boldsymbol{x_q} - \boldsymbol{q}\|_{\mathbf{M}}^{p-2}}_{\lambda^\star} \mathbf{M}(\boldsymbol{x} - \boldsymbol{q}) = 0 \ . \tag{21}$$

This motivates the approximation in our lemma statement, with us asking for a $\frac{1}{2}$-approximate MS oracle (Definition B.1) for $f$. This also tells us that at optimality in equation 21, we have,

$$
\begin{aligned}
&\nabla f(\boldsymbol{x_q}) + ep^{p+1} \|\boldsymbol{x_q} - \boldsymbol{q}\|_{\mathbf{M}}^{p-2} \mathbf{M}(\boldsymbol{x} - \boldsymbol{q}) = 0 \ , \\
&\Leftrightarrow \mathbf{M}^{-1/2} f(\boldsymbol{x_q}) = -pC_p \|\boldsymbol{x_q} - \boldsymbol{q}\|_{\mathbf{M}}^{p-2} \mathbf{M}^{1/2}(\boldsymbol{x} - \boldsymbol{q}) \ , \\
&\Rightarrow \left\| \mathbf{M}^{-1/2} f(\boldsymbol{x_q}) \right\|_2 = pC_p \|\boldsymbol{x_q} - \boldsymbol{q}\|_{\mathbf{M}}^{p-2} \left\| \mathbf{M}^{1/2}(\boldsymbol{x} - \boldsymbol{q}) \right\|_2 \ , \\
&\Leftrightarrow \|\boldsymbol{x_q} - \boldsymbol{q}\|_{\mathbf{M}} = \left( \frac{\left\| \mathbf{M}^{-1} \nabla f(\boldsymbol{x_q}) \right\|_{\mathbf{M}}}{pC_p} \right)^{\frac{1}{p-1}} \ .
\end{aligned}
$$

We now break up our analysis into two cases. In the first, suppose that $\left\| \mathbf{M}^{-1} \nabla f(\boldsymbol{x_q}) \right\|_{\mathbf{M}} \leq \varepsilon / \|\boldsymbol{x_q} - \boldsymbol{x}^\star\|_{\mathbf{M}}$. Then, by convexity, we have

$$f(\boldsymbol{x_q}) - f(\boldsymbol{x}^\star) \leq \langle \nabla f(\boldsymbol{x_q}), \boldsymbol{x_q} - \boldsymbol{x}^\star \rangle \leq \left\| \mathbf{M}^{-1} \nabla f(\boldsymbol{x_q}) \right\|_{\mathbf{M}} \|\boldsymbol{x_q} - \boldsymbol{x}^\star\|_{\mathbf{M}} \leq \varepsilon.$$

Hence, for the rest of the proof, assume that $\left\| \mathbf{M}^{-1} \nabla f(\boldsymbol{x_q}) \right\| \geq \varepsilon / \|\boldsymbol{x_q} - \boldsymbol{x}^\star\|_{\mathbf{M}}$ (because if this is not the case, in the algorithm we can simply check whether the MS condition is satisfied – if not, then we know this assumption was violated and we are done anyway). We run the algorithm implied by Lemma D.19 and obtain an approximate solution $\boldsymbol{x}$ for which

$$\|\boldsymbol{x} - \boldsymbol{x_q}\|_{\mathbf{M}} \leq \alpha \|\boldsymbol{x_q} - \boldsymbol{q}\|_{\mathbf{M}} \text{ for } \alpha = \frac{1}{5} \min \left\{ \frac{C_p}{ep(p-1)} \left( \frac{\|\boldsymbol{x_q} - \boldsymbol{q}\|_{\mathbf{M}}}{f(\boldsymbol{q})^{\frac{1}{p}}} \right)^{p-2}, 1 \right\} \ . \tag{22}$$

Since $\alpha < 1$ the guarantee in equation 22 gives us,

$$\|\boldsymbol{x} - \boldsymbol{x_q}\|_{\mathbf{M}} \leq \alpha \|\boldsymbol{x} - \boldsymbol{q}\|_{\mathbf{M}} \leq \frac{\alpha}{1-\alpha} \|\boldsymbol{x} - \boldsymbol{q}\|_{\mathbf{M}} \ , \tag{23}$$

and further applying triangle inequality gives us

$$
\begin{aligned}
\|\boldsymbol{x_q} - \boldsymbol{q}\|_{\mathbf{M}} &\leq \|\boldsymbol{x} - \boldsymbol{q}\|_{\mathbf{M}} + \|\boldsymbol{x_q} - \boldsymbol{x}\|_{\mathbf{M}} \ , \\
&\leq \frac{1-\alpha}{1-\alpha} \|\boldsymbol{x} - \boldsymbol{q}\|_{\mathbf{M}} + \frac{\alpha}{1-\alpha} \|\boldsymbol{x} - \boldsymbol{q}\|_{\mathbf{M}} \ , \\
&\leq \frac{1}{1-\alpha} \|\boldsymbol{x} - \boldsymbol{q}\|_{\mathbf{M}} \ .
\end{aligned} \tag{24}
$$

Hence, we get

$$
\begin{aligned}
\frac{ep(p-1)f(\boldsymbol{q})^{1-\frac{2}{p}}}{C_p \|\boldsymbol{x} - \boldsymbol{q}\|_{\mathbf{M}}^{p-2}} \cdot \|\boldsymbol{x} - \boldsymbol{x_q}\|_{\mathbf{M}} &= \frac{ep(p-1)}{C_p} \cdot \left( \frac{f(\boldsymbol{q})^{\frac{1}{p}}}{\|\boldsymbol{x} - \boldsymbol{q}\|_{\mathbf{M}}} \right)^{p-2} \cdot \|\boldsymbol{x} - \boldsymbol{x_q}\|_{\mathbf{M}} \ , \\
&\leq^{equation\ 22} \frac{1}{5} \|\boldsymbol{x_q} - \boldsymbol{q}\|_{\mathbf{M}} \ , \\
&\leq^{equation\ 24} \frac{1}{5} \cdot \frac{1}{1-\alpha} \|\boldsymbol{x} - \boldsymbol{q}\|_{\mathbf{M}} \ , \\
&\leq \frac{1}{4} \|\boldsymbol{x} - \boldsymbol{q}\|_{\mathbf{M}} \ ,
\end{aligned} \tag{25}
$$

where in the last inequality, we used that $\alpha \leq \frac{1}{5}$ due to our choice in equation 22. We now call Lemma D.15, divide both sides by $\lambda$, and get

$$\left\| \frac{1}{ep^{p+1} \|\boldsymbol{x} - \boldsymbol{q}\|_{\mathbf{M}}^{p-2}} \mathbf{M}^{-1} \nabla f(\boldsymbol{x}) + (\boldsymbol{x} - \boldsymbol{q}) \right\|_{\mathbf{M}}$$

$$\leq^{\text{(Lemma D.15)}} ep(p-1) \left( \frac{f(\boldsymbol{q})^{1-\frac{2}{p}}}{C_p \|\boldsymbol{x} - \boldsymbol{q}\|_{\mathbf{M}}^{p-2}} + \max\left\{ 1, \left( \frac{\|\boldsymbol{x_q} - \boldsymbol{q}\|_{\mathbf{M}}}{\|\boldsymbol{x} - \boldsymbol{q}\|_{\mathbf{M}}} \right)^{p-2} \right\} \right) \|\boldsymbol{x} - \boldsymbol{x_q}\|_{\mathbf{M}} \ ,$$

$$\leq^{\text{equation } 24} ep(p-1) \left( \frac{f(\boldsymbol{q})^{1-\frac{2}{p}}}{C_p \|\boldsymbol{x} - \boldsymbol{q}\|_{\mathbf{M}}^{p-2}} + \frac{1}{(1-\alpha)^{p-2}} \right) \|\boldsymbol{x} - \boldsymbol{x_q}\|_{\mathbf{M}} \ ,$$

$$\leq^{\text{equation } 23} \frac{ep(p-1)f(\boldsymbol{q})^{1-\frac{2}{p}}}{C_p \|\boldsymbol{x} - \boldsymbol{q}\|_{\mathbf{M}}^{p-2}} \cdot \|\boldsymbol{x} - \boldsymbol{x_q}\|_{\mathbf{M}} + \frac{ep(p-1)\alpha}{(1-\alpha)^{p-1}} \|\boldsymbol{x} - \boldsymbol{q}\|_{\mathbf{M}} \ ,$$

$$\leq^{\text{equation } 24, \text{ equation } 22} \frac{1}{4} \|\boldsymbol{x} - \boldsymbol{q}\|_{\mathbf{M}} + \frac{ep(p-1)5^{p-2}}{4^{p-1}} \|\boldsymbol{x} - \boldsymbol{q}\|_{\mathbf{M}} \ ,$$

$$\leq \frac{1}{2} \|\boldsymbol{x} - \boldsymbol{q}\|_{\mathbf{M}} \ ,$$

giving us the approximation guarantee.

It remains to understand the complexity of solving the proximal subproblem to the accuracy required in equation 22. Plugging in $\gamma = \alpha \|\boldsymbol{x_q} - \boldsymbol{q}\|_{\mathbf{M}}$ into Lemma D.19 and using our bound on $h_{\boldsymbol{q}}(\boldsymbol{x_q})$ from Lemma D.17 gives an iteration complexity of (ignoring the constant in front of the big-$O$)

$$p^{O(1)} \log\left( p h_{\boldsymbol{q}}(\boldsymbol{x_q}) \left( \frac{2}{p\alpha \|\boldsymbol{x_q} - \boldsymbol{q}\|_{\mathbf{M}}} \right)^p \right)$$

$$\leq p^{O(1)} \log\left( p \left( p(p-1)f(\boldsymbol{q})^{1-\frac{2}{p}} \|\boldsymbol{x_q} - \boldsymbol{q}\|_{\mathbf{M}}^2 + C_p \|\boldsymbol{x_q} - \boldsymbol{q}\|_{\mathbf{M}}^p \right) \left( \frac{2}{p\alpha \|\boldsymbol{x_q} - \boldsymbol{q}\|_{\mathbf{M}}} \right)^p \right)$$

$$= p^{O(1)} \log\left( \left( \frac{2}{p} \right)^p p \left( \frac{p(p-1)f(\boldsymbol{q})^{1-\frac{2}{p}} \|\boldsymbol{x_q} - \boldsymbol{q}\|_{\mathbf{M}}^2 + C_p \|\boldsymbol{x_q} - \boldsymbol{q}\|_{\mathbf{M}}^p}{\alpha^p \|\boldsymbol{x_q} - \boldsymbol{q}\|_{\mathbf{M}}^p} \right) \right)$$

$$= p^{O(1)} \log\left( \left( \frac{2}{p} \right)^p p \left( \frac{p(p-1)f(\boldsymbol{q})^{1-\frac{2}{p}}}{\alpha^p \|\boldsymbol{x_q} - \boldsymbol{q}\|_{\mathbf{M}}^{p-2}} + \frac{C_p}{\alpha^p} \right) \right)$$

We have two cases to analyze for the value of $\alpha$. In the first, suppose we get $\alpha = \frac{1}{5}$. By the definition of $\alpha$, this means we have

$$\frac{C_p}{ep(p-1)} \left( \frac{\|\boldsymbol{x_q} - \boldsymbol{q}\|_{\mathbf{M}}}{f(\boldsymbol{q})^{\frac{1}{p}}} \right)^{p-2} \geq 1,$$

which means the complexity we get is $p^{O(1)} \log p$. We now handle the other case, i.e., $\alpha = \frac{C_p}{5ep(p-1)} \left( \frac{\|\boldsymbol{x_q} - \boldsymbol{q}\|_{\mathbf{M}}}{f(\boldsymbol{q})^{\frac{1}{p}}} \right)^{p-2}$. Here, it will be useful to keep track of the timestep $t$ that we are working with. Recall that

$$\|\boldsymbol{x_{q_t}} - \boldsymbol{q_t}\|_{\mathbf{M}}^p = \left( \frac{\|\mathbf{M}^{-1}\nabla f(\boldsymbol{x_{q_t}})\|_{\mathbf{M}}}{pC_p} \right)^{\frac{p}{p-1}} \geq \left( \frac{\varepsilon}{pC_p \|\boldsymbol{x_{q_t}} - \boldsymbol{x}^\star\|_{\mathbf{M}}} \right)^{\frac{p}{p-1}} \ , \qquad (26)$$

so the complexity we want to control is given by

$$p^{O(1)} \log\left( \left( \frac{2}{p} \right)^p p \left( \frac{2f(\boldsymbol{q_t})}{\alpha^p \|\boldsymbol{x_{q_t}} - \boldsymbol{q_t}\|_{\mathbf{M}}^p} \right) \right)$$

$$\lesssim^{\text{equation } 22} p^{O(1)} \log\left( \left( \frac{2}{p} \right)^p p \left( \frac{2 \left(5ep(p-1)\right)^p f(\boldsymbol{q_t})^{p-1}}{C_p^p \|\boldsymbol{x_{q_t}} - \boldsymbol{q_t}\|_{\mathbf{M}}^{p(p-2)} \|\boldsymbol{x_{q_t}} - \boldsymbol{q_t}\|_{\mathbf{M}}^p} \right) \right) \ ,$$

$$\lesssim p^{O(1)} \log \left( p \left( \frac{2 \left( 10(p-1) \right)^p f(\boldsymbol{q}_t)^{p-1}}{p^{p^2} \left\| \boldsymbol{x}_{\boldsymbol{q}_t} - \boldsymbol{q}_t \right\|_{\mathbf{M}}^{p(p-1)}} \right) \right) \, ,$$

$$\lesssim^{equation\ 26} p^{O(1)} \log \left( p \left( \frac{2 \left( 10e(p-1) \right)^p p^{p(p+1)} f(\boldsymbol{q}_t)^{p-1}}{p^{p^2} \epsilon^p} \right) \left\| \boldsymbol{x}_{\boldsymbol{q}_t} - \boldsymbol{x}^\star \right\|_{\mathbf{M}}^p \right) \, ,$$

$$\lesssim^{equation\ 26} p^{O(1)} \log \left( \left( \frac{2 \left( 10e(p-1) \right)^p p^{p+1} f(\boldsymbol{q}_t)^{p-1}}{\epsilon^p} \right) \left\| \boldsymbol{x}_{\boldsymbol{q}_t} - \boldsymbol{x}^\star \right\|_{\mathbf{M}}^p \right) \, ,$$

$$\lesssim p^{O(1)} \log \left( \frac{p f(\boldsymbol{q}_t) \left\| \boldsymbol{x}_{\boldsymbol{q}_t} - \boldsymbol{x}^\star \right\|_{\mathbf{M}}}{\varepsilon} \right) \, ,$$

$$\lesssim^{(\text{Lemma D.16})} p^{O(1)} \log \left( \frac{p f(\boldsymbol{q}_t) df(\boldsymbol{x}_t)}{\varepsilon} \right) \, ,$$

$$\lesssim^{(\text{Lemma D.8})} p^{O(1)} \log \left( \frac{p f(\boldsymbol{x}_t)}{\varepsilon} \right) \, ,$$

completing the proof of Lemma D.20. $\qquad\square$

### D.4 THE ALGORITHM

We are now ready to combine the results from the previous two subsections to build our algorithm for $\mathcal{G}_p$-regression and prove Theorem 2. The main algorithmic object here is Algorithm 5.

---

**Algorithm 5** GpRegression: Optimizes equation 4 up to $(1 + \varepsilon)$-multiplicative error

**Require:** Regression problems $(\mathbf{A}_{S_1}, \boldsymbol{b}_{S_1}), \ldots, (\mathbf{A}_{S_m}, \boldsymbol{b}_{S_m})$, accuracy $\varepsilon > 0$
1: Using (Manoj & Ovsiankin, 2025, Algorithm 2) with input $[\mathbf{A}|\boldsymbol{b}]$, find nonnegative diagonal $\mathbf{W}$ such that for all $\boldsymbol{x} \in \mathbb{R}^d$ and $c \in \mathbb{R}$,

$$\left\| \mathbf{A}\boldsymbol{x} - c\boldsymbol{b} \right\|_{\mathcal{G}_\infty} \leq \left\| \mathbf{W}^{\frac{1}{2} - \frac{1}{p}} \mathbf{A}\boldsymbol{x} - c\mathbf{W}^{1/2} \boldsymbol{b} \right\|_2 \leq (2(d+1))^{\frac{1}{2} - \frac{1}{p}} \left\| \mathbf{A}\boldsymbol{x} - c\boldsymbol{b} \right\|_{\mathcal{G}_\infty} .$$

2: Let $\boldsymbol{x}_0 = \left( \mathbf{A}^\top \mathbf{W}^{1 - \frac{2}{p}} \mathbf{A} \right)^{-1} \mathbf{A}^\top \mathbf{W}^{1 - \frac{2}{p}} \boldsymbol{b}.$ $\quad \triangleright \boldsymbol{x}_0 := \underset{\boldsymbol{x} \in \mathbb{R}^d}{\operatorname{argmin}} \left\| \mathbf{W}^{\frac{1}{2} - \frac{1}{p}} \mathbf{A}\boldsymbol{x} - \mathbf{W}^{\frac{1}{2} - \frac{1}{p}} \boldsymbol{b} \right\|_2.$
3: Using Algorithm 4 and Lemma D.20, implement a $\frac{1}{2}$-MS oracle for $f$ (Definition B.1)
4: Run Algorithm 3 with the oracle from the previous line and with $\boldsymbol{x}_0$ as the initialization for $O \left( \mathsf{poly}(p) \min \left\{ \mathsf{rank}\left( \mathbf{A} \right), m \right\}^{\frac{p-2}{3p-2}} \log \left( \frac{d}{\varepsilon} \right)^3 \right)$ iterations.
5: **return** $\widehat{\boldsymbol{x}}$ the output of the previous step.

---

*Proof of Theorem 2.* By writing the stationary condition of the proximal problem, it makes sense to choose $\lambda_{t+1} = ep^{p+1} \left\| \widetilde{\boldsymbol{x}}_{t+1} - \boldsymbol{q}_t \right\|_{\mathbf{M}}^{p-2}.$

It is easy to check that

$$\left\| \widetilde{\boldsymbol{x}}_{t+1} - \boldsymbol{q}_t \right\|_{\mathbf{M}} = \left( \frac{ep^{p+1} \left\| \widetilde{\boldsymbol{x}}_{t+1} - \boldsymbol{q}_t \right\|_{\mathbf{M}}^{p-2}}{\left( (ep^{p+1})^{\frac{1}{p-1}} \right)^{p-1}} \right)^{\frac{1}{(p-1)-1}} ,$$

and therefore the triple $\left( \widetilde{\boldsymbol{x}}_{t+1}, \boldsymbol{q}_t, ep^{p+1} \left\| \widetilde{\boldsymbol{x}}_{t+1} - \boldsymbol{q}_t \right\|_{\mathbf{M}}^{p-2} \right)$ always satisfies a $(p-1, (ep^{p+1})^{1/(p-1)})$-movement bound (Definition B.2).

Next, we calculate the iteration complexity we need to reduce the error to half of what we started with. For an arbitrary initial iterate $\boldsymbol{x}$, let $\delta = 0.5(f(\boldsymbol{x}) - f(\boldsymbol{x}^\star))$. By Lemma D.2, we have

$$\left\| \boldsymbol{x} - \boldsymbol{x}^\star \right\|_{\mathbf{M}}^{s+1} = \left\| \boldsymbol{x} - \boldsymbol{x}^\star \right\|_{\mathbf{M}}^p \leq 2^{3p/2} d^{p/2-1} (f(\boldsymbol{x}) - f(\boldsymbol{x}^\star)),$$

so combining this along with the fact that $c^s = ep^{p+1}$ and applying Theorem B.3 with our proximal solver Lemma D.20 yields

$$T_{\min} = \frac{p-1}{3} \left( pC_p \cdot 2^{3p/2+1} d^{p/2-1} \right)^{\frac{2}{3p-2}} \lesssim p^{5/3} d^{\frac{p-2}{3p-2}} .$$

Next, we initialize $\boldsymbol{x}_0 := \left(\mathbf{A}^\top \mathbf{W}^{1-2/p} \mathbf{A}\right)^{-1} \mathbf{A}^\top \mathbf{W}^{1-2/p} \boldsymbol{b}$. Using Theorem E.3 and Theorem E.4, we have

$$f(\boldsymbol{x}_0) \leq (2d)^{p/2-1} f(\boldsymbol{x}^\star),$$

so reaching an iterate $\boldsymbol{x}$ for which $f(\boldsymbol{x}) - f(\boldsymbol{x}^\star) \leq \varepsilon f(\boldsymbol{x}^\star)$ takes $T_{\min} \cdot \log\left(d^{p/2-1}/\varepsilon\right) = p^{8/3} d^{\frac{p-2}{3p-2}} \log\left(\frac{d}{\varepsilon}\right)$ calls to $\mathcal{O}_{\mathsf{prox}}$.

We now resolve the full iteration complexity, including the bootstrapping step to show that $f(\boldsymbol{x}_t)$ is reasonably bounded so that we get an unconditional upper bound from Lemma D.20. At the end of iteration $t$, from (loosely) inverting the bound in Theorem B.3, we know that

$$f(\boldsymbol{x}_t) - f(\boldsymbol{x}^\star) \leq \frac{(Cp^3)^{\frac{3p-2}{2}} (2d)^{\frac{p}{2}-1}}{t^{\frac{3p-2}{2}}}.$$

Since $\widetilde{\boldsymbol{x}}_{t+1}$ only depends on $\boldsymbol{q}_t$, which in turn only depends on $\boldsymbol{x}_t$ and $\boldsymbol{v}_t$, it suffices to use the above bound for $f(\boldsymbol{x}_t)$, which gives us an iteration complexity of $p^{O(1)} \log\left(\frac{pd}{\varepsilon}\right)$ to compute $\widetilde{\boldsymbol{x}}_{t+1}$ (which we get from plugging into Lemma D.20).

Combining this with the iteration complexity of $\mathcal{O}_{\mathsf{prox}}$ gives us the result of Theorem 2. □

# E  BLOCK LEWIS WEIGHTS AND PROPERTIES

In this section, we introduce *block Lewis weights* and explore some of their properties. Several of these statements can be found in Jambulapati et al. (2023a); Manoj & Ovsiankin (2025), but we include definitions and proofs here for self-completion.

We first need to define *leverage scores*.

**Definition E.1** (Leverage scores). *For a matrix $\mathbf{A} \in \mathbb{R}^{n \times d}$ with rows $\boldsymbol{a}_1, \ldots, \boldsymbol{a}_n$, let $\tau_j$ denote the $j$th leverage score of $\mathbf{A}$, which we define to be*

$$\tau_j(\mathbf{A}) := \max_{\boldsymbol{x} \in \mathbb{R}^d \setminus \{0\}} \frac{\langle \boldsymbol{a}_j, \boldsymbol{x} \rangle^2}{\|\mathbf{A}\boldsymbol{x}\|_2^2} = \boldsymbol{a}_j^\top \left(\mathbf{A}^\top \mathbf{A}\right)^{-1} \boldsymbol{a}_j .$$

We now introduce the main object of interest in this section, Definition E.2. Our version of the definition is adapted from (Manoj & Ovsiankin, 2025, Definition 1.2) (there, we set $p_1 = \cdots = p_m = 2$, let their $\mathbf{W} = \mathbf{I}$, replace $\boldsymbol{\lambda}$ with $\boldsymbol{w}/\|\boldsymbol{w}\|_1$, and rescale $F^\star$ appropriately).

**Definition E.2** (Adapted from (Manoj & Ovsiankin, 2025, Definition 1.2)). *Let $\boldsymbol{w} \in \mathbb{R}_{\geq 0}^m$ and $\mathbf{W} \in \mathbb{R}_{\geq 0}^{n \times n}$ be a diagonal matrix for which for all $j \in S_i$, we have $\mathbf{W}_{jj} = w_i$. Let $p > 0$. We say that $\boldsymbol{w}$ is a block Lewis overestimate if for all $i \in [m]$, we have*

$$\frac{\sum_{j \in S_i} \tau_j \left(\mathbf{W}^{\frac{1}{2}-\frac{1}{p}} \mathbf{A}\right)}{w_i} \leq 1 .$$

The main reason that Definition E.2 is interesting is that it gives us a formula with which we can relate the level sets of the group norm $\|\cdot\|_{\mathcal{G}_p}$ to $\ell_2$. See Theorem E.3.

**Theorem E.3** (Block Lewis weights give us ellipsoidal approximations to $\|\cdot\|_{\mathcal{G}_p}$). *Let $p \geq 2$. If $\boldsymbol{w}$ is a block Lewis overestimate, then for all $\boldsymbol{x} \in \mathbb{R}^d$, we have*

$$\frac{\left\|\mathbf{W}^{\frac{1}{2}-\frac{1}{p}} \mathbf{A}\boldsymbol{x}\right\|_2}{\|\boldsymbol{w}\|_1^{\frac{1}{2}-\frac{1}{p}}} \leq \|\mathbf{A}\boldsymbol{x}\|_{\mathcal{G}_p} \leq \left\|\mathbf{W}^{\frac{1}{2}-\frac{1}{p}} \mathbf{A}\boldsymbol{x}\right\|_2 .$$

We prove Theorem E.3 in Appendix E. An analogous statement can also be shown for $p \leq 2$, but since we do not use it in this paper, we do not write it here.

Observe that if we can get $\boldsymbol{w}$ that satisfies Definition E.2 and for which $\|\boldsymbol{w}\|_1 = \mathsf{rank}(\mathbf{A})$, then Theorem E.3 gives us the optimal relationship between $\ell_2$ and $\|\cdot\|_{\mathcal{G}_p}$ whenever $\mathsf{rank}(\mathbf{A}) \leq m$.

Furthermore, for intuition, suppose $p = \infty$. By John's theorem, we know that for any symmetric convex body, there exists an ellipsoid such that the ellipsoid approximates the convex body up to a $\sqrt{d}$ distortion. Moreover, this is worst-case tight (e.g. the best distortion we can get when we approximate $\ell_1^d$ with $\ell_2$ is $\sqrt{d}$). Thus, assuming we can find $\|w\|_1 \approx \operatorname{rank}(\mathbf{A})$, in this case, we get a guarantee that is similar to what John's theorem tells us.

Now, assuming we can find a low-distortion ellipsoidal approximation to the level sets of our loss, we get that the "effective" diameter of our problem is $\sim \sqrt{d}$. Combining this and the discussion in Section 2.3 (or, more formally, Theorem B.3), we can see why we should expect an iteration complexity of $\sim d^{1/3}$ (or better, if we can find a better ellipsoid).

What is left is whether weights $w$ satisfying Definition E.2 with small sum can be found. To this end, we invoke (Manoj & Ovsiankin, 2025, Algorithm 2).

**Theorem E.4** ((Manoj & Ovsiankin, 2025, Algorithm 2 and Lemma 5.6)). *There exists an algorithm that returns a block Lewis overestimate $w$ for which $\|w\|_1 \leq 2\operatorname{rank}(\mathbf{A})$. The algorithm runs in $O(\log m)$ linear system solves with matrices of the form $\mathbf{A}^\top \mathbf{D}\mathbf{A}$ for nonnegative diagonal $\mathbf{D}$.*

Thus, by applying Theorem E.4 as a preprocessing step, we get an $\ell_2$ geometry under which we can run the accelerated proximal algorithms. As an example of the power of this, observe the following.

**Lemma E.5.** *Consider the matrix $\widehat{\mathbf{A}} := \mathbf{A}|b \in \mathbb{R}^{n \times (d+1)}$ that is formed by appending the column vector $b$ to the right of the matrix $\mathbf{A}$. If we have a vector $w$ of block Lewis overestimates for the matrix $\widehat{\mathbf{A}}$, then there exists an algorithm that finds an initialization $x_0$ for which*

$$\|x_0 - x^\star\|_{\mathbf{A}^\top \mathbf{W}^{\frac{1}{2} - \frac{1}{p}} \mathbf{A}} \leq 2\left(2\operatorname{rank}(\mathbf{A})\right)^{\frac{1}{2} - \frac{1}{p}} \|\mathbf{A}x^\star - b\|_{\mathcal{G}_p}$$
$$\|\mathbf{A}x_0 - b\|_{\mathcal{G}_p} \leq \left(2\operatorname{rank}(\mathbf{A})\right)^{\frac{1}{2} - \frac{1}{p}} \|\mathbf{A}x^\star - b\|_{\mathcal{G}_p}$$

*The algorithm runs in 1 linear system solve in $\widehat{\mathbf{A}}^\top \mathbf{D}\widehat{\mathbf{A}}$.*

*Proof of Lemma E.5.* By Theorem E.3, our weights $w$ are such that for all $x \in \mathbb{R}^n$ and reals $c \in \mathbb{R}$,

$$\frac{\left\|\mathbf{W}^{\frac{1}{2} - \frac{1}{p}} \mathbf{A}x - c\mathbf{W}^{\frac{1}{2} - \frac{1}{p}} b\right\|_2}{(2(d+1))^{\frac{1}{2} - \frac{1}{p}}} \leq \|\mathbf{A}x - cb\|_{\mathcal{G}_p} \leq \left\|\mathbf{W}^{\frac{1}{2} - \frac{1}{p}} \mathbf{A}x - c\mathbf{W}^{\frac{1}{2} - \frac{1}{p}} b\right\|_2.$$

Let $x_0$ be the solution to the least squares regression problem

$$x_0 := \operatorname*{argmin}_{x \in \mathbb{R}^d} \left\|\mathbf{W}^{\frac{1}{2} - \frac{1}{p}} \mathbf{A}x - \mathbf{W}^{\frac{1}{2} - \frac{1}{p}} b\right\|_2 = \left(\mathbf{A}^\top \mathbf{W}^{1 - \frac{2}{p}} \mathbf{A}\right)^{-1} \mathbf{A}^\top \mathbf{W}^{\frac{1}{2} - \frac{1}{p}} b.$$

It is easy to see that computing $x_0$ amounts to 1 linear system solve in $\mathbf{A}^\top \mathbf{D}\mathbf{A}$.

Next, let $\mathbf{M} := \mathbf{A}^\top \mathbf{W}^{1 - \frac{2}{p}} \mathbf{A}$ and observe that

$$\|x_0 - x^\star\|_{\mathbf{M}} = \left\|\left(\mathbf{W}^{\frac{1}{2} - \frac{1}{p}} \mathbf{A}x_0 - \mathbf{W}^{\frac{1}{2} - \frac{1}{p}} b\right) - \left(\mathbf{W}^{\frac{1}{2} - \frac{1}{p}} \mathbf{A}x^\star - \mathbf{W}^{\frac{1}{2} - \frac{1}{p}} b\right)\right\|_2$$
$$\leq 2\left\|\mathbf{W}^{\frac{1}{2} - \frac{1}{p}} \mathbf{A}x^\star - \mathbf{W}^{\frac{1}{2} - \frac{1}{p}} b\right\|_2 \leq 2(2d)^{\frac{1}{2} - \frac{1}{p}} \|\mathbf{A}x^\star - b\|_{\mathcal{G}_p}.$$

Finally, write

$$\|\mathbf{A}x_0 - b\|_{\mathcal{G}_p} \leq \left\|\mathbf{W}^{\frac{1}{2} - \frac{1}{p}} \mathbf{A}x_0 - \mathbf{W}^{\frac{1}{2} - \frac{1}{p}} b\right\|_2$$
$$\leq \left\|\mathbf{W}^{\frac{1}{2} - \frac{1}{p}} \mathbf{A}x^\star - \mathbf{W}^{\frac{1}{2} - \frac{1}{p}} b\right\|_2 \leq (2d)^{\frac{1}{2} - \frac{1}{p}} \|\mathbf{A}x^\star - b\|_{\mathcal{G}_p},$$

giving us the conclusion of Lemma E.5. $\square$

*Proof of Theorem E.3.* Let $\lambda := w/\|w\|_1$ and $\Lambda := \mathbf{W}/\|w\|_1$. It is easy to check that $\lambda$ is a probability measure on $[m]$. When $p \geq 2$, using monotonicity of $L_p$ norms taken under probability measures, we get

$$\left(\sum_{i=1}^m \|\mathbf{A}_{S_i} x\|_2^p\right)^{\frac{1}{p}} = \left(\sum_{i=1}^m \lambda_i \left\|\lambda_i^{-\frac{1}{p}} \mathbf{A}_{S_i} x\right\|_2^p\right)^{\frac{1}{p}} \geq \left(\sum_{i=1}^m \lambda_i \left\|\lambda_i^{-\frac{1}{p}} \mathbf{A}_{S_i} x\right\|_2^2\right)^{1/2}.$$

Expanding the RHS and substituting $\lambda_i = w_i / \|\boldsymbol{w}\|_1$ gives

$$\|\mathbf{A}\boldsymbol{x}\|_{\mathcal{G}_p} \geq \frac{\left\|\mathbf{W}^{\frac{1}{2}-\frac{1}{p}}\mathbf{A}\boldsymbol{x}\right\|_2}{\|\boldsymbol{w}\|_1^{\frac{1}{2}-\frac{1}{p}}}.$$

For the "hard" direction, we will use Definition E.2 in a nontrivial way. Notice that

$$\left(\sum_{i=1}^m w_i \left\|w_i^{-\frac{1}{p}}\mathbf{A}_{S_i}\boldsymbol{x}\right\|_2^p\right)^{\frac{1}{p}} = \left(\sum_{i=1}^m w_i \left\|w_i^{-\frac{1}{p}}\mathbf{A}_{S_i}\boldsymbol{x}\right\|_2^2 \left\|w_i^{-\frac{1}{p}}\mathbf{A}_{S_i}\boldsymbol{x}\right\|_2^{p-2}\right)^{\frac{1}{p}}$$

$$\leq \left(\sum_{i=1}^m w_i \left\|w_i^{-\frac{1}{p}}\mathbf{A}_{S_i}\boldsymbol{x}\right\|_2^2 \cdot \max_{\boldsymbol{x}\in\mathbb{R}^d\setminus\{0\}} \frac{\left\|w_i^{-\frac{1}{p}}\mathbf{A}_{S_i}\boldsymbol{x}\right\|_2^{p-2}}{\left\|\mathbf{W}^{\frac{1}{2}-\frac{1}{p}}\mathbf{A}\boldsymbol{x}\right\|_2^{p-2}}\right)^{\frac{1}{p}}$$

$$= \left(\sum_{i=1}^m w_i \left\|w_i^{-\frac{1}{p}}\mathbf{A}_{S_i}\boldsymbol{x}\right\|_2^2 \cdot \left(\max_{\boldsymbol{x}\in\mathbb{R}^d\setminus\{0\}} \frac{\left\|w_i^{-\frac{1}{p}}\mathbf{A}_{S_i}\boldsymbol{x}\right\|_2^2}{\left\|\mathbf{W}^{\frac{1}{2}-\frac{1}{p}}\mathbf{A}\boldsymbol{x}\right\|_2^2}\right)^{\frac{p}{2}-1} \cdot \left\|\mathbf{W}^{\frac{1}{2}-\frac{1}{p}}\mathbf{A}\boldsymbol{x}\right\|_2^{p-2}\right)^{\frac{1}{p}}$$

$$\leq \left(\sum_{i=1}^m w_i \left\|w_i^{-\frac{1}{p}}\mathbf{A}_{S_i}\boldsymbol{x}\right\|_2^2 \cdot \left(\frac{\sum_{j\in S_i}\tau_j\left(\mathbf{W}^{\frac{1}{2}-\frac{1}{p}}\mathbf{A}\right)}{w_i}\right)^{\frac{p}{2}-1} \left\|\mathbf{W}^{\frac{1}{2}-\frac{1}{p}}\mathbf{A}\boldsymbol{x}\right\|_2^{p-2}\right)^{\frac{1}{p}}$$

$$\overset{\text{Definition E.2}}{\leq} \left(\sum_{i=1}^m w_i \left\|w_i^{-\frac{1}{p}}\mathbf{A}_{S_i}\boldsymbol{x}\right\|_2^2 \left\|\mathbf{W}^{\frac{1}{2}-\frac{1}{p}}\mathbf{A}\boldsymbol{x}\right\|_2^{p-2}\right)^{\frac{1}{p}} = \left\|\mathbf{W}^{\frac{1}{2}-\frac{1}{p}}\mathbf{A}\boldsymbol{x}\right\|_2,$$

so combining our upper and lower bounds gives the conclusion of Theorem E.3. $\qquad\square$

# F    EMPIRICAL EVALUATION OF OUR APPROACH AGAINST OTHER BASELINES

## F.1    SYNTHETIC HETEROGENEOUS REGRESSION CONSTRUCTION

We construct synthetic group-structured regression problems designed to test optimization under severe group heterogeneity. The data consist of $m$ groups. Each group $i$ has its own design matrix $\mathbf{A}_{S_i}$ and target vector $\boldsymbol{b}_{S_i}$, and defines a quadratic loss

$$\ell_i(x) = \frac{1}{n_i}\|\mathbf{A}_{S_i}\boldsymbol{x} - \boldsymbol{b}_{S_i}\|_2^2 .$$

The objective of interest is the worst-group loss (as in equation 2)

$$F(x) = \max_{i\in[m]} \ell_i(\boldsymbol{x}) .$$

We generate two qualitatively distinct group types to separate average performance from worst-group performance. The majority of groups are benign and geometrically aligned, whereas a small number have very high curvature, are geometrically misaligned, and are far from the population center. This construction ensures that minimizing the average loss does not coincide with minimizing the worst-group loss, and that curvature heterogeneity strongly affects optimization behavior.

We construct all group covariances relative to a shared orthonormal coordinate system. This allows us to control curvature direction-by-direction while keeping the ambient geometry comparable across groups. Each group, therefore, has a quadratic loss whose Hessian shares eigenvectors with the others but whose eigenvalues vary across groups.

**Normal groups.** Most groups are generated with a moderate condition number. Their Hessians have eigenvalues that vary across coordinates but remain within a controlled range. The corresponding optimal parameters are sampled from a distribution concentrated around a common center in parameter space. Independent noise is added to each group so that these losses are smooth and moderately curved. As a result, normal groups are geometrically aligned: their curvature structure is similar, and their optima lie in a relatively small region of parameter space.

**Outlier groups.** A small subset of groups is constructed to be adversarial in two distinct ways. First, each adversarial group has one direction with extremely large curvature, while the remaining directions retain moderate curvature. These sharp directions differ across adversarial groups. Second, the optimal parameter of each adversarial group lies far from the population center along its corresponding high-curvature direction. In addition, the noise level in these groups is set to be very small, so their losses are sharply concentrated around their optima. Together, these properties ensure that deviations along the sharp directions incur very large increases in the worst-group loss.

**Implications of our construction.** This construction produces three key phenomena:

1. The stacked design matrix has a large condition number.

2. The curvature directions that dominate different groups are misaligned.

3. The empirical risk minimizer (which minimizes the average loss) can perform well on most groups while incurring substantial loss on a small number of adversarial groups.

Because most groups share similar geometry, gradient averaging implicitly emphasizes their curvature structure. Therefore, first-order methods, reduce the average loss efficiently but make comparatively slow progress along the sharp directions that control the worst-group objective. In contrast, methods that adapt to local curvature or explicitly control worst-case behavior can continue to decrease the max-loss objective.

For the default instance used in Figure 1, the problem dimension is $d = 10$ with 100 groups, of which 5 are adversarial. The stacked Gram matrix has a condition number on the order of $10^5$, and the empirical risk minimizer exhibits a clear gap relative to the robust optimum computed via convex programming. For more details on data generation, see the included Jupyter notebook.

### F.2 COMPUTING THE ROBUST OPTIMUM VIA CONVEX PROGRAMMING

To obtain a reliable reference point for evaluation, we compute the exact optimum value of the worst-group objective using a convex solver. Concretely, we solve the robust regression problem that minimizes the maximum group mean-squared error. Although the objective takes the form of a pointwise maximum over groups, it remains a convex function of the parameter vector because each group loss is a convex quadratic.

We programatically formulate this problem using the epigraph trick. We introduce an auxiliary scalar variable $t$ that upper-bounds every group loss, and then minimize this upper bound. If we write the group loss as the mean squared residual for that group, then the epigraph reformulation becomes

$$\min_{x, t} \quad t \quad \text{subject to} \quad \ell_i(\boldsymbol{x}) \leq t \text{ for every group } i \in [m] \ .$$

Each constraint is a convex quadratic inequality, so the resulting problem is a convex quadratically constrained program. We implement this formulation in CVXPY (Diamond & Boyd, 2016; Agrawal et al., 2018) by declaring decision variables for the model parameters and the epigraph variable, adding one quadratic constraint per group, and calling a standard convex solver. We treat the returned epigraph value as the robust optimum value.

In all plots, we report the gap between an algorithm's current worst-group loss and this robust optimum value. This subtraction makes the figures directly comparable across instances and highlights whether a method continues to reduce the true robust suboptimality, rather than merely decreasing a surrogate objective.

### F.3 BASELINES

We compare the following methods.

**Subgradient method.** We run subgradient descent directly on the nonsmooth max-loss objective 2, using both fixed and diminishing step-size schedules, and report the best variant.

**Smoothed gradient methods.**  We apply log-sum-exp smoothing (see 7) to approximate the max operator and optimize the resulting smooth objective using:

- Gradient descent,

- Heavy-Ball (Polyak) momentum,

- Nesterov acceleration.

**Interior-point method.**  We implement a log-barrier-based interior-point method that solves a sequence of smooth approximations using Newton steps.

**Ball-oracle methods (ours).**  We implement two trust-region style methods that repeatedly solve the smoothed objective using a damped Newton solver (Section 2.2):

- Euclidean geometry (naive ball),

- Lewis-weight geometry, where the trust region is defined using a data-dependent positive definite matrix constructed from block Lewis weights.

After each outer step, the center is updated to the new solution, and the trust-region radius is optionally shrunk. For simplicity, we do not consider the acceleration of the ball-oracle method.

### F.4 Hyperparameter tuning

We tune every method via grid search over its relevant hyperparameters:

- Step sizes for gradient and subgradient methods;

- Smoothing parameters and momentum coefficients for smoothed methods;

- Barrier parameters and inner iteration counts for the interior-point method;

- Initial trust-region radius, smoothing strength, and radius decay factors for the ball-oracle methods.

All methods use the same warm start. For each configuration, we run a fixed number of outer iterations and select the configuration that achieves the lowest worst-group loss within this budget.

### F.5 Empirical behavior

Notably, the meaning of an iteration differs across algorithms:

- For subgradient and smoothed gradient methods, one iteration corresponds to one full gradient or subgradient update using all groups.

- For the interior-point method, one iteration corresponds to one outer Newton step of the barrier procedure.

- For the ball-oracle methods, one iteration corresponds to one call to the trust-region Newton solver (i.e., one outer iteration).

In all iteration- omplexity plots, we compare methods using their own natural outer iteration count.

**Iteration complexity.**  On the adversarial instances, first-order methods make limited progress. Subgradient descent improves briefly but quickly plateaus. Even the best-tuned smoothed-gradient variant stalls far above the optimum. In contrast, both ball-oracle methods steadily decrease the worst-group loss across outer iterations, whereas the IPM converges rapidly. We also note that the IPM achieves the best final loss among all methods, which is unsurprising, as `CVXPY` natively uses the same algorithm to compute the maximum-loss optimum.

**Runtime complexity.**  While we do not spend significant effort simulating a time-complexity model or hyper-optimizing our code, we also plot the runtime complexity of all the algorithms to control for the different meanings of an iteration. Because Newton steps are computationally more expensive, first-order methods initially appear competitive in wall-clock time. However, they plateau early and fail to approach high accuracy. Interior-point and ball-oracle methods continue to reduce the worst-group loss gap and eventually achieve near-optimal solutions, whereas first-order methods remain stuck far from the optimum. We observe a very slight benefit from using the Lewis geometry in our ball oracle method.

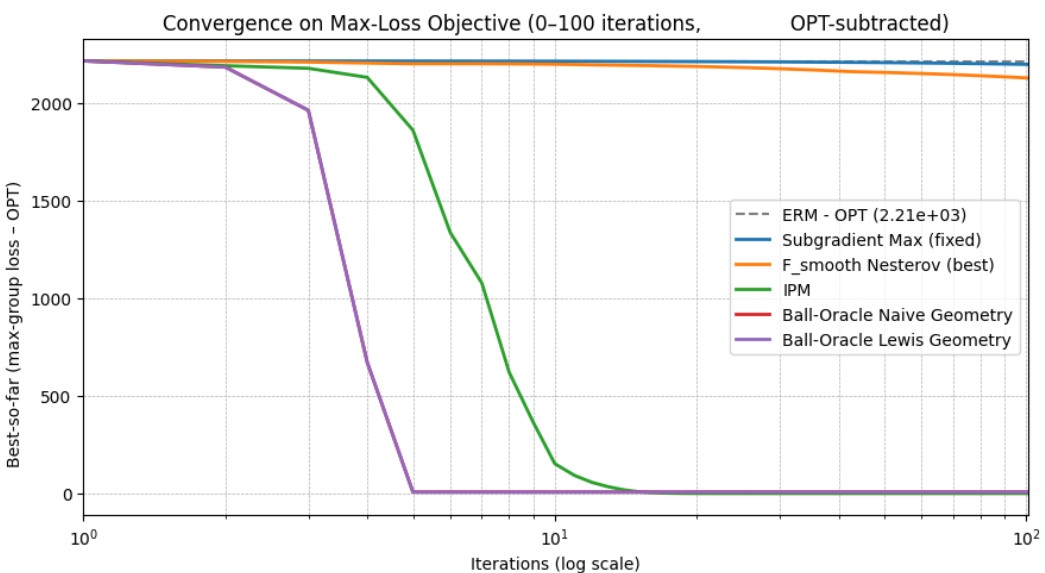

(a) Max-loss suboptimality versus iteration count (log scale on the x-axis).

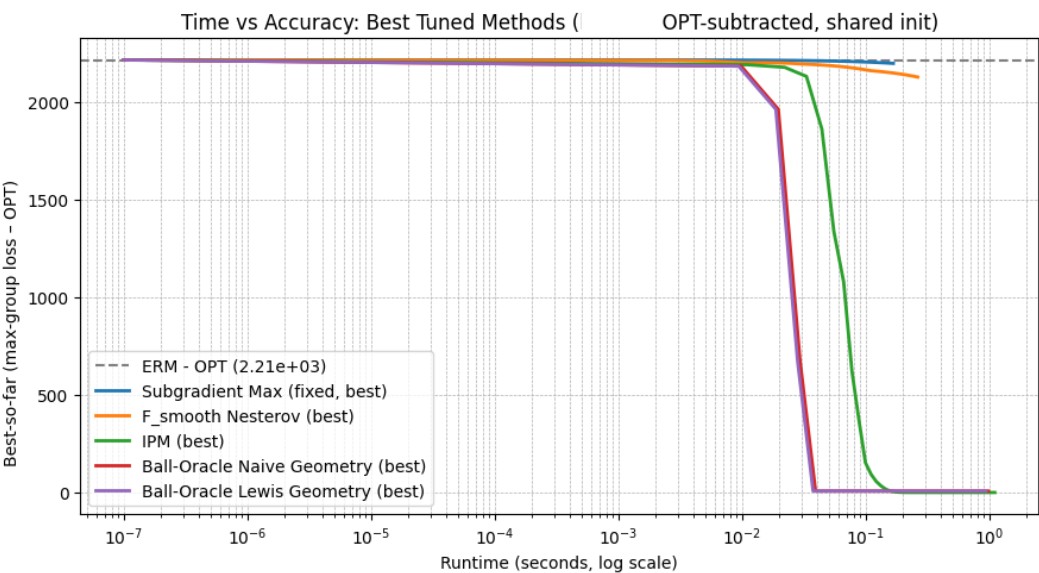

(b) Max-loss suboptimality versus runtime in seconds (log scale on the x-axis).

Figure 1: Comparison of first-order, interior-point, and ball-oracle methods on adversarial heterogeneous regression instances. All curves report the worst-group loss minus the optimal value computed via CVXPY.

