# OpenReview forum: "Distributionally Robust Linear Regression with Block Lewis Weights"
_ICLR.cc/2026/Conference — ICLR 2026 Poster_

### Official Review · Reviewer_kYbK · 2025-10-29

**Soundness:** 3
**Presentation:** 3
**Contribution:** 3
**Rating:** 8
**Confidence:** 3

**Summary:**

The paper proposes two algorithms for solving the “egalitarian”/“group DRO objective” of multi-group regression: one algorithm for the “robust” version of it—i.e., the maximum over each group’s squared prediction error—and one which interpolates from the “robust” version to the fully “nonrobust” version—i.e., the average squared prediction error across groups. The contribution is to obtain, in terms of linear-system-solves of some specific matrices of the problem, approximation guarantees that: (i) holds for more families of objective functions than currently in the literature, and (ii) can match best-known error accuracies in the literature for particular cases.

**Strengths:**

The strengths of the paper are:
- It is rigorous and the authors have tried to explain the layout of their proof in the main paper, making explicit connections with previously published results that they use, expand, or improve on.
- The contribution of the paper is clear and the problem they study can benefit diverse communities in machine learning, statistics, and signal processing.
- Results are clearly compared with the existing literature, e.g., see Table 1.

**Weaknesses:**

I have a series of things to point out about the paper.

**>>About clarity and results:**
- Line 107 mentions that, compared to all other methods in Table 1, the paper’s results do not have “geometry-dependent terms”. Can the authors pinpoint what those “geometry-dependent terms” are in the other methods from Table 1? A definition of what this phrase means will be useful.
- In line 151 inside Remark 1.2: the term “morally equivalent” does not sound technically correct. What do the authors mean by this? Perhaps another phrase could be used.
- Theorem 2 is also valid for $p=2$, which is also the setting of Theorem 1, though the complexity of linear-system-solved are different between both theorems. Can the authors comment on the differences between the complexity of these two approximation results? Can the authors also comment on the main differences between Algorithms 3 and 5 and how these reflect on the approximation/complexity results?
- Lines 225-226 states that “our results suggest that optimizing (...) between non-robust and robust objectives may be *computationally easier* than optimizing for the robust objective alone”. Could you indicate how this is suggested by simply looking at Theorems 1 and 2, or is there some missing information?
- Equation (6), which is the decomposition of equation (2) or (4) into subproblems, is introduced (i) without explanation of how it was obtained and (ii) without a formal justification for it. Can the authors explain this? Also, the authors claim that this decomposition is a common procedure in convex optimization—see line 285—however, no citation is added.
- I am trying to understand the paragraph that starts in line 356. I am sure that the second sentence is, using the notation of Definition 2.1, referring to $f(\cdot)=\sqrt{\delta^2+||\cdot - b\_{s\_i}||\_2^2}-\delta$ and the norm $||\cdot||\_2$. Can the authors confirm this? For some reason the authors are introducing the term “$A_{S_i}x$” as an argument, which is confusing.
- When finishing Section 2.1.1, I don’t see any indication on how to choose the radius $r_q$ in the proof outlined so far. Is this something which is missing or am I missing something? This is important since line 366 of the next Section 2.1.2 seems to indicate that $r_q$ has already been “explicitly” constrained in Section 2.1.1.
- Line 407: it is mentioned that the “diameter of our problem will have decreased by a constant factor”. However, what does “decreased by a constant factor” mean here? Is it an *additive* constant factor or a *multiplicative* one? I suppose it is not an additive factor, unless it somehow shrinks due to some parameter.

**>> Paper organization:**
- It is important to include Algorithm 3 in the main paper (if possible, I would also suggest including Algorithm 5, but I understand that this could be more complicated since it uses Algorithm 4 and other results that may be difficult to fully include). This will make things more self-contained. Also, Algorithm 3 shows mathematical constructions and references mentioned in Section 2, which can help the reader to better follow the paper.
- Move Table 1 earlier in the paper, close to the first text where it is referred to: right after Theorem 1.

**>>About the title:**
- The “Block Lewis Weights” are only mentioned as part of a technique for the papers’ proofs: the reader has to wait **until the last page of the paper**, where Theorem 2.3 is, to know what they are. It is strange that nowhere earlier in the paper, **including the abstract itself**, an idea of what “BLock Lewis Weights” are provided, even though the phrase is in the title itself. Thus, I strongly suggest two things:
  - Change the title to something like “Solving Distributionally Robust Linear Regression Using Block Lewis Weights”.
  - Mention earlier in the paper, perhaps Section 1.1 when introducing the results, what Block Lewis Weights are and how they are used in the proof.


**>>Other things:**
- There is a typo in equation (4): the exponent “$2$” of the $l_2$-norm of the prediction error should be “$p$” instead.
- In the related literature mentioned in Section 1.2, I suggest mentioning works that have studied the connection between distributionally robust optimization (DRO) and the effect of regularization on regression problems and other estimation problems. For example, two works come into mind:
  - “Robust Wasserstein profile inference and applications to machine learning” by Blanchet et al., 2019 studies this problem in the context of logistic regression and linear regression (including lasso).
  - “Distributionally Robust Formulation and Model Selection for the Graphical Lasso” by Cisneros-Velarde et al., 2020, studies this problem in the context of precision matrix estimation.
- The last sentence of the first paragraph (line 029) needs a citation.
- I suggest using parentheses every time an equation is referenced. E.g., instead of “equation 1” use “equation (1)”.
- The paper mentions references where equation (2) has been used; however, no references are provided where equation (1) has also been used.
- Line 145: instead of the word “promised”, a more appropriate term would be “guaranteed”.
- Line 201: I suggest indicating in the paper that “DRO” stands for “Distributionally Robust Optimization”.
- Line 294: when mentioning the “three questions” that arise when dealing with the subproblems, I would suggest adding the fact that these three questions correspond to three *consecutive* subproblems/steps of the main proof.
- Title of Lemma D.3.: it may be better to use “$||\cdot||_2^p$” instead of “$||y||_2^p$”
- Lines 411-422: I suggest replacing the expression “It would be nice to obtain this” by “It was difficult to obtain this”.
- Line 436: the word “existentially” does not seem appropriate, perhaps replace it with “further”.

**Questions:**

Please, see the Weaknesses section.

---

> ### Author Response · Authors · 2025-12-01
>
> We thank the reviewer for the careful read of our paper and for their positive feedback. We will fix all the typos and further clarify the reviewer’s questions in the revision. We respond to the questions below.
>
> ## Weaknesses
>
> We address the reviewer’s comments in the same order as the original review.
>
> ### Comments about clarity and results
>
> > Geometry-dependent terms
>
> By this, we mean terms that depend on the values in the input (e.g., the scale of the input, the condition numbers, $\|x^{\star}\|$, etc) as opposed to the structure of the input itself (i.e., the dimension and the number of groups).
>
> > “Morally equivalent”
>
> We mean that for applications and in the context of our motivations, there is not much point in solving the problem for $p \gtrsim \log m$ because up to constant factors, this is equivalent to solving the fully robust version of the problem ($p = \infty$). We will change the wording accordingly.
>
> > The complexity of linear-system-solves are different between both theorems
>
> The setting of $p=\infty$ recovers Theorem 1 and not $p=2$ ($p=2$ recovers the nonrobust version of the problem, which is just a single least squares problem). We clarify this in line 126 of the submission. Additionally, the structure of the linear systems we solve in both settings is the same (they are both in $A^{\top}BA$ for block-diagonal $B$).
>
> Next, let us more explicitly compare the guarantees of Theorems 1, 2, and the trivial nonrobust problem. On the one extreme ($p=2$, nonrobust), we can check that the objective is just a least-squares problem. This can be optimized using just $1$ linear-system-solve. On the other extreme ($p=\infty$, robust), we get a guarantee of $\min(m, \mathrm{rank}(A))^{1/3}\varepsilon^{-2/3}$ linear-system-solves. Finally, in the interpolating case ($2 < p < \infty$), we get a guarantee of $\min(m, \mathrm{rank}(A))^{(p-2)/(3p-2)}\mathrm{polylog}(1/\varepsilon)$ linear-system-solves. Notice that the dependence in the dimension term monotonically interpolates the two extremes in this range of $p$. Furthermore, notice that for $2 < p < \infty$, we get a high-accuracy guarantee, whereas in the $p=\infty$ case, we get a moderate-accuracy guarantee.
>
> > “non-robust and robust objectives may be computationally easier than optimizing for the robust objective alone”
>
> Please see the above discussion – by “easier”, we mean from a computational standpoint.
>
> > decomposition of equations (2) or (4) into subproblems
>
> This is just the algorithmic approach we choose to adopt, where we decompose the full problem into a sequence of optimization problems subject to a Euclidean geometry constraint. Please see the paragraph in lines 290-293 and the references therein for why this is a natural approach. We will add further citations and examples showing why this is natural.
>
> > line 356
>
> This is correct as written. The “inner” functions are those that are taken from the composition of the lse function with the smoothened Euclidean norms (see lines 324-330). We will add an equation reference to clarify this.
>
> > choose the radius $r_q$
>
> We state this in line 339, but will further highlight this.
>
> > “decreased by a constant factor”
>
> Your interpretation is correct. We will reword this to say something to the effect of “the diameter of our problem decreases so that we have $\|x_t - x^{\star}\|_M \le 0.5\|x_0 - x^{\star}\|_M$  ”.
>
> ### Comments about paper organization and title
>
> > include Algorithm 3 in the paper [...] move Table 1
>
> Reviewer fntP also pointed this out. We will be happy to include the algorithm in the main body of the paper, and will also move Table 1.
>
> >  block Lewis weights
>
> Thank you for the feedback. We will more explicitly allude in the introduction and in the abstract to how we use these in our solution.
>
> ### Other comments about writing and typos
>
> > typo in equation (4)
>
> Actually, this is correct as written. One sanity check is to plug in $p=2$ and $p=\infty$ and observe that these recover the extreme nonrobust and robust problems, as desired.
>
> > related work connecting DRO and regulraized regression [...] line 029
>
> Thank you for the pointers. We will be happy to discuss the relationship between the work you mentioned and ours.
>
> Finally, thanks for the remaining feedback about our writing. We will be happy to incorporate these in the final version.
>
> ---
>
> We thank the reviewer again for their appreciation of our contribution and for the positive assessment. We are working towards incorporating all their suggestions in our revision.

---

### Official Review · Reviewer_Q7n9 · 2025-11-01

**Soundness:** 3
**Presentation:** 3
**Contribution:** 4
**Rating:** 8
**Confidence:** 4

**Summary:**

In this paper, the author proposes a novel second-order algorithm based on accelerated proximal methods to solve the empirical group distributionally robust (GDR) least-squares problem, aiming to improve group-level fairness in linear regression.

The proposed algorithm achieves a (1+ε)-multiplicative optimal solution through a series of linear-system solves. Moreover, the author introduces a continuous interpolation between robust and non-robust optimization, which elegantly bridges the utilitarian and egalitarian objectives. Theoretical analyses are provided for both the accuracy bounds and the iteration complexity, allowing readers to compare the proposed method against existing optimization approaches.

From my perspective, the writing of this paper is clear, professional, and well-organized. The author explains the intuition and derivations of the algorithm in detail, and systematically contrasts it with classical methods. The theoretical sections are rigorous, with definitions and notation clearly presented. Additionally, the discussion of parameter roles and practical implications is thoughtful and accessible, making this paper almost textbook-level for readers who may not be deeply familiar with theoretical optimization.

**Strengths:**

1.	Elegant writing and well-structured mathematical presentation.
2.	Extensive and up-to-date literature review that situates the work within the broader DRO and optimization landscape.
3.	Careful and coherent technical overview that guides readers through the proofs.
4.	The paper clearly acknowledges limitations and points out meaningful future directions.

**Weaknesses:**

1.	Although the algorithm avoids geometry-dependent convergence and relies only on linear-system solves, its numerical accuracy remains lower than that of interior-point methods (IPM) in high-precision regimes.
2.	Given the complex form of the theoretical bounds, it would be valuable to include numerical experiments that empirically illustrate the relationship between ε and runtime or accuracy, and directly compare with IPM and Lewis.
3.	Many recent DRO frameworks adopt a regret-minimization perspective, which also leads to convex formulations with desirable properties. It would strengthen the paper if the author could discuss or contrast their approach with regret-based DRO methods.

**Questions:**

1. In some DRO frameworks, researchers may add more assumption on different groups, e.g. the weighted addition or Wasserstein distance. What will happen if we extend the algorithm to this area?
2. In real world, the number of group m is rarely greater than the number of features, which influences the Rank of A. So, will the gap between proposed method and standard log-barrier IPM be not so significant in practice?

---

> ### Author Response · Authors · 2025-11-30
>
> We thank the reviewer for the detailed and generous comments regarding the clarity, structure, and literature integration of the paper. We address the concerns below.
>
> ---
>
> ## Weaknesses
>
> > Although the algorithm avoids geometry-dependent convergence and relies only on linear-system solves, its numerical accuracy remains lower than that of IPM in high-precision regimes.
>
> We agree that interior-point methods attain a better complexity when the desired accuracy $\varepsilon$ is extremely small. However, even the existence of an efficient algorithm that achieves $d^{100}\mathrm{polylog}(1/\varepsilon)$ iteration complexity for general convex QP is an open question. Our result yields a guarantee that depends **only on the input dimension** and not on the geometry of the feasible region. This is already highly nontrivial.
>
> Our primary focus in this work is to design an algorithm for DRO linear regression that:
>
> - avoids condition-number dependence,
> - scales effectively in regimes with many groups $m$,
> - leverages low-rank structure in $A$ and the geometry of the loss,
> - and achieves improved dependence on $\varepsilon$ in the accuracy range that dominates practical ML systems.
>
> We will highlight this in the revision.
>
> Also, note that our Theorem 2 yields high accuracy, comparable to that of the IPM, in the setting of interpolating between the robust and nonrobust objectives. In that setting, we strictly improve over the traditional IPM and its analysis for the corresponding problem. The IPM only gives a guarantee of $m^{½}\mathrm{polylog}(1/\varepsilon)$ linear system solves, whereas our method gets $\min(m, \mathrm{rank}(A))^{(p-2)/(3p-2)}\mathrm{polylog}(1/\varepsilon) < min(m, \mathrm{rank}(A))^{1/3}\mathrm{polylog}(1/\varepsilon)$ system solves.
>
> ---
>
> > Given the theoretical bounds, numerical results would help illustrate the $\varepsilon$–runtime relationship and comparisons with IPM and Lewis.
>
> We agree. We implemented the algorithm on synthetic regression tasks and found that it performs competitively with IPM and first-order methods, both in terms of iteration complexity and wall-clock time. In practice, even the non-accelerated version improves over IPM when $m$ is large.
>
> We are preparing these experiments and aim to include them either in the December 3rd revision or, at the latest, in the camera-ready version.
>
>
> ---
>
> > Many recent DRO frameworks adopt a regret-minimization perspective. It would strengthen the paper to contrast with those methods.
>
> Thank you for the suggestion. We already briefly discussed this perspective in lines 66–74, where we interpret the problem as a min–max game. Regret-minimization approaches in this setting, however:
>
> - behave similarly to (or worse than) subgradient descent,
> - inherit condition-number dependence,
> - and do not match our dimension-only iteration guarantees.
>
> We will expand this discussion in the revision. These approaches are valuable conceptually, but they do not provide the type of geometry-independent or condition-free rates that our method achieves.
>
> ---
>
> ## Questions
>
> > In some DRO frameworks, researchers add assumptions on different groups (weighted sums, Wasserstein distance). What happens if we extend the algorithm?
>
> We appreciate the question. If the reviewer is referring to additional **structure relating the groups**—for example, weighted group losses or simple rescaling—then our framework extends naturally. Weighted group terms simply rescale the corresponding blocks of $A$, and the smoothing and strong-convexity arguments remain valid.
>
> For Wasserstein DRO or transport-based ambiguity sets, the situation is more subtle. We suspect some of our ideas will transfer (such as smoothing and proximal modeling) and generalize naturally, while others (especially the geometry construction) require new ideas. We will clarify this distinction.
>
> ---
>
> > Since in the real world $m < d$ is common, will the gap with log-barrier IPM be small?
>
> Not necessarily. While $m < d$ is common, the key driver of our iteration bound is **$\mathrm{rank}(A)$**, not the ambient dimension or the raw number of groups. When there is shared structure across clients—something that appears frequently in federated learning, fairness, and multi-distribution tasks—we often have $\mathrm{rank}(A) \ll d$. In this setting, our method improves even for moderate values of $m$.
>
> Moreover, the **large-$m$ regime is itself practically relevant**, especially in cross-device FL deployments where $m$ can reach tens of thousands or even millions. For such applications, exact least-squares regression remains a core primitive, and $m$-dependence becomes a central bottleneck. Our algorithm reduces this dependence to $\min\{m,\mathrm{rank}(A)\}^{1/3}$.
>
> ---
>
> We thank the reviewer again for their thoughtful comments and for highlighting the paper's strengths.

---

### Official Review · Reviewer_caGj · 2025-11-01

**Soundness:** 3
**Presentation:** 3
**Contribution:** 3
**Rating:** 4
**Confidence:** 2

**Summary:**

The paper presents condition-number-free algorithms for empirical group distributionally robust (GDR) least-squares regression and for an $\ell_p$-style interpolation between average and worst-group loss.
The main theoretical claims are:

1. An algorithm achieving a $(1+\varepsilon)$-approximate solution to
   $\displaystyle \min_x \max_i \frac{\|A_{S_i}x-b_{S_i}\|_2}{\sqrt{n_i}}$
   using $\tilde O(\min\{\mathrm{rank}(A),m\}^{1/3}\varepsilon^{-2/3})$ linear-system solves of the form $A^\top B A$ for block-diagonal $B$.

2. An extension to $\ell_p$-type interpolants between the average and robust objectives with iteration complexity $\tilde O(\mathrm{poly}(p)\min\{\mathrm{rank}(A),m\}^{(p-2)/(3p-2)})$.

**Strengths:**

The paper provides an algorithm that can solve empirical group distributionally robust least-squares regression with STOA complexities.

**Weaknesses:**

Honestly, I am not an expert in optimization theory. However, I think this paper may be more suitable for a journal such as *Mathematical Programming* rather than ICLR. Moreover, the paper does not provide any numerical results.

The complexity analysis depends on the oracle for solving linear systems, which may not be a fair comparison to the gradient evaluation oracle.

**Questions:**

na

---

> ### Author Response · Authors · 2025-11-25
>
> We thank the reviewer for their time and effort in reviewing our work. We respond to the reviewer’s concerns below.
>
> ---
>
> > However, I think this paper may be more suitable for a journal such as Mathematical Programming rather than ICLR.
>
> We appreciate the concern. While the paper introduces new optimization tools, the motivation comes directly from **practical machine learning settings** where group heterogeneity and robustness matter, such as fairness, multi-distribution learning, and cross-device federated learning. These problems appear frequently in ICLR, NeurIPS, and ICML, and recent work on group distributionally robust learning, $\ell_p$ regression, and distributed optimization using trust-region methods has been published in these venues. Some examples of such papers (this is not an exhaustive list) are:
>
>
> - [Global Linear and Local Superlinear Convergence of IRLS for Non-Smooth Robust Regression (Peng, Kümmerle, Vidal, NeurIPS 2022)](https://proceedings.neurips.cc/paper_files/paper/2022/file/ba3354bcfeae4f166a8bfe75443ac8f7-Paper-Conference.pdf)
>
> - [Coresets for Multiple $\ell_p$ Regression (Woodruff et al., ICML 2024)](https://openreview.net/pdf?id=4UWjqrMmFp)
>
> - [Acceleration with a Ball Optimization Oracle (Carmon et al., NeurIPS 2020)](https://proceedings.neurips.cc/paper/2020/hash/dba4c1a117472f6aca95211285d0587e-Abstract.html)
>
> - [A Stochastic Newton Algorithm for Distributed Convex Optimization (Bullins, Patel, Shamir, Srebro, Woodworth, NeurIPS 2021)](https://proceedings.neurips.cc/paper/2021/file/0bf40b1adf1b0b5c1418a0943eebfb45-Paper.pdf)
>
>
> Our algorithm targets *exactly* these ML settings: many groups, heterogeneous data, and the need for robustness guarantees that do not degrade with conditioning. For this reason, the problem and techniques are well aligned with the ICLR community.
>
> > Moreover, the paper does not provide any numerical results.
>
> The central contribution of this work is theoretical: we give a condition-free algorithm with iteration complexity that improves as data heterogeneity decreases, rather than scaling with the number of clients. That said, we have implemented the method on synthetic regression tasks and observed that it performs competitively with interior-point baselines and standard first-order methods. We are preparing these experiments and aim to include them either in the December 3rd revision or, at the latest, in the camera-ready version.
>
>
> > The complexity analysis depends on the oracle for solving linear systems, which may not be a fair comparison to the gradient evaluation oracle.
>
> We thank the reviewer for raising this question. Below, we clarify why the linear-system oracle is standard and meaningful for this problem class.
>
> **1. First-order methods suffer from poor conditioning.**
> Group DRO regression can easily become poorly conditioned due to heterogeneity across groups. As far as we are aware, no first-order method provides **condition-free** guarantees here. Although each gradient call is inexpensive, the total number of iterations can become very large when the data is heterogeneous. In contrast, linear-system solves allow the algorithm to avoid dependence on the condition number, which is essential in this setting.
>
> **2. This oracle model is standard in recent STOA work.**
> Work on $\ell_\infty$ regression, $\ell_p$ regression, and linear programming (e.g., Lee & Sidford 2019; Jambulapati et al. 2022; Adil et al. 2019, 2024) all measure complexity in terms of the number of solves of systems of the form $A^\top B A$, where $B$ is diagonal or block-diagonal. Our analysis uses the *same* complexity model and the same matrix structure.
>
> **3. Structure makes these solves faster in practice.**
> Because consecutive systems are closely related, inverse-maintenance techniques and sparsification methods can accelerate repeated solves. These improvements remain orthogonal to our contribution and can be applied directly. For these reasons, the linear-system oracle is a standard complexity measure for robust regression problems and aligns with the conventions of the state-of-the-art literature.
>
> ---
>
> We thank the reviewer again and hope that these clarifications address the concerns raised.

---

### Official Review · Reviewer_fntP · 2025-11-03

**Soundness:** 3
**Presentation:** 3
**Contribution:** 3
**Rating:** 6
**Confidence:** 3

**Summary:**

This paper develops an algorithm for 'distributionally robust linear regression'. Namely, it assumes m datasets for linear regression to be given, corresponding (for instance) to m distinct data sources. Instead of aggregating the data and fitting a linear model via least squares, the distributionally robust optimization (DRO) approach minimizes the maximum ell-2 error across the m datasets.
The paper also considers an objective that interpolates between the aggregate ell-2 error and the maximum of ell-2 error by averaging the p-th powers of ell-2 errors.
The new algorithm follows a general scheme from Carmon et. al (2020), and proceeds by solving a sequence of proximal problems defined on suitably constructed ellipsoidal trust regions.
The main technical innovations are a smoothening of the objective, and a new construction of the trust regions.

**Strengths:**

The problem is quite fundamental, and this paper improves over the state of the art.
The results appear to be sound and technically novel.

**Weaknesses:**

- The new method only improves over interior point methods (Lee ans Sidford, 2019) for large m
- It is unclear how practical is the algorithm, given the nested structure
- The authors apply a general framework from earlier work, and innovations is rather in specific aspects of the problem.

**Questions:**

1. A general suggestion. I think that the technical overview is the most interesting and important part of the paper. I would suggest to make it more precise and detailed, eventually stating auxiliary lemmas. In contrast, it would be sufficient to focus on the p=\infty case.

2. Are the complexity of the inner calls in IPM and in the present method comparable? They appear to require the solution of linear systems with somewhat different structure.

3. It would be useful to have an overall presentation of the algorithm. The definition is now dispersed across multiple sections.

Minor:
-I do not think the adjective "existential" is used correctly at the top of p 9

---

> ### Author Response · Authors · 2025-11-25
>
> We thank the reviewer for the careful reading and the constructive feedback. We respond to each point below.
>
> ---
>
> ## Weaknesses
>
> > The new method only improves over interior point methods for large m.
>
> We agree that the improvement is most visible when the number of groups is large. However, this regime is relevant for several applications, including [cross-device federated learning](https://arxiv.org/abs/1912.04977), where $m$ can reach tens of thousands or millions while the data size $n$ per client is relatively small. Since linear regression remains one of the basic workhorses in these systems, this setting is practically motivated.
>
> In such a regime, one of the main conceptual messages of this work is that the complexity of solving the robust problem depends more on the actual data heterogeneity than simply on the total number of groups. To illustrate this, consider a strawman with only one group, duplicated $m$ times. The IPM yields a $\sqrt{m}$ system-solve complexity, which can be very high. On the other hand, the complexity of our algorithm will scale proportionally to $rank(A)$ (consider $mn \gg d \gg n$).
>
> More generally, this highlights that the improvement is not limited to extremely large $m$. When the data has substantial shared structure and $\mathrm{rank}(A)$ is small, the iteration complexity
> $$
> O\left(\min(\mathrm{rank}(A), m)^{1/3}\varepsilon^{-2/3}\right)
> $$
> can improve over $\widetilde O(\sqrt{m})$ IPM complexity even when $m$ is moderate.
>
> We will highlight this point and include the duplication example more explicitly in lines 75-85 in Section 1.
>
> ---
>
> > It is unclear how practical the algorithm is, given the nested structure.
>
> We are unsure which nested structure the reviewer is referring to, as IPM, the default solver in CVXPY, also has one: the outer loop updates the barrier regularization, while the inner loop uses the Newton algorithm to solve the regularized problem. Furthermore, recall that our algorithm can be simplified to just repeatedly querying the ball oracle (see discussion in Section 2.2), leading to a slightly worse guarantee of $O\left(\min(\mathrm{rank}(A), m)^{1/2}\varepsilon^{-1}\right)$.
>
> If we misunderstood which nesting the reviewer was referring to, we are happy to clarify further.
>
> ---
>
> > The authors apply a general framework from earlier work, and innovation is rather in specific aspects of the problem.
>
> We agree that the proximal-oracle framework has roots in earlier work. However, applying this framework in our setting of group DRO linear regression is highly nontrivial. To do this, we require the following new technical components:
>
> - A **new composition lemma** (Lemma C.3) that proves quasi-self-concordance for softmax composed with non-linear “inner” functions, extending earlier results that only handled linear functions. We apply this with our new smoothing of the objective to show the requisite Hessian stability, which we need for the ball oracle implementation. Due to its generality, we expect that **this result is of independent interest**.
>
> - A **new strong convexity inequality** (Lemma D.3) for the $p$th power of the $\ell_2$ norm, which crucially allows us to relate function value suboptimality to argument suboptimality. This is necessary for us to exploit the problem's underlying geometric structure (i.e. relating function suboptimality to argument suboptimality within the geometry of our choice).
>
> We will emphasize this more clearly in the introduction.
>
> ---
>
> ## Questions
>
> > The technical overview is interesting. Please make it more precise and detailed, possibly by stating auxiliary lemmas.
>
> Thank you for the suggestion. We have already referred to the core technical lemmas necessary for our results and also discussed Lemma D.3 in the main paper. But we will expand Section 2 with more explicit statements of the central lemmas (including Lemma C.3) and provide a full walk-through of the $p=\infty$ case in the main text.
>
> ---
>
> > Are the complexities of the inner calls in IPM and in the present method comparable?
>
> Yes. Both methods require solving linear systems of the form $A^\top B A$, where $B$ is block-diagonal with blocks of size $n_i$.
>
> This structure arises in:
>
> - IPM: $A^{\top}BA$ is the structure of the Hessian of the log-barrier.
> - Our method: $A^{\top}BA$ is again the Hessian of our smoothed loss and is also the form of the block Lewis quadratic.
>
> Thus, the core primitive is identical. Only the number of solves differs. Our method uses $O\left(\min(\mathrm{rank}(A),m)^{1/3}\varepsilon^{-2/3}\right)$ calls, while IPM uses $O\left(\sqrt{m}\log(1/\varepsilon)\right)$.
>
> We state this equivalence in Table 1, which we will shift to feature more prominently along with our results.
>
> ---
>
> > It would be useful to have an overall presentation of the algorithm.
>
> We agree. We will move the pseudocode to the main body.
>
> ---

---

### Author Response · Authors · 2025-12-04

# Top-Level Response to All Reviewers

We thank all reviewers for their time and thoughtful assessments. The feedback will be extremely useful for us as we revise the paper. We have already incorporated the major comments in one revision (uploaded here) and will continue to polish the paper.

We are encouraged that the reviewers uniformly praised the soundness, presentation, and contribution of the work (each providing at least a score of 3), and that they agreed the problem addressed here is fundamental. Reviewers also noted the clarity of the exposition, the structure of the mathematical arguments, the quality of the literature review, and the coherence of the technical overview.

A few reviewers asked us about numerical results. Although this is mainly a theoretical work, we have included some preliminary experiments where we compare the wall-clock time/iteration complexity of our methods against standard techniques on a family of synthetic instances. We will add further trials to this in subsequent versions. Please see the following anonymized repository and included plots: https://anonymous.4open.science/r/dro-linear-regression-68CB/README.md

---

### Meta-Review · Area_Chair_w3dn · 2026-01-06

**Summary:**

This paper addresses distributionally robust linear regression. The reviewers highlight that the problem the paper addresses is important, and that the paper improves over the state of the art. There are some minor concerns about the practicality of the algorithms and how widely applicable these are. Overall, the reviewers suggest acceptance of the paper and I follow that suggestion.

**Reviewer Concerns:**

There are some minor concerns in terms of practicality which are somewhat addressed in the rebuttal.

**Reviewer Scores:**

I don't think that the reviewers would change their ratings. Reviewer caGj could potentially change their score but given the lack of expertise of the reviewer I do not believe that they would adapt their score.

---

### Decision · Program_Chairs · 2026-01-26

Accept (Poster)